



# A standardized database of Marine Isotopic Stage 5e sea-level proxies on tropical Pacific Islands

Nadine Hallmann[1], Gilbert Camoin[1], Jody M. Webster[2], Marc Humblet[3]

[1]Aix-Marseille Université, CNRS, IRD, INRAE, Coll France, CEREGE, Europôle Méditerranéen de l'Arbois,
BP80, 13545 Aix-en-Provence cedex 4, France
[2]Geocoastal Research Group, School of Geosciences, The University of Sydney, Sydney, New South Wales 2006,
Australia
[3]Department of Earth and Planetary Sciences, Graduate School of Environmental Studies, Nagoya University,
Nagoya 464-8601, Japan

*Correspondence to*: Nadine Hallmann (hallmann@cerege.fr)

**Abstract**

Marine isotope stage 5 deposits have been reported on many tropical Pacific islands. This paper presents a database compiled through the review of MIS 5e (Last Interglacial – LIG) coral reef records from islands belonging to French Polynesia (Anaa, Niau, Makatea, Moruroa, Takapoto, Bora Bora), the Hawaiian Islands (Oahu, Lana'i, Midway Atoll), Tuvalu, Kiribati (Christmas Island, Tarawa), the Cook Islands (Mangaia, Atiu, Mitiaro, Mauke, Pukapuka, Rakahanga, Rarotonga), Tonga, Samoa, the Federal States of Micronesia, the Mariana Islands, the Marshall Islands (Enewetak, Bikini), New Caledonia, Papua New Guinea, Vanuatu, Fiji and Niue. Studies reporting other sea-level indicators dated to other Pleistocene interglacials and Holocene sea-level indicators were not inserted in the database, but are included in this data description paper for completeness. Overall, about 300 studies concerning Pleistocene and Holocene sea-level indicators have been reviewed, and finally 284 data points from 35 studies on the MIS 5e have been inserted in the database. The main sea-level indicators include emerged coral reef terraces, but also reef units recovered in drill cores from a few islands, thus reflecting the diversity of tectonic settings and sampling approaches. Future research should be directed towards better constrained relative sea-level (RSL) reconstructions, including more precise chronological data, more accurate elevation measurements as well as a better refinement of the palaeo-water depth significance of coralgal assemblages. The database for Tropical Pacific Islands is available open-access at this link: http://doi.org/10.5281/zenodo.3991672 (Hallmann and Camoin, 2020).

## 1 Introduction

In this paper, we describe in detail the background information used to build the standardized database of tropical Pacific islands. This database was compiled as contribution to the World Atlas of Last Interglacial Shorelines (WALIS) ESSD Special Issue. The database was created using the WALIS interface, that is available at this link: https://warmcoasts.eu/world-atlas.html. The WALIS interface allows the standardized insertion into a mySQL database data on relative sea-level indicators and associated ages. The database presented in this study has been

downloaded from the interface as .xls file. The database presented here is open access and is available under this link: http://doi.org/10.5281/zenodo.3991672 (Hallmann and Camoin, 2020). Each field in our database is described at the following link: https://doi.org/10.5281/zenodo.3961543 (Rovere et al., 2020). The open access will enable the rapid and wide dissemination of the data that form the basement of this work and which in turn may trigger new studies in this field. The database presented here will be useful for scientists studying relative

sea-level changes, reef geology, tectonics and geodynamics of the Pacific region, as well as palaeoceanographers. More than 200 studies on MIS 5 coral reef records in the tropical Pacific have been reviewed. Of those, data from 35 studies have been included in the database. The geographical distribution of the studied islands from nine nations is shown in Figure 1. Overall, 281 U-series data points and three electron spin resonance data points have been inserted in the database. Those 284 data points have been summarized in 84 RSL indicators.

In the following, we first give an overview of the published literature related to LIG sea-level indicators in the tropical Pacific Islands, in order to give the reader a sense of the historical background upon which our review was done.  Then, we describe the types of sea-level indicators as well as the elevation measurement techniques and sea-level datums reported in literature. We then report a detailed description of the data points located in each administrative province/region within the area of interest, where sea-level data was reviewed (Section 5). Section

6.1 contains the description of a number of studies where the presence of different peaks in the LIG sea-level record was discussed in literature. While in our database we only reviewed specifically Marine Isotopic Stage 5e data, we encountered a number of studies where other Pleistocene shorelines were dated or described. These are listed in Section 6.2 and 6.3.

In the final two sections, we discuss further details on other metadata on palaeo-sea-level indicators that are not

included in our database, but that might be useful as research on Quaternary shorelines will progress in the tropical Pacific regions. We further present future research directions that may stem from our data compilation.



## 2 Literature overview

The sea-level database described by this article focuses on tropical Pacific islands, which cover most of the
intertropical realm of the Pacific Ocean from the Hawaiian archipelago to the North to New Caledonia to the
South, and from Mariana Islands and Papua New Guinea to the West to French Polynesia to the East.

LIG coral reef records of RSL change on tropical Pacific islands were mostly investigated on subaerially exposed
reef terraces. Since the pioneer work of Veeh (1966), which focused on the dating of corals from different islands
(Hawaiian Islands, French Polynesia, Cook Islands), extensive studies have been conducted in many islands across
this vast region. Some of these studies have used the LIG shorelines to reconstruct tectonic uplift, especially in
Papua New Guinea and Vanuatu. Submerged coral reef deposits related to the LIG, forming locally well-identified
terraces, have been investigated since the beginning of the 20th century and the first drilling in Funafuti (Tuvalu).
Drill cores including reef units related to the LIG have been collected between the 1950s and the 1980s on
Enewetak (Marshall Islands), Midway Atoll, Moruroa (French Polynesia) and Tarawa (Kiribati), and more recently
in New Caledonia and Bora Bora (French Polynesia).

All reviewed studies concerning LIG reef records from tropical Pacific islands (Fig. 1) are summarized hereafter.

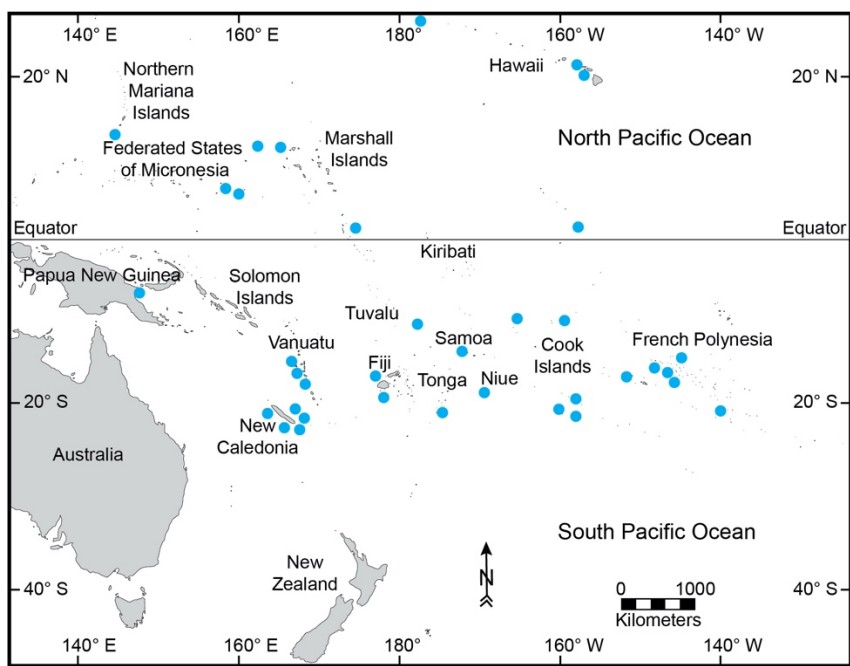

**Figure 1: Map showing the tropical Pacific islands considered in this article (blue dots).**

**French Polynesia** (Fig. 2): Since the early 1960s, the geology of Moruroa has been investigated through the drilling of a number of deep vertical and inclined boreholes carried out in the modern lagoon and on the reef rim. Lalou et al. (1966) and Labeyrie et al. (1969) studied reef units from the last 300,000 years. Trichet et al. (1984) described the general stratigraphy of Moruroa based on three cores. Camoin et al. (2001) dated four 300 m-long 80 drill cores from the reef rim and reconstructed RSL changes over the past 300,000 years, including the identification of MIS 5e. The most recent study on a Moruroa drill core by Braithwaite and Camoin (2011) focused on the study of carbonate sedimentology and diagenetic features of these cores.

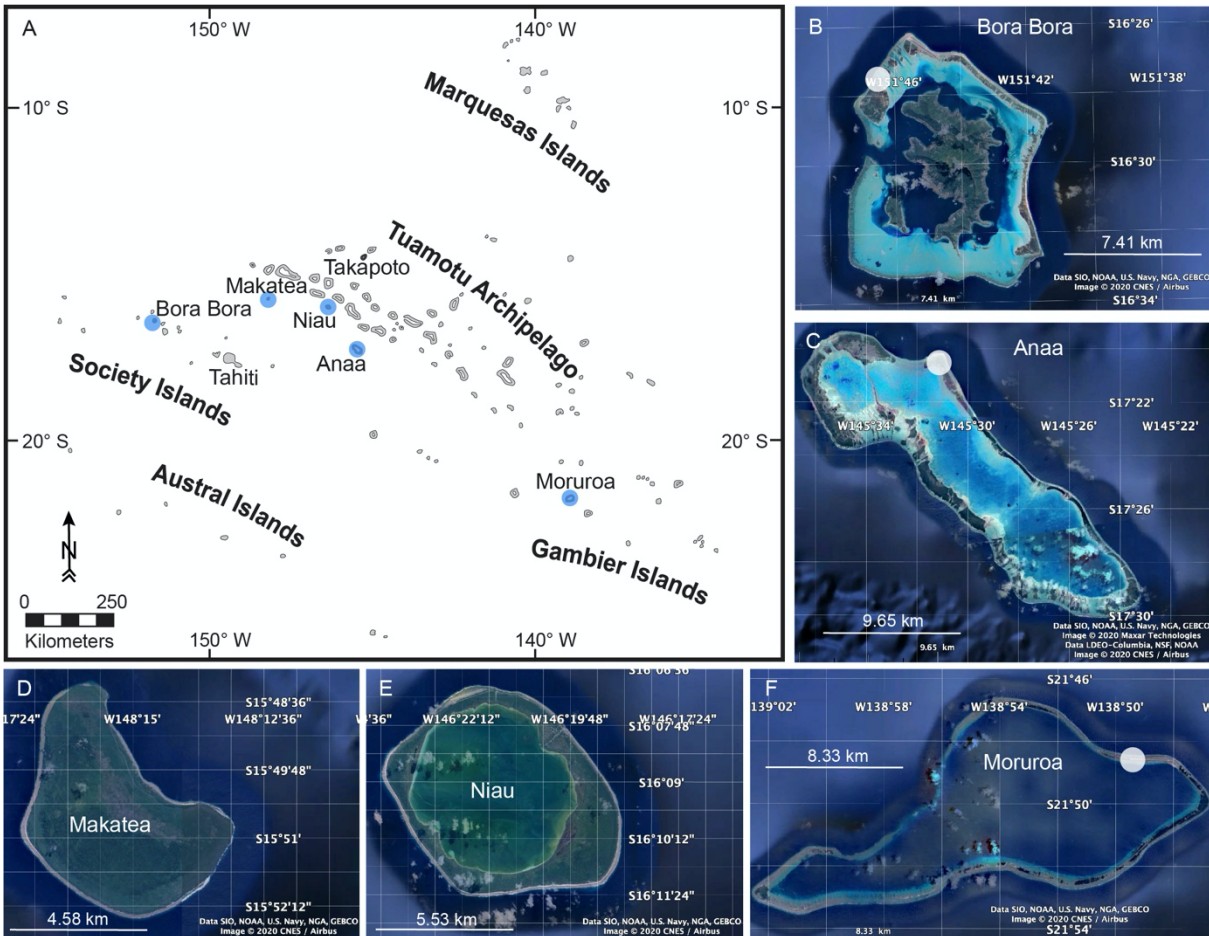

**Figure 2: Map showing islands in French Polynesia considered in this article (blue dots). Study sites are indicated by white circles. © Google Earth 2020 (Figs 2B-F).**




The earliest study on emerged Pleistocene coral reef terraces has been performed by Veeh (1966) who dated *in situ* fossil corals on Anaa, Niau and Makatea, followed by Pirazzoli et al. (1988a) who dated LIG reefs on Anaa.

Montaggioni (1985), Montaggioni et al. (1985) and Montaggioni and Camoin (1997) studied LIG reef terraces on Makatea. More recent studies by Montaggioni et al. (2018, 2019a, b) considered that some reef units have formed during Pleistocene high RSL, including MIS 5e, on Takapoto, but did not provide dating results. In addition, Gischler et al. (2016, 2019) reported the occurrence of a Pleistocene reef unit attributed to MIS 5e in Bora Bora drill cores.


**Cook Islands** (Figs 3 and 4): In 1910, Schofield reported on late Quaternary RSL changes on Rarotonga in the southern Cook Islands. The earliest study providing datings on LIG reef limestones in the southern Cook Islands (Mangaia) has been performed by Veeh (1966). Subsequent studies have been conducted by Stoddart et al. (1985), Spencer et al. (1987, 1988, 1989) and Stoddart et al. (1990). Woodroffe et al. (1991) worked on the stratigraphy and chronology of the southern Cook Islands and presented new U-series ages for Atiu, Mitiaro and Mauke. Gray

et al. (1992) studied the geochronology and subsurface stratigraphy of Pukapuka and Rakahanga in the northern Cook Islands based on drill cores from the lagoon.

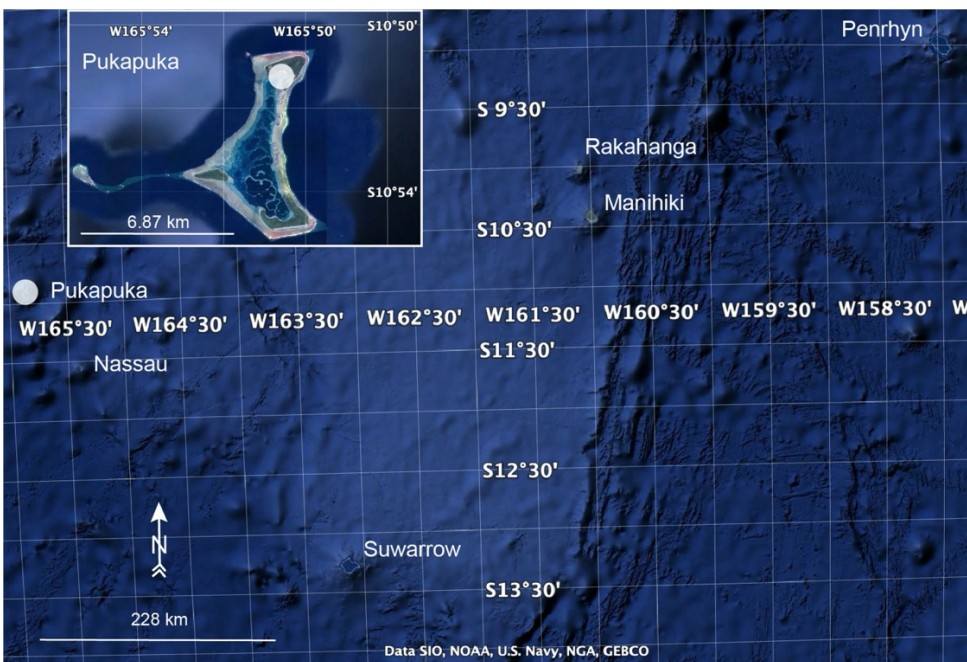

**Figure 3: Map showing the northern Cook Islands. Study site on Pukapuka is indicated by a white circle. © Google**
**Earth 2020.**

**Figure 4: Map showing islands in the southern Cook Islands considered in this article (white circles). Study sites are indicated by white circles. © Google Earth 2020.**



**Samoa**: The early study by Kear and Wood (1959) described LIG reefs in Samoa. In 1968, Stice and McCoy studied the geology of the Manu'a Islands and reported on the occurrence of carbonate deposits.

     **Niue** (Fig. 5): Different terraces on Niue have been explored at the beginning of the 20$^{th}$ century (Agassiz, 1903; David, 1904). Schofield (1959) identified seven terraces. A LIG reef sequence has been reported by Paulay (1988),

Paulay and Spencer (1992), Spencer and Paulay (1994) and Wheeler (2000). More recent studies further described the different terraces on Niue (Terry and Nunn, 2003; Nunn and Britton, 2004; Kennedy et al., 2012).

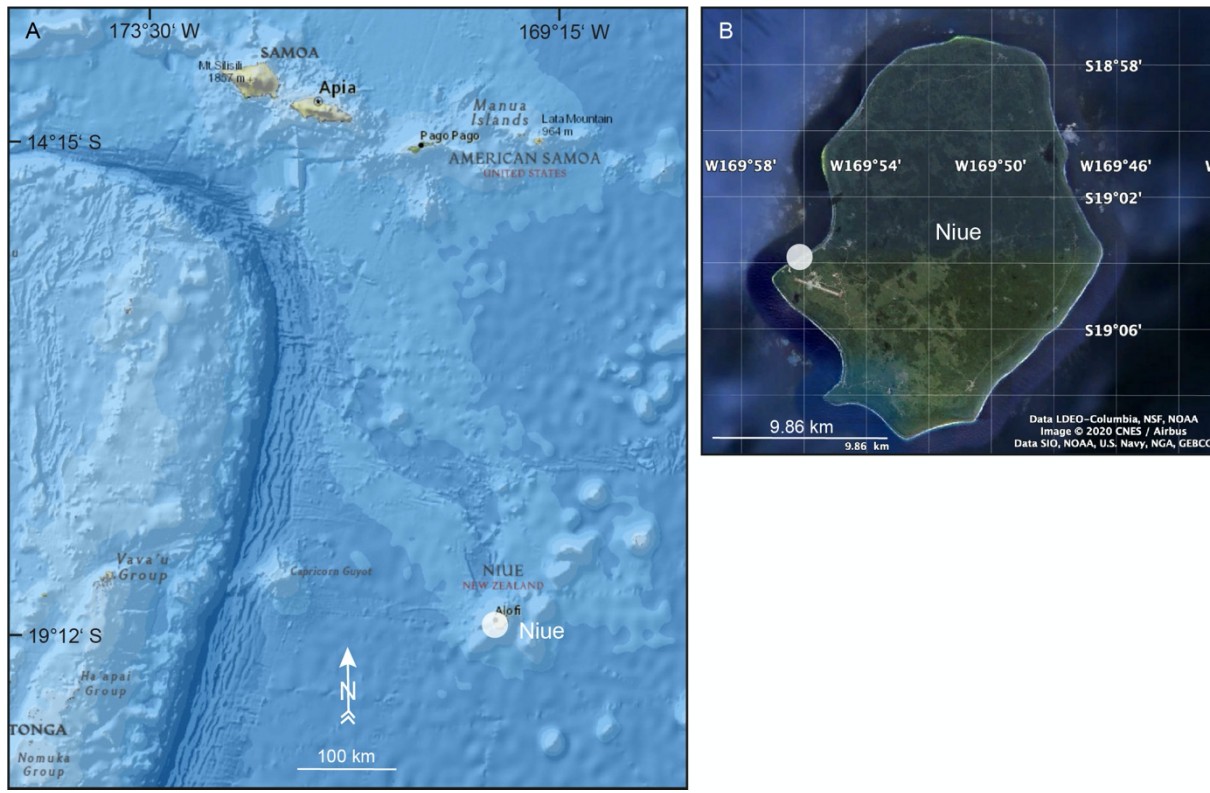

**Figure 5: Map showing the island of Niue. Study site on Niue is indicated by a white circle. Basemap: "National**
**Geographic Map", with data from National Geographic, Esri, Garmin, HERE, UNEP-WCMC, USGS, NASA, ESA, METI, NRCAN, GEBCO, NOAA, INCREMENT P Corp (Fig. 5A). © Google Earth 2020 (Fig. 5B).**

     **Tonga** (Fig. 6): Taylor (1978) and Yonekura (1983) have attributed reef deposits from the Tongatapu block to the LIG.




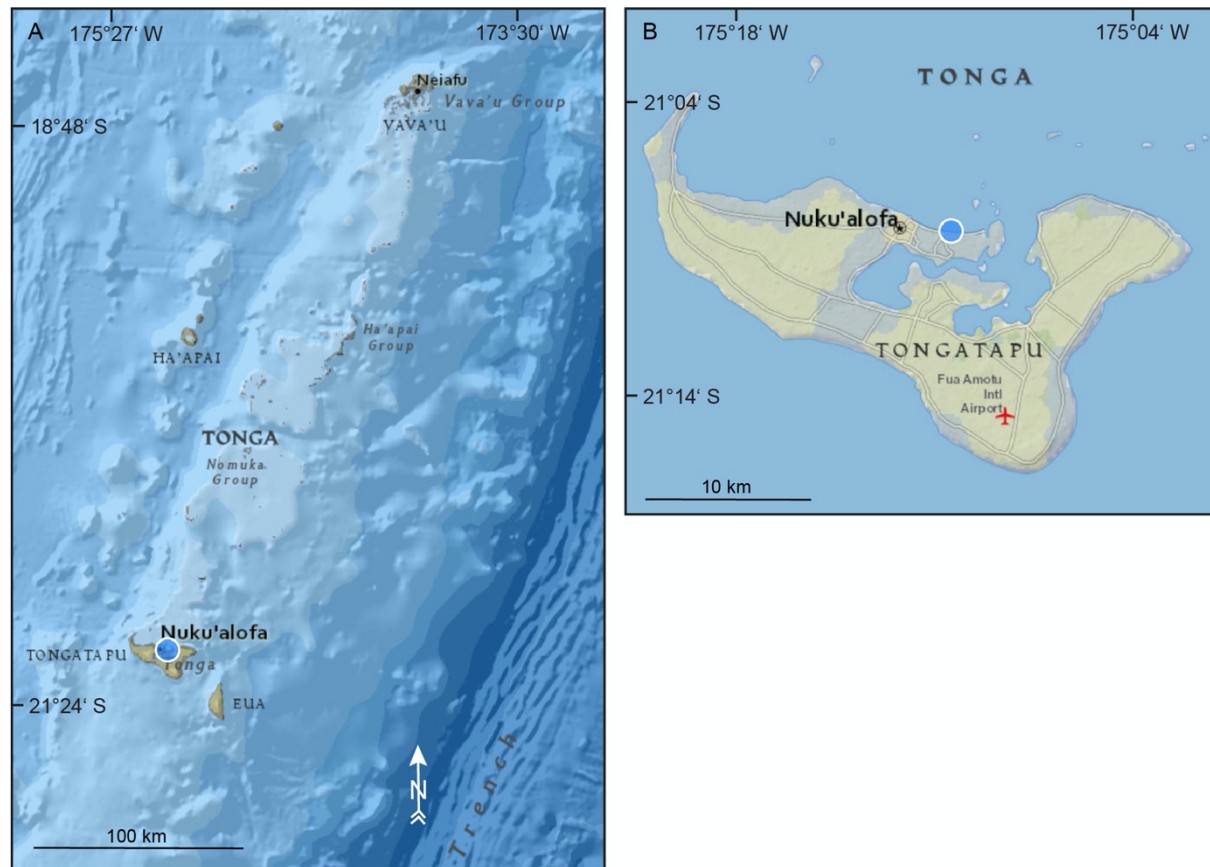

**Figure 6: Map showing Tonga. Study site on Tongatapu is indicated by a blue circle. Basemap: "National Geographic Map", with data from National Geographic, Esri, Garmin, HERE, UNEP-WCMC, USGS, NASA, ESA, METI, NRCAN, GEBCO, NOAA, INCREMENT P Corp.**


**Fiji** (Fig. 7): Despite pioneer geomorphological works by Dana (1872) and Moore (1889), and reports on late Quaternary RSL changes by Schofield (1910), LIG deposits from Fiji were reported for the first time in the late 1970s by Taylor (1978) who dated reefs on Yasawa Island. In the late 1990s, Nunn and Omura (1999) studied exposed reef terraces around Kadavu Island. The most recent studies by Nunn et al. (2002) and Nunn and Omura
(2003) summarized the levels of emerged LIG shorelines on the northeastern Fiji Islands and compared them to those from the western Fiji Islands.

**Figure 7: Map showing islands in Fiji considered in this article (white circles). Study sites on Kaibu Island are indicated by white circles. Basemap: "National Geographic Map", with data from National Geographic, Esri, Garmin, HERE,**

**UNEP-WCMC, USGS, NASA, ESA, METI, NRCAN, GEBCO, NOAA, INCREMENT P Corp (Fig. 7A). © Google Earth 2020 (Figs 7B-C).**





**Tuvalu**: Data regarding LIG deposits from Tuvalu are restricted to the 340 m-long drill core that was carried out by the Coral Reef Expedition of The Royal Society in 1896–1898 on Funafuti Atoll. This drill core was used to interpret the origin and history of atoll formation (Bonney, 1904) and test the Darwin's subsidence theory (Darwin,
1842). Despite a detailed study of these reef cores (Bonney, 1904), the core was originally interpreted as a modern reef deposit overlying Pleistocene limestone (Hinde, 1904) and no chronological frame was established before Ohde et al. (2002) who considered that LIG reef deposits must be deeper than 24.2 m below the modern surface. Since then, the Funafuti cores have been mentioned in general reviews regarding the development of coral reef islands (e.g. McLean and Woodroffe 1994; Spencer et al., 2008).


**Kiribati** (Fig. 8): In 1981, 30 m-long drill cores reaching Pleistocene limestone have been collected on Tarawa during a hydrogeological investigation by an Australian government team. These cores have been dated later by Marshall and Jacobson (1985). Woodroffe and McLean (1998) reported U-series ages from an outcrop of LIG limestone on Christmas Island.




**Figure 8: Map showing islands in Kiribati considered in this article (white circles). Study sites on Tarawa and Christmas Island are indicated by white circles. Basemap: "National Geographic Map", with data from National Geographic, Esri, Garmin, HERE, UNEP-WCMC, USGS, NASA, ESA, METI, NRCAN, GEBCO, NOAA, INCREMENT P Corp (Figs 8A-B). © Google Earth 2020 (Figs 8C-D).**




**Solomon Islands**: Stearns (1945) studied elevated benches in the South-West Pacific and correlated the different levels between the Solomon Islands, Vanuatu, the Mariana and Hawaiian Islands. The geology of the Solomon Islands was then studied in the 1950s by Grover (1958), while their geomorphology has been investigated later by Stoddart (1969) who reported on elevated shorelines, terraces and tidal notches on different island groups.

**Vanuatu** (Fig. 9): Uplifted Quaternary reef terraces on Vanuatu islands, specifically Santo, Malakula and Efate, have been identified and partially described since the late 1960s and the 1970s (Mitchell, 1966, 1969, 1971; Robinson, 1969; Mallick and Greenbaum, 1977; Ash et al., 1978). After these first descriptions, a reconnaissance study has been carried out by Neef and Veeh (1977) who published the first U-series ages on Malo and Efate, immediately followed by Bloom et al. (1978) who focused on the dating of LIG reefs from Efate. In the 1980s, the vertical movements in the New Hebrides (Vanuatu) Island Arc have been reconstructed based on the dating of the uplifted coral reef terraces and several investigations provided additional dating results on terraces from Santo, Malakula, Efate and Torres Islands (Jouannic et al. 1980, 1982; Lecolle and Bernat, 1985; Taylor et al., 1980, 1985). Taylor et al. (1987) compared contemporary coseismic and nonseismic with quaternary vertical movements based on the analysis of partially emerged corals and reef terraces in the central Vanuatu Arc. Edwards et al. (1987a, b) have provided precise chronological data on exposed reef terraces from Efate. Chronological data obtained on uplifted coral reef terraces have been used by Taylor et al. (2005) to reconstruct rapid vertical movements in the New Hebrides forearc. Since the beginning of the current century, additional studies have been carried out on Malakula (Cabioch and Ayliffe, 2001) and Tanna (Neef et al., 2003) and significantly advanced the chronological frame of Pleistocene raised reefs. On Tanna, the studied reefs and raised lagoonal deposits range in age from the Holocene to MIS 7 (Neef et al., 2003).



**Figure 9: Map showing islands in Vanuatu considered in this article (green and orange circles). Study sites are indicated by white and green circles. Basemap: "National Geographic Map", with data from National Geographic, Esri, Garmin, HERE, UNEP-WCMC, USGS, NASA, ESA, METI, NRCAN, GEBCO, NOAA, INCREMENT P Corp.**

**New Caledonia** (Fig. 10): Early studies on Pleistocene reef terraces, including the LIG, have been carried out in the 1970s and early 1980s by Launay and Récy (1972), Coudray (1976), Bernat et al. (1976), Marshall and Launay (1978) and Gaven and Bourrouilh-Le Jan (1981), and were focused on their dating and use for the reconstruction of neotectonic movements in the Loyalty Islands (Maré, Lifou and Ouvéa). These results have been summarized by Maurizot and Lafoy (2003) in their geological map of Maré Island. The drilling of modern reefs from the western coast of New Caledonia in the 1970s (Coudray, 1971), late 1990s (Cabioch et al., 1996, 1999) and after 2000 (Frank et al., 2006; Cabioch et al., 2008; Montaggioni et al., 2011; Hongo and Wirrmann, 2015) aimed at reconstructing the development pattern of fringing and barrier reefs. Drill cores from the Chesterfield Islands that are located about 500 km to the WNW of New Caledonia have been studied with the same objective by Degaugue-Michalski (1993).

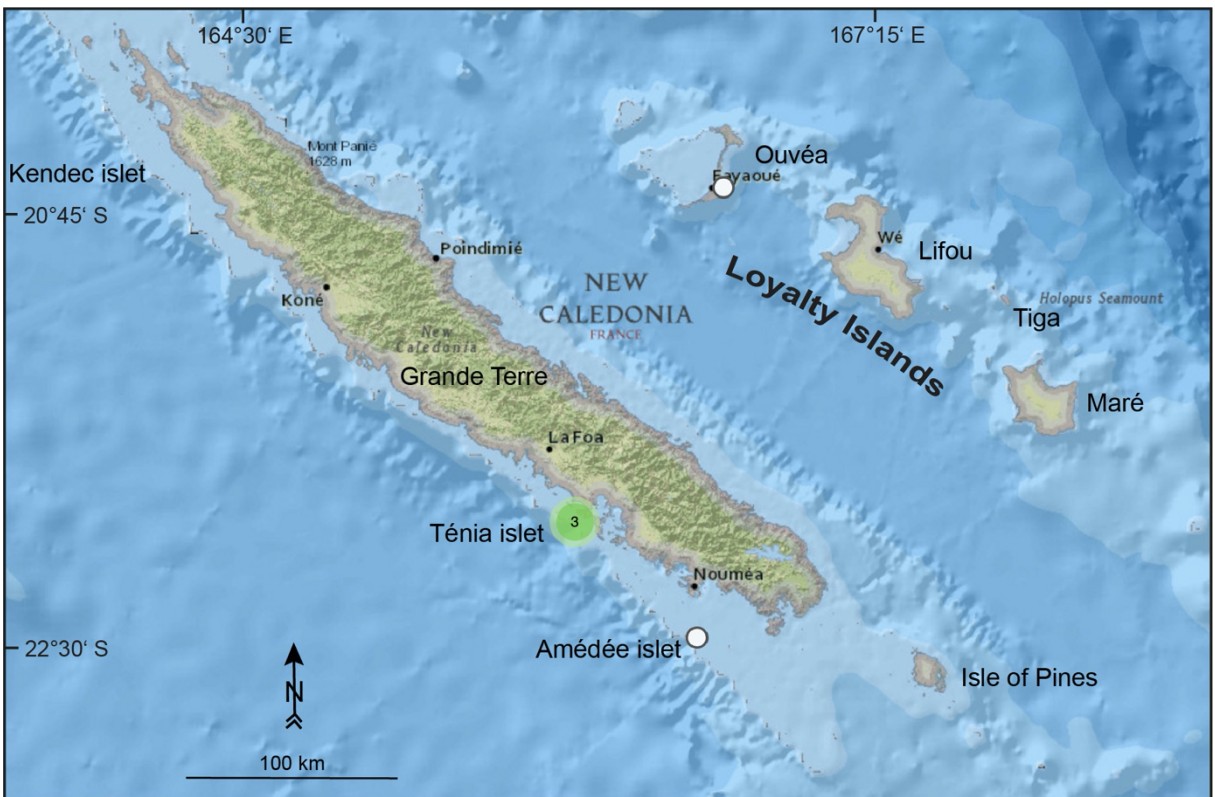

**Figure 10: Map showing islands in New Caledonia considered in this article (green and white circles). Basemap: "National Geographic Map", with data from National Geographic, Esri, Garmin, HERE, UNEP-WCMC, USGS, NASA, ESA, METI, NRCAN, GEBCO, NOAA, INCREMENT P Corp.**

**Papua New Guinea** (Fig. 11): Uplifted Quaternary reef terraces from Huon Peninsula (Papua New Guinea) represent a classical site of RSL change studies in support of the astronomical theory of climate change. Early

studies and first attempts to establish an accurate chronological frame for these coral reef terraces were carried out in the 1970s with Veeh and Chappell (1970), Chappell (1974) and Bloom et al. (1974). Since then, the uplifted terraces from Huon Peninsula have been regularly investigated with the same general objective in the 1980s by Aharon (1983), Chappell (1983) and Aharon and Chappell (1986), in the 1990s by Stein et al. (1993) and Esat et al. (1999) and, in the early 2000s, by Cutler et al. (2003).


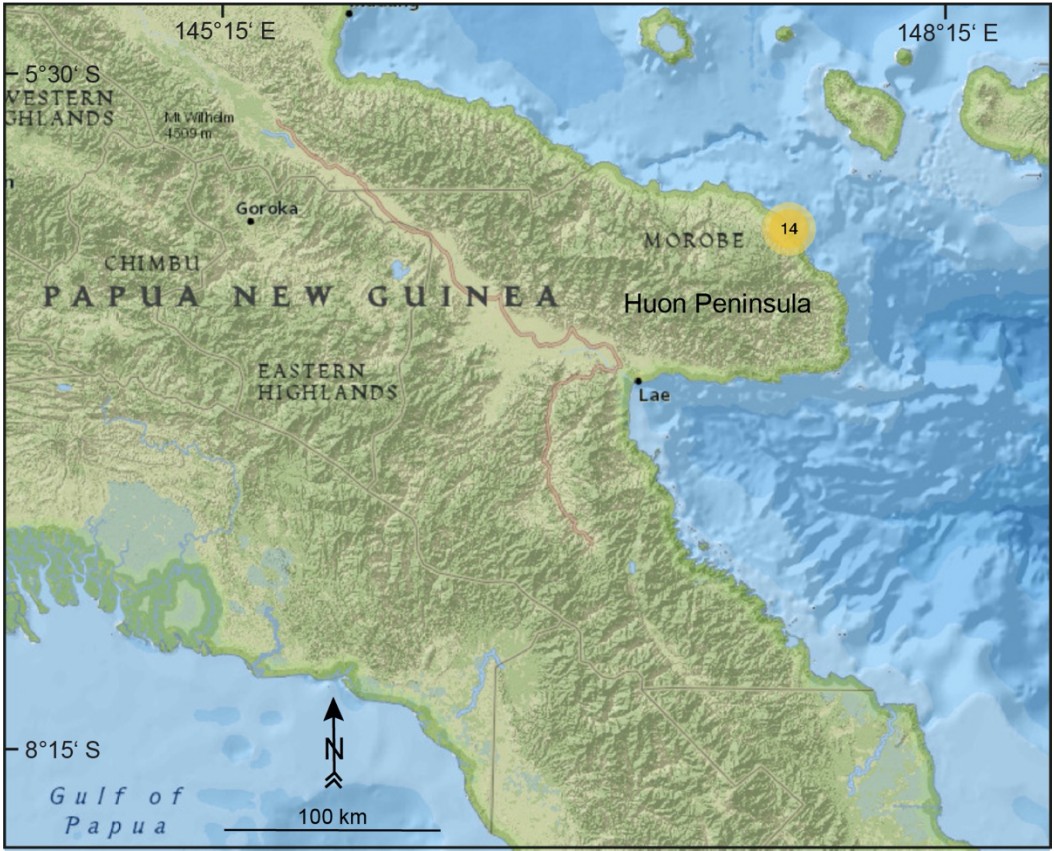

**Figure 11: Map showing study sites on the Huon Peninsula, Papua New Guinea, considered in this article (orange circle). Basemap: "National Geographic Map", with data from National Geographic, Esri, Garmin, HERE, UNEP-WCMC, USGS, NASA, ESA, METI, NRCAN, GEBCO, NOAA, INCREMENT P Corp.**






**Federated States of Micronesia**: Ayers and Vacher (1986), Anthony (1996) and Fletcher and Richmond (2010) mentioned Pleistocene limestones, including the LIG period, in this region.

**Mariana Islands**: The geologic history of Guam was first studied in 1937 by Stearns who then reported in 1940 on the deposition of Pleistocene limestone on volcanic substrate. Stearns (1945), Tayama (1952) and Emery (1962) described 'shore benches' related to sea level. In 1954, Cloud correlated stratigraphic sections throughout the West Pacific, including Guam. In the early 1950s, a programme by the Corps of Engineers, U. S. Army and the U. S. Geological Survey has focused on geology, soils and water of Guam (Tracey et al., 1959, 1964), including a

detailed description of submarine and emerged coral reef terraces. Lower terraces on Guam, less than 15 m above modern sea level, are comparable to limestones that were described at the same elevation on Saipan (Tracey et al., 1964) and assigned to the late Pleistocene (Cloud et al., 1956). Further studies on Guam reef terraces included their lithological description (Emery, 1963), the provision of U-series ages obtained on MIS 5e limestone and estimated tectonic movements (Randall and Siegrist, 1996) and the reconstruction of RSL changes since MIS 5e

(Miklavič et al., 2012).

**Marshall Islands**: Five drill cores extracted from Bikini atoll in 1947 were first studied by Ladd et al. (1948) who described subaerial horizons in those cores. In 1954, Emery et al. further described cores from Bikini. Drillings on Enewetak have been performed in the early 1950s and drill cores have been studied by Ladd et al. (1953), Potratz

et al. (1955), Ladd and Schlanger (1960), Schlanger (1963) and Thurber et al. (1965). Schlanger (1963) has shown similarities between cores from Enewetak and Bikini. Two shallow drilling programmes have been initiated in the early 1970s: PACE (1971-1972; Henry et al., 1974) and EXPOE with 46 holes down to 90 m on Enewetak (1973-1974, Couch et al., 1975). Couch et al. (1975) confirmed the occurrence of subaerial unconformities in many drill cores. Tracey and Ladd (1974) described unconformities in the Bikini drill core, which could be partly also

recognized in Enewetak drill cores. Szabo et al. (1985) presented dating results from two drill cores extracted from the Enewetak reef flat during the PACE Program. During the Pacific Enewetak Atoll Crater Exploration (PEACE) Program (1984-1985), the U.S. Geological Survey investigated two craters formed by nuclear detonations in 1958 (Henry et al., 1986; Henry and Wardlaw, 1990). Ludwig et al. (1988) studied a 350 m-long drill core of Neogene lagoonal, shallow-water carbonates from Enewetak. Quinn and Matthews (1990), Wardlaw and Quinn (1991) and

Quinn (1991) studied the post-Miocene sea-level history of Enewetak.





**Hawaii** (Fig. 12): Stearns (1935a, b, c) and Wentworth and Hoffmeister (1939) first studied Pleistocene shorelines on the islands of Oahu and Maui. In 1960, drill cores have been collected on Midway Atoll and studied in the following decades (e.g. Ladd et al., 1967, 1970; Wells, 1982). Stearns and Chamberlain (1967) described deep

cores extracted from Oahu to better understand the geologic history. Several studies provided ages and demonstrated that the Waimanalo limestone on Oahu correspond to the LIG. Veeh (1966) and Ku et al. (1974) provided the first dating results of those emerged terraces. Further chronological studies concerning the Waimanalo deposits included those from Sherman et al. (1993), Muhs and Szabo (1994) and Szabo et al. (1994). In 2002, Muhs et al. performed additional datings with a higher precision at the Szabo et al. (1994) study sites.

The latest ages of reef deposits from Oahu have been reported by McMurtry et al. (2010). Rubin et al. (1995, 2000) provided U-series data for the Hulopoe gravel on Lana'i containing MIS 5e and 7 corals.



**Figure 12: Map showing islands in Hawaii considered in this article (green and orange circles). Study sites on Lana'i**
**and Oahu are indicated by blue and green circles. Basemap: "National Geographic Map", with data from National**
**Geographic, Esri, Garmin, HERE, UNEP-WCMC, USGS, NASA, ESA, METI, NRCAN, GEBCO, NOAA,**
**INCREMENT P Corp.**



**3 Sea-level indicators**

The main features associated with LIG RSL on tropical Pacific islands are exposed coral reef terraces. In addition, submerged coral reef deposits related to the LIG, forming locally well-identified terraces, have been drilled on

land on several islands, such as Funafuti (Tuvalu), Enewetak (Marshall Islands), Midway Atoll, Moruroa and Bora Bora (French Polynesia), Tarawa (Kiribati) and New Caledonia.

| Name of RSL indicator | Description of RSL indicator | Description of Reference Water Level | Description of Indicative Range | Indicator reference(s) |
|---|---|---|---|---|
| Coral reef terrace (general description) | Coral-built flat surface, corresponding to shallow-water reef terrace to reef crest. The definition of indicative meaning is derived from Rovere et al., 2016, and it represents the broadest possible indicative range, that can be refined with information on coral living ranges. | (Mean Lower Low Water + Breaking depth)/2 | Mean Lower Low Water - Breaking depth | Rovere et al., 2016 |

**Table 1: Different types of RSL indicators reviewed in this study.**

In general, paleo RSL is determined from the average elevation of the terrace or, if present, from the elevation of the highest *in situ* corals which are usually found on the paleo reef crest (Rovere et al., 2016). However, the identification of coral assemblages that characterize coral reef terraces and the definition of the relevant water-depth range estimates may provide a more accurate indication of the paleo RSL. Single corals and coral assemblages have been described in some studies related to LIG reefs on tropical Pacific islands. Since their

palaeo-water depth significance has been barely reported in the relevant studies, we have compiled data published in two databases: OBIS (Ocean Biodiversity Information System; https://obis.org) and IUCN (International Union for Conservation of Nature; https://www.iucnredlist.org), thus following a similar approach to the one used by Hibbert et al. (2016, 2018). The datasets used are listed on the OBIS and IUCN websites and summarized on Table 2.

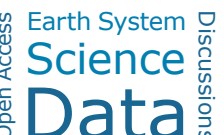

Concerning the OBIS database, data were extracted using the Mapper tool. Two filters were used on Mapper: (1) Time range: 1990-2020, and (2) Depth range: 0-150 (to eliminate potential aberrant depths). Records which had the following characteristics were excluded from the analysis: (1) those for which the difference between Max Depth and Min Depth were > 0,2 m, (2) those for which the "basis of record" was not "human observation" or "machine observation" (i.e., collection specimens were excluded).

All available data were used to define palaeo-water depth intervals for each species (Table 2). However, the data that have been compiled from OBIS and other databases cover the whole Indo-Pacific region. The value that have been calculated for average and median depths can therefore differ when considering specific islands or areas of the studied area.

In most cases, it has been difficult to assign a palaeo-water depth based on a single species, as most corals have a

wide depth range. Accurate palaeo-water depth intervals can be usually defined based on coral assemblages. We have primarily selected data for which a precise depth, not a range, was available in the OBIS database.

For some species, which were not identified with certainty, we used close species to define a depth interval (e.g., for *Acropora* cf *aspera*, we have compiled data concerning close species that belong to the 'aspera group' of Wallace, 1999: *A. aspera, A. pulchra*, and *A. millepora*).

*Porites* is the most frequent coral reported in LIG reefs from tropical Pacific islands. However, it cannot be generally used to constrain the palaeo-water depth interval as it displays a wide depth range when it is not identified at the species level. The morphology of *Porites* colonies may give nevertheless some indications on a gross palaeo-water depth distribution.






| | OBIS Database | | | | IUCN Database | |
|---|---|---|---|---|---|---|
| | Median depth | Average depth | Min depth | Max depth | Min depth | Max depth |
| *Acropora aspera* | 3.0 | 3.0 | 0.0 | 11.0 | 0.0 | 5.0 |
| *Acropora millepora* | 3.0 | 3.9 | 0.0 | 12.0 | 2.0 | 12.0 |
| *Acropora pulchra* | 3.0 | 4.0 | 0.0 | 18.0 | 1.0 | 20.0 |
| *Cyphastrea serailia* | 4.5 | 5.2 | 0.0 | 24.9 | 0.0 | 50.0 |
| *Dipsastraea* | 6.0 | 11.1 | 0.0 | 60.0 | | |
| *Dipsastraea laxa* | 7.0 | 5.9 | 0.0 | 11.0 | 1.0 | 25.0 |
| *Gardineroseris planulata* | 3.0 | 5.9 | 0.0 | 50.0 | 2.0 | 30.0 |
| *Goniastrea retiformis* | 8.0 | 17.9 | 0.0 | 53.0 | 0.0 | 20.0 |
| *Goniastrea stelligera* | 0.0 | 13.6 | 0.0 | 91.0 | 0.0 | 20.0 |
| *Leptastrea* | 7.0 | 16.8 | 0.0 | 86.0 | | |
| *Leptoria* | 4.0 | 7.5 | 0.0 | 86.0 | | |
| *Leptoria phrygia* | 4.0 | 5.0 | 0.0 | 47.0 | 0.0 | 30.0 |
| *Oulophyllia crispa* | 6.0 | 6.4 | 0.0 | 50.0 | 2.0 | 30.0 |
| *Platygyra* | 4.0 | 6.9 | 0.0 | 84.0 | | |
| *Platygyra lamellina* | 4.0 | 4.4 | 0.0 | 27.4 | 1.0 | 30.0 |
| *Platygyra sinensis* | 5.0 | 5.0 | 0.0 | 17.0 | 0.0 | 30.0 |
| *Plesiastrea versipora* | 6.0 | 5.8 | 0.0 | 30.0 | 1.0 | 40.0 |
| *Pocillopora* | 0.0 | 15.6 | 0.0 | 142.0 | | |
| *Porites** | 0.0 | 18.5 | 0.0 | 142.0 | | |
| *Porites lobata* | 0.0 | 19.2 | 0.0 | 98.0 | 0.0 | 30.0 |
| *Porites lutea* | 0.0 | 2.7 | 0.0 | 84.0 | 0.0 | 30.0 |

* Data for Indo-Pacific region only

**Table 2: Modern depth distribution (in meters below MSL) of corals identified in LIG records from the tropical Pacific islands. This table is based on two databases OBIS (Ocean Biodiversity Information System; https://obis.org) and IUCN (International Union for Conservation of Nature; https://www.iucnredlist.org).**





## 4 Elevation measurements

Almost all studies used auto/hand level to measure elevations of RSL indicators (Table 3); fewer studies used theodolite and rod or meter tape and rod. The elevation measurement technique has not been reported in many of the reviewed studies, but was most probably hand level or metered tape (Table 3). The sea-level datums reviewed 325 in this study include Mean Sea Level (MSL), Mean High Tide (MHT), Mean Low Tide (MLT), Mean Low Water Springs (MLWS) and modern reef (Table 4). In several studies the sea-level datum has not been reported, and has been assumed to be MSL (Table 4).

| Measurement technique | Description | Typical accuracy |
|---|---|---|
| Total station or Auto/hand level | Total stations or levels measure slope distances from the instrument to a particular point and triangulate relative to the XYZ coordinates of the base station. The accuracy of this process depends on how well defined the reference point and on the distance of the surveyed point from the base station. Thus, it is necessary to benchmark the reference station with a nearby tidal datum, or use a precisely (DGPS) known geodetic point. The accuracy of the elevation measurement is also inversely proportional to the distance between the instrument and the point being measured. | ±0.1/±0.2m for total station ±0.2/±0.4 m for hand level |
| Theodolite and rod | Elevation derived from triangulation with a theodolite. | Usually very precise, centimetric accuracy, depending on distance. |
| Barometric altimeter | Difference in barometric pressure between a point of known elevation (often sea level) and a point of unknown elevation. Not accurate and used only rarely in sea-level studies | Up to ±20% of elevation measurement |
| Cross-section from publication | The elevation was extracted from a published sketch/topographic section. | Variable, depending on the scale of the sketch or topographic section. |
| Metered tape or rod | The end of a tape or rod is placed at a known elevation point, and the elevation of the unknown point is | Up to ±10% of elevation measurement |



| | calculated using the metered scale and, if necessary, clinometers to calculate angles. | |
|---|---|---|
| Not reported | The elevation measurement technique was not reported, most probably hand level or metered tape. | 20% of the original elevation reported added in root mean square to the sea level datum error |

**Table 3: Measurement techniques used to establish the elevation of MIS 5e shorelines in the tropical Pacific islands.**

| Datum name | Datum description | Datum uncertainty |
|---|---|---|
| Mean Sea Level / General definition | General definition of MSL, with no indications on the datum to which it is referred to. | A datum uncertainty can be established on a case-by-case basis. |
| Modern reef | Upper surface of modern reef | A datum uncertainty may be established on a case-by-case basis. |
| Mean High Tide (MHT) | From NOAA definitions: "It is obtained by subtracting the mean of all the high waters from the mean of the higher high waters. One-half the average difference between the two low waters of each tidal day observed over the National Tidal Datum Epoch." | Depending on the quality of tidal data used. |
| Mean Low Tide (MLT) | From tideschart.com: "The Mean Low Tide (or Mean Low Water) is the average height of all low tides in a given place, deriving from a long series of observations (NTDE) of all levels of low tide in that spot." | Depending on the quality of tidal data used. |
| Mean Low Water Springs (MLWS) | From www.coastalwiki.org: "The height of mean low water springs is the average throughout a year of the heights of two successive low waters during those periods of 24 hours (approximately once a fortnight) when the range of the tide is greatest." | Depending on the quality of tidal data used. |

**Table 4: Sea-level datums reviewed in this study.**



## 5 Relative sea-level indicators

Because they correspond to easily accessible coral reef archives of sea-level change on many tropical and subtropical coasts, the LIG terraces have received much attention since the seminal works from Mayer (1917) and

Daly (1920) on Samoa and other Pacific islands (Camoin and Webster, 2015).

Most of the tropical Pacific islands are underlain by the large northwestward-moving Pacific plate that is generated at the East Pacific Rise and the Pacific-Antarctic Ridge (see Fig. 1 in Neall and Trewick, 2008). The archipelagos and islands from the western part of the studied area developed in a complex geodynamic setting resulting from the relative motions of the Pacific and the Australian and Eurasian plates.

The sixteen archipelagos and islands reported in the database illustrate the exceptional diversity of tropical Pacific islands, which is mainly related to their dynamic and complex geological history. They have originated as linear chains of volcanic islands on the above plates, atolls, uplifted coral reef islands, fragments of continental crust, obducted portions of adjoining lithospheric plates, and have also resulted from subduction along convergent plate margins (Neall and Trewick, 2008). This complex history has generated a great diversity of tectonic movements

(uplift vs subsidence), which have guided the initiation and the evolution of LIG reefs described below.

### 5.1 French Polynesia

LIG reef deposits have been reported from outcrops on several Polynesian islands and from drill cores collected on Moruroa and Bora Bora.

The first dates on LIG reef terraces from French Polynesia, which can be found at elevations of +4 to +5 m, have

been obtained by Veeh (1966) who dated *in situ* fossil corals from emerged Pleistocene reef terraces on Anaa, Niau and Makatea. *Leptoria* specimens at elevations of +2 to +4 ± 0.5 m MLWS on Anaa have been dated at 110 ± 20 ka, 140 ± 30 ka and 150 ± 40 ka (WALIS U-series IDs 1731-1733; ANAA5, ANAA15 and ANAA16). However, the use of the alpha-counting technique resulted in large age uncertainties.

Pirazzoli et al. (1988a) used electron spin resonance (ESR) to date the 4-5 m-high raised reefs from Anaa (see Fig.

2 in Pirazzoli et al., 1988a). A *Leptoria*? coral colony in growth position with an aragonite content of 97% yielded an age of 109 + 8 / -4 ka (WALIS ESR ID 103; 3AN14). The authors reported an elevation of 3.85 ± 0.2 m above MSL and a palaeo-MSL at an elevation of +4.15 ± 0.2 m. Modern *Leptoria* is living at a median and average depths of 4 and 7.5 m, respectively (see Table 2). Consequently, the palaeo-RSL based on those four *Leptoria* specimen from Anaa is assumed to have been at +8.75 ± 2.3 m (WALIS RSL ID 548).

An emerged reef terrace at the same elevation has been dated by Veeh (1966) on Niau and Makatea. Two *Leptoria* specimens at hand-leveled elevations of +3 to +4 ± 0.5 m MLWS on Niau have been dated at 160 ± 40 ka and 120

± 20 ka (WALIS U-series IDs 1735 and 1736; NIAU4 and NIAU6). The palaeo-RSL on Niau is assumed to have been at +9.25 ± 2.0 m (WALIS RSL ID 549). *Favia* and *Leptoria* on Makatea at an elevation of +3 ± 0.5 m MLWS have yielded ages of 140 ± 30 ka and 100 ± 20 ka (WALIS U-series IDs 1737-1738; MAKA2 and MAKA3). The

palaeo-RSL on Makatea is assumed to have been at +8.75 ± 1.8 m (WALIS RSL ID 550). In 1997, Montaggioni and Camoin identified two well defined Pleistocene terraces on Makatea, one of which is at an elevation of +7 m and can be related to the 125-ka RSL highstand. It can be assumed that Makatea reached relative vertical stability just prior to the 120-ka marine transgression (Montaggioni et al., 1985; Montaggioni and Camoin, 1997).

LIG reef deposits have been reported in drill cores collected in Moruroa (Camoin et al., 2001) and Bora Bora

(Gischler et al., 2016). The reef units related to the LIG in Moruroa drill cores (see Figure 13 and Figure 2 in Braithwaite and Camoin, 2011) are mainly composed of coralgal frameworks built by an *Acropora gr. danai-robusta / Porolithon onkodes* assemblage, which typifies a very shallow depositional environment (i.e. shallower than 6 m). However, only one out of six datings has been accepted (Camoin et al., 2001) and concerns a reworked colony of *Acropora* specimen at 92.7 m below MSL yielding an age of 124.4 ± 2 ka (WALIS U-series IDs 1017-

1022; WALIS RSL ID 551; FIL5/30,92.7m). The subsidence rate of Moruroa has been estimated at 0.007-0.008 mm/yr (Camoin et al., 2001), i.e. about 1 m since 125 ka.

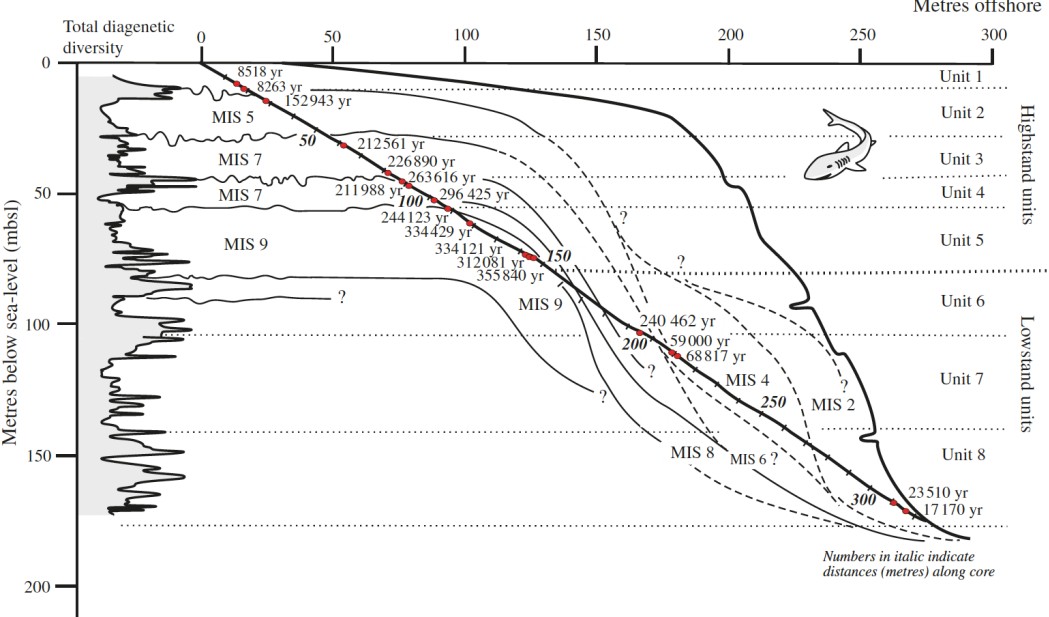

**Figure 13: Distribution of radiometric ages and reconstruction of successive reef units (MIS = Marine Isotopic Stages) in Moruroa cores FIL 8/30 and 8/40 (from Camoin et al., 2001 and Braithwaite and Camoin, 2011).**



Gischler et al. (2016) dated a barrier reef drill core from Bora Bora. Core TEV1 has been drilled at MSL and a *Pocillopora* specimen at $30.6 \pm 0.6$ m below MSL has given an age of $116.9 \pm 1.1$ ka (WALIS U-series ID 1869; TEV1). This sample is 100% aragonitic and the dating is considered as reliable. The subsidence rate of Bora Bora has been estimated at 0.05 to 0.14 mm/yr.

In a recent study by Montaggioni et al. (2019a) a 'eustatic sea-level' of about 7 m above MSL during MIS 5e has been considered on Takapoto, but no U-series age has been reported.

### 5.2 Cook Islands

LIG reef deposits have been reported from outcrops on several southern Cook Islands and from drill cores collected on Pukapuka, northern Cook Islands.

The southern Cook Islands are located in a complex tectonic setting related to volcanic processes since 1.65 Ma, the mean age for main shield-building phase (Te Manga) on Rarotonga (Thompson et al. 1998; Neal and Trewick, 2008). The formation of this volcanic system led to the subsidence of some islands (Rarotonga, Aitutaki, and Manuae) and the subsequent uplift of other islands related to a flexural response to loading of the ocean floor (McNutt and Menard, 1978). However, this flexural response has been the subject of many debates and

controversies regarding its course, timing and amplitude (see review in Woodroffe et al., 1991).

Gray et al. (1992) reported an average subsidence rate of 0.03 to 0.06 mm/yr for the Cook Islands, resulting in a subsidence of up to 7.5 m since 125 ka. In contrast, uplift rates have been estimated at approximately 0.044 mm/yr on Mangaia, 0.011 mm/yr on Atiu, and 0.005-0.007 mm/yr on Mitiaro and Mauke (Spencer et al., 1988). Uplift rates over the last 120 ka have been estimated at 0.07-0.10 mm/yr, 0.05 mm/yr, 0.03 mm/yr and 0.03-0.05 mm/yr

for Mangaia, Atiu, Mitiaro and Mauke, respectively, based on a 6-m sea-level highstand during the LIG (Woodroffe et al., 1991). These rates would imply an uplift of these islands of about 3.6 to 12 m since 120 ka. Conversely, a subsidence rate of 0.02 mm/yr has been considered for Rarotonga (Woodroffe et al., 1991). However, it is uncertain if the assumed 6-m LIG sea-level highstand for the Cook Islands is appropriate.

Gray et al. (1992) reported a subsidence rate of $0.045 \pm 0.028$ mm/yr for Pukapuka in the northern Cook Islands.

Exposed reef terraces can be found at different elevations, up to 15 m above MSL, on different southern Cook Islands, including Mangaia, Rarotonga, Atiu, Mauke and Mitiaro. The first datings of LIG reef limestones in the southern the Cook Islands have been provided by Veeh (1966). He dated *in situ* fossil corals from emerged Pleistocene reef terraces on Mangaia using the alpha-counting technique. Two corals, *Favia* (now called *Dipsastraea*; Huang et al., 2014) and *Leptoria*, at a hand-leveled elevation of $+2 \pm 0.5$ m MLWS have been dated





at 90 ± 20 ka and 110 ± 20 ka, respectively (WALIS U-series IDs 1744 and 1746; MANG1 and MANG2; WALIS
       RSL ID 585). Modern *Leptoria* are living at median and average depths of 4 and 7.5 m, respectively; however,
       their depth range is large, from the surface down to more than 80 m (see Table 2). Spencer et al. (1988) dated four
       *Porites* samples from an encrusting groove wall on Mangaia using the alpha-counting technique. All samples
       contained less than 1% of calcite and all datings have been accepted. The four samples at elevations of +15, +11,
+3 and +13 m MLT have been dated at 118 ± 12 ka, 107 ± 18 ka, 101 ± 12 ka and 135 ± 15 ka, respectively
       (WALIS U-series IDs 1848 and 1850-1852; M2, M3, M47 and M54; 1-sigma uncertainty; WALIS RSL ID 584;
       see Fig. 14). The tidal range is approximately 80 cm, i.e. the sample elevations are of +15.4, +11.4, +3.4 and +13.4
       MSL. *Porites* is living at median and average depths of 0 and 18.5 m, respectively in modern environments;
       however, this genus displays a wide depth range, from the surface down to more than 100 m (see Table 2).


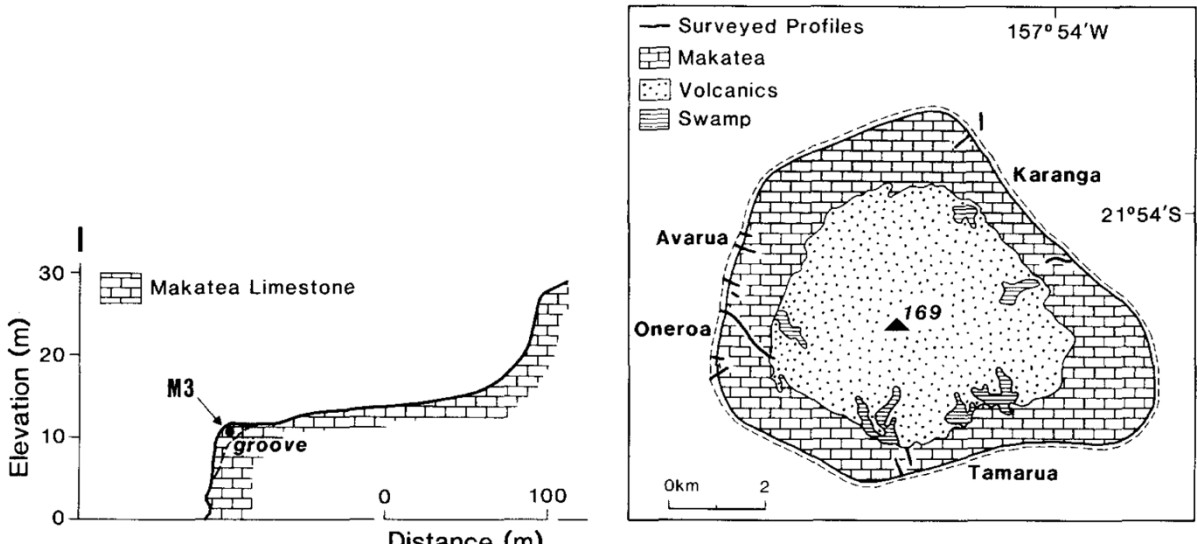

**Figure 14: Sea-level indicators on Mangaia, Cook Islands. Selected profiles and dated samples are shown (from
Woodroffe et al., 1991).**

       In 1985, Stoddart et al. presumed the occurrence of LIG reefs on Rarotonga, southern Cook Islands, at a maximum
elevation of 3.5 m above MSL. They assumed that these reefs have not been tectonically uplifted.
       Spencer et al. (1987, 1989) identified deposits of probable Late Pleistocene age at +12.2 m on Atiu, +10.0 m on
       Mauke and +9.8 m on Mitiaro in the southern Cook Islands. Stoddart et al. (1990) found evidence of late
       Pleistocene RSL up to +12.2 m on Atiu, +10.0 m on Mauke, and +9.8 m on Mitiaro. They also reported

unconformities within the Pleistocene at +1.75 to +2.65 m on Atiu, +1.5 to +2.75 m on Mauke and +4.1 to +6.0 m

on Mitiaro. Woodroffe et al. (1991) identified MIS 5e reefs on Atiu, Mauke and Mitiaro using the alpha-counting

technique. Three samples from Atiu have been dated (WALIS U-series IDs 1812-1817; AT8, AT9 and AT11; see

Fig. 15): 1) an *in situ Porites* on an erosional bench at an elevation of less than +2.7 m MLT (interpreted elevation

+1.5 ± 1.5 m MSL) with an age of 132 ± 10 and 111 ± 7 ka (WALIS U-series IDs 1812 and 1813; AT8; WALIS

RSL ID 557); 2) a *Porites* colony from the encrusting upper unit at an elevation of more than +2.7 m MLT

(interpreted elevation +6.5 ± 3.5 m MSL) with an age of 135 ± 10 and 151 ± 12 ka (WALIS U-series IDs 1814

and 1815; AT9; WALIS RSL ID 558); and 3) a massive *in situ* coral on the cliff top at an elevation of +10 to +12

m MLT (interpreted elevation +11 ± 1 m MSL) with an age of 111 ± 6 and 113 ± 9 ka (WALIS U-series IDs 1816

and 1817; AT11; WALIS RSL ID 559).

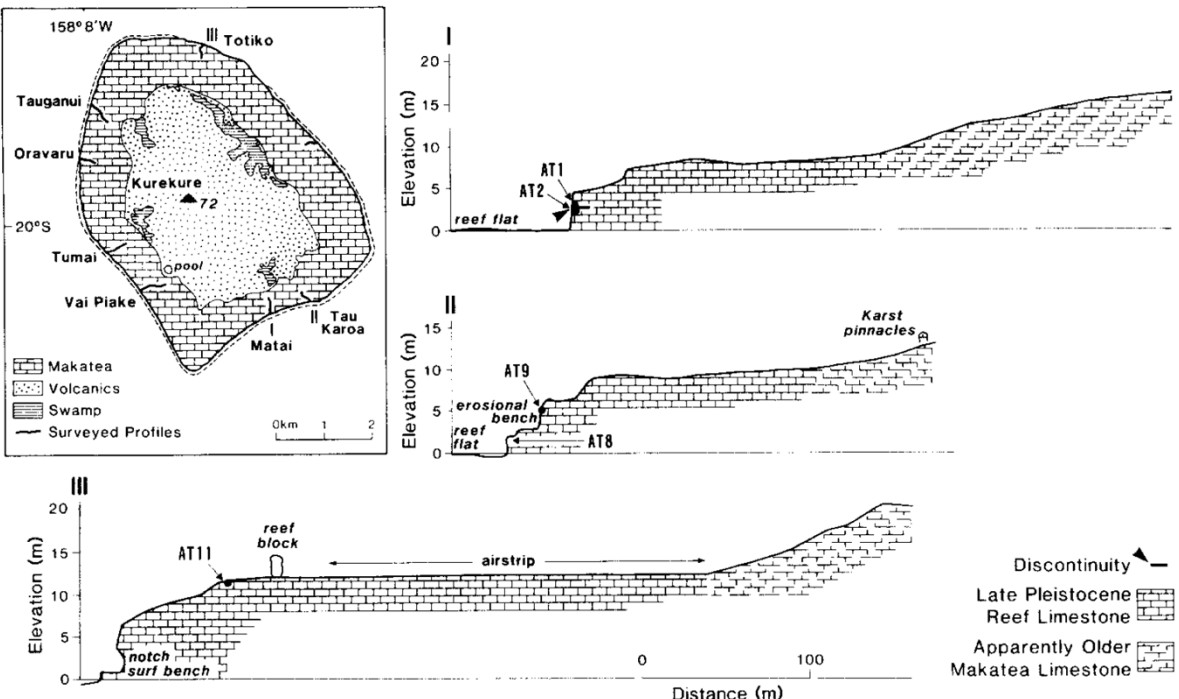


**Figure 15: Sea-level indicators on Atiu, Cook Islands. Selected profiles and dated samples are shown (from Woodroffe et al., 1991).**

Woodroffe et al. (1991) also dated four samples from Mauke (see Fig. 16): 1) a *Porites colony* from a perched reef

block at an elevation of +7 to +8 m MLT (interpreted elevation +7.5 ± 1 m MSL) with an age of 136 ± 10 and 138



± 10 ka (WALIS U-series IDs 1818 and 1819; MK9; WALIS RSL ID 560); 2) an encrusting coral from the upper

unit at an elevation of more than +2 m MLT (interpreted elevation +3 ± 1 m MSL) with an age of 123 ± 8 and 138

± 12 ka (WALIS U-series IDs 1820 and 1821; MK15; WALIS RSL ID 561); 3) a *Porites* colony from the

encrusting upper unit at an elevation of +5 m MLT (interpreted elevation +5 ± 1 m MSL) with an age of 136 ± 11

and 145 ± 10 ka (WALIS U-series IDs 1822 and 1823; MK16; WALIS RSL ID 561); and 4) a *Porites* boulder in

the upper unit at an elevation of +6 m MLT (interpreted elevation +6 ± 1 m MSL) with an age of 119 ± 8 and 119

± 7 ka (WALIS U-series IDs 1824 and 1825; MK18; WALIS RSL ID 561). *Porites* is living at median and average

depths of 0 and 18.5 m, respectively in modern environments; however, this genus displays a wide depth range

(see Table 2).

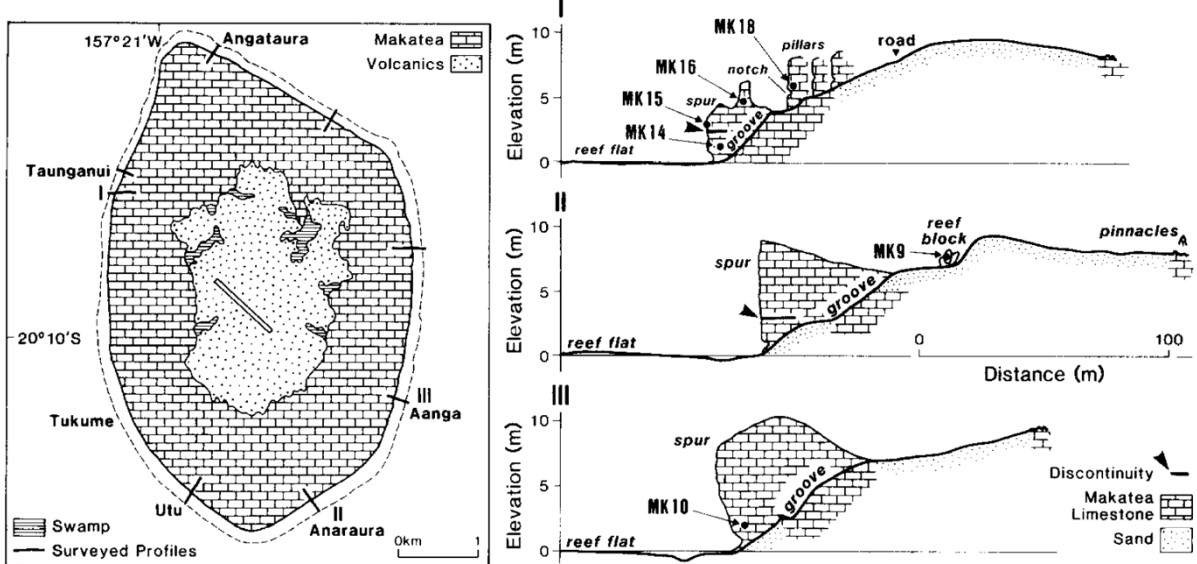

**Figure 16: Sea-level indicators on Mauke, Cook Islands. Selected profiles and dated samples are shown (from Woodroffe**
       **et al., 1991).**

Five samples from Mitiaro (see Fig. 17) have been dated by Woodroffe et al. (1991): 1) two specimens of Faviidae

(now called Merulinidae; Huang et al., 2014) from Te Unu at an elevation of more than +4 m (interpreted elevation

+6.5 ± 2.5 m MSL) have been dated at 128 ± 6 ka, 140 ± 10 ka, 98.1 ± 5.4 ka, 112 ± 7 ka and 132 ± 9 ka (WALIS

U-series IDs 1826-1830; MT1 and MT2; WALIS RSL ID 580); Merulinidae mainly indicate a depth range of 0-

30 m; 2) one *Plesiastrea* specimen from the north coast at an elevation of more than +4 m (interpreted elevation

+6.5 ± 2.5 m MSL) has been dated at 120 ± 6 ka and 132 ± 9 ka (WALIS U-series IDs 1832 and 1834; MT9;

WALIS RSL ID 581); in modern reef environments, *Plesiastrea* is mainly living at depths ranging from the surface down to 10 m; 3) one *Porites* specimen from the village at an elevation of more than +2 m (interpreted elevation

+5.5 ± 3.5 m MSL) has yielded ages of 116 ± 8 ka, 144 ± 9 ka and 106.3 ± 5.4 ka (WALIS U-series IDs 1836-1838; MT13; WALIS RSL ID 582); and 4) one *Porites* specimen from Vaikoua at an elevation of more than +3 m (interpreted elevation +4 ± 1 m MSL revealed an age of 118 ± 7 (WALIS U-series IDs 1841 and 1842; MT15; WALIS RSL ID 583). *Porites* is living at median and average depths of 0 and 18.5 m, respectively in modern environments; however, this genus displays a wide depth range (see Table 2). All samples contained less than 1-

2% of calcite and all datings have been accepted (1-sigma uncertainty). Elevations have been hand-leveled and the tidal range is approximately 80 cm.

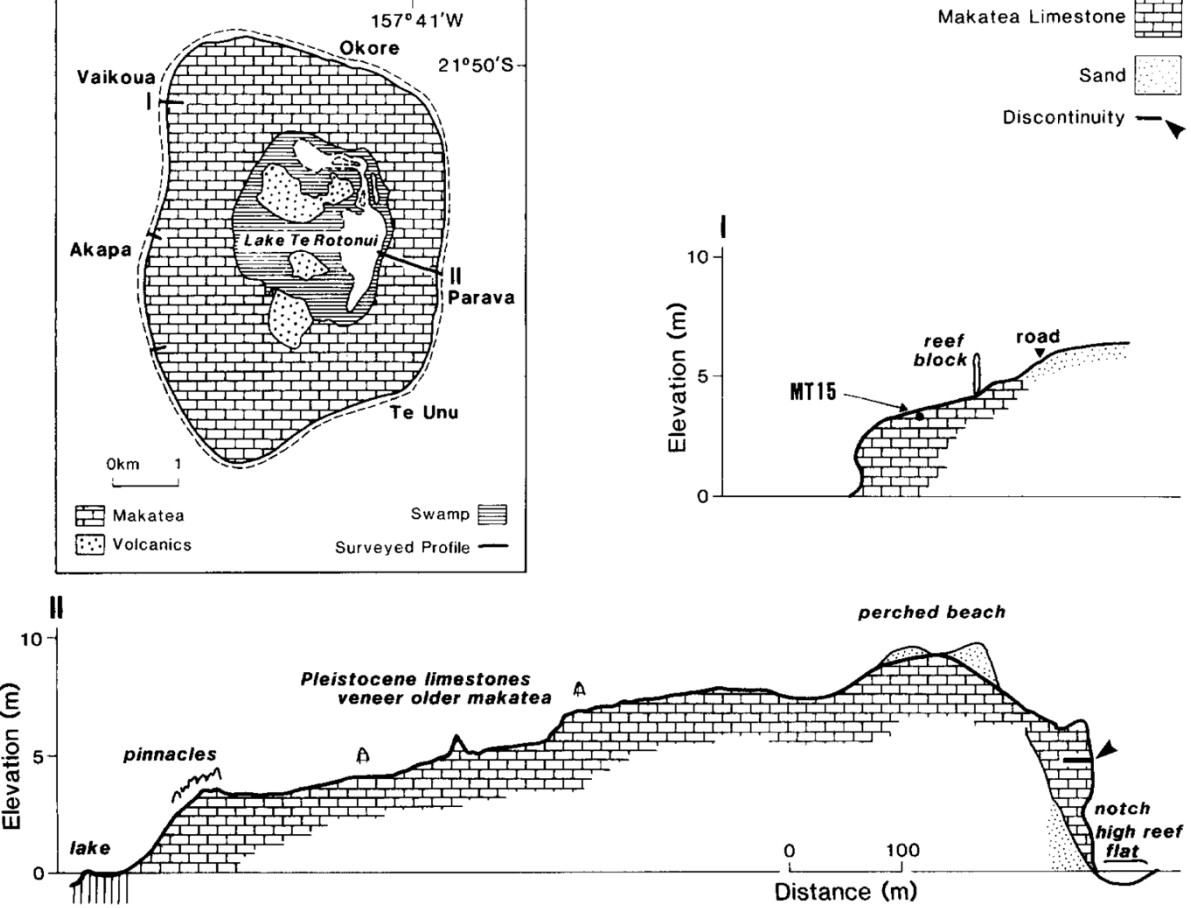

**Figure 17: Sea-level indicators on Mitiaro, Cook Islands. Selected profiles and dated samples are shown (from Woodroffe et al., 1991).**



Gray et al. (1992) extracted drill cores and described widespread LIG reefs on Pukapuka (see Fig. 3 in Gray et al.,
1992), however, they only obtained reliable ages on two *Porites* specimens (100% aragonite; WALIS RSL ID
586). Electron spin resonance (ESR) dating provided two ages of $144 \pm 22$ and $136 \pm 20$ ka for samples at depths
of 21.77 and 24.85 m MSL, respectively (WALIS ESR IDs 104 and 105; 991 and 951). The *Porites* specimen at
a depth of 24.85 m MSL was also dated using the alpha-counting technique, which revealed an age of $130 \pm 9$ ka

(WALIS U-series ID 1855, 1-sigma uncertainty; 951). *Porites* is living at median and average depths of 0 and 18.5
m, respectively in modern environments.

    Most of the dated samples from the southern Cook Islands by Woodroffe et al. (1991) can be related to the LIG
RSL highstand, but a MIS 7 reef, i.e. the Penultimate Interglacial, has been also observed (see Fig. 18). Woodroffe
et al. (1991) did not observe two periods of high RSL separated by a period of lower RSL during the LIG. However,

they describe several stratigraphic features, such as erosional benches and notches, that might indicate RSL
fluctuations during MIS 5e.

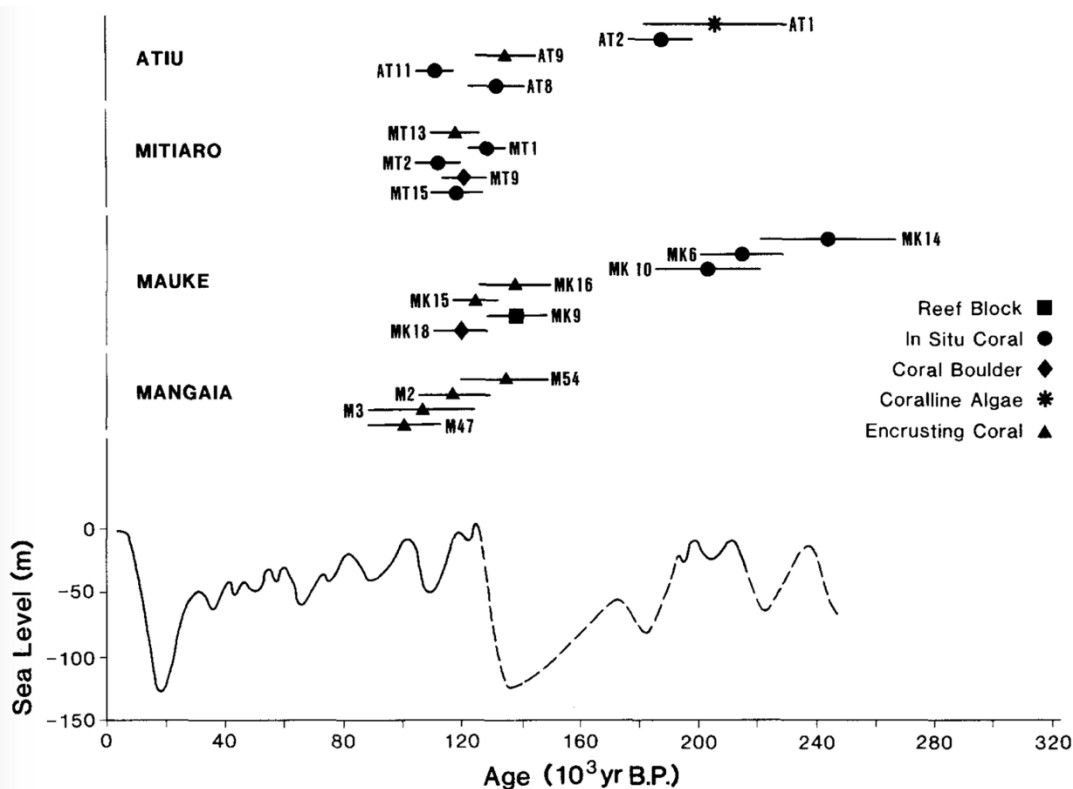

**Figure 18: Uranium-series ages (± 1 standard error) from corals of the southern Cook Islands, and comparison with
the late Quaternary sea-level curve of Chappell and Shackleton (1986) (from Woodroffe et al., 1991).**




Overall, the large variability in elevations and the lack of accurate ages from the Cook Islands does not allow any precise conclusion about the LIG RSL.

### 5.3 Samoa

Exposed Pleistocene reef terraces have been described on several Samoan Islands, but no dates have been reported
in the reviewed studies. Kear and Wood (1959) reported terraces occurring at up to 15 m and 10 m higher than present RSL on Upolu and Upolu's offshore islands, respectively and that might correspond to the LIG period (125 ka). They also described benches near Fagaloa Bay at elevations of +39 to +60 m, which might relate to the LIG or Penultimate Interglacial.

Stice and McCoy (1968) observed cliffs with benches at elevations of 4.5 to 1.5 m above MSL, probably indicating
former higher RSL on three Manu'a Islands. The +3.7 to +3.0 m terrace is composed of sand and coral shingle or entirely of sand.

### 5.4 Niue

LIG reef deposits have been dated from outcrops by Kennedy et al. (2012). The earliest studies by Agassiz (1903), David (1904) and Schofield (1959) described different raised reef terraces on Niue. The widest terrace (Alofi
Terrace; see Figs 19 and 20) is the most continuous one around the island (Agassiz, 1903). David (1904) recognized three terraces at +6 m, +24 to +27 m and +60 m and mentioned also a terrace at an elevation of 40 m above MSL. In 1959, Schofield distinguished the Alofi Terrace at about +23 m and the Mutalau Platform, a +69 m-high terrace, and identified five more terraces, including two submerged ones at -37 to -33 m and -15 to -11 m, and three exposed ones at +12 to +14 m, +35 to +40 m and at +55 m. At least parts of the Alofi Terrace date to
230-260 ka (Wheeler, 2000). The Alofi Terrace is overlain by a 6 m-thick LIG reef sequence (Paulay, 1988; Paulay and Spencer, 1992; Spencer and Paulay, 1994; Wheeler, 2000). Beach deposits and erosional features indicate a 6 m-high RSL (Paulay and Spencer, 1992). More recent studies describe the different terraces on Niue from close to present MSL up to an elevation of +60 m (Terry and Nunn, 2003; Nunn and Britton, 2004; Kennedy et al., 2012). Nunn and Britton (2004) indicated the presence of at least seven terraces, at +58, +52, +43.5, +34, +28,
+23 and +18 m with ages back to 700 ka. Kennedy et al. (2012) dated the Alofi Terrace (see Fig. 20) based on the dating of one *Porites* specimen at 133 ± 0.9 ka (WALIS U-series ID 1285; NT11) at an elevation of 3.1 m above the height of living corals. This coral head consisted of more than 95% of aragonite and the dating is considered as reliable. *Porites* is living at median and average depths of 0 and 18.5 m, respectively in modern environments;



however, this genus displays a wide depth range (see Table 2). Niue is affected by uplift with estimated rates of
0.13–0.16 mm/yr, with uplift considered to be ongoing (Dickinson, 2001), thus indicating an uplift of 16 to 20 m
since 125 ka.

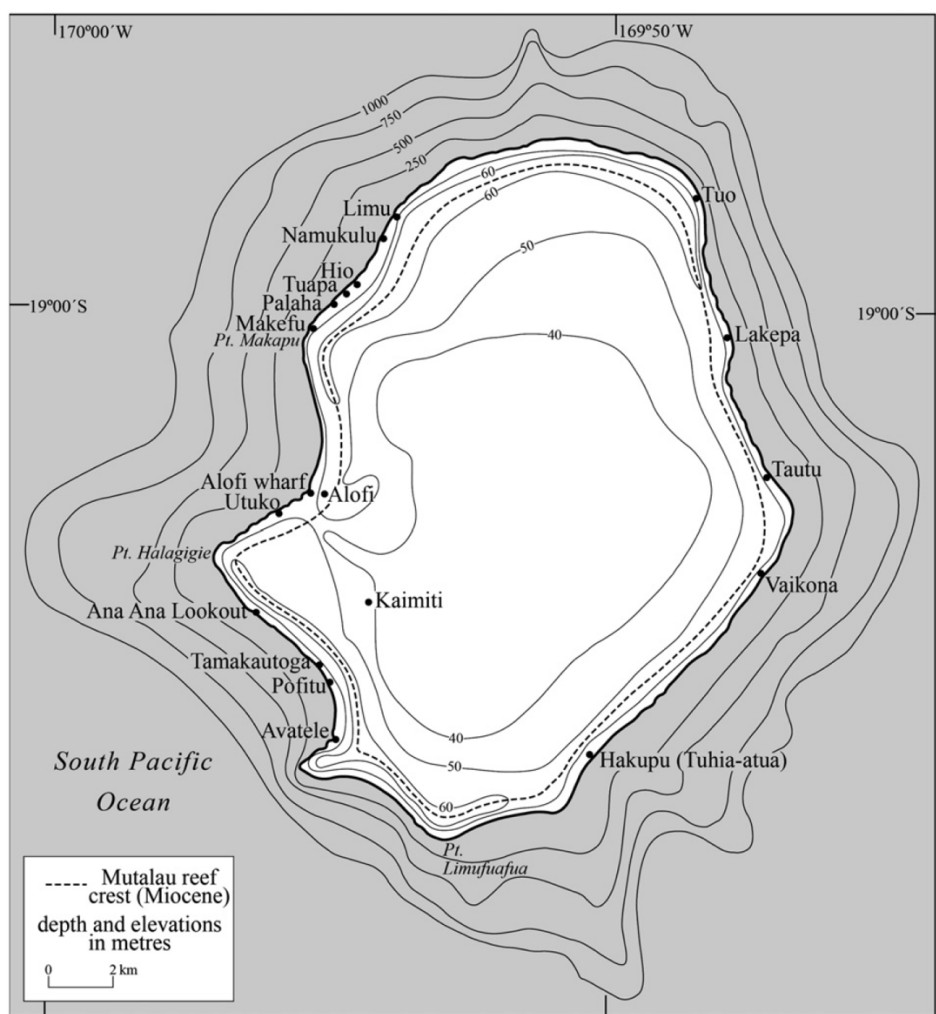

**Figure 19: Bathymetric and topographic map of Niue (from Kennedy et al., 2012).**


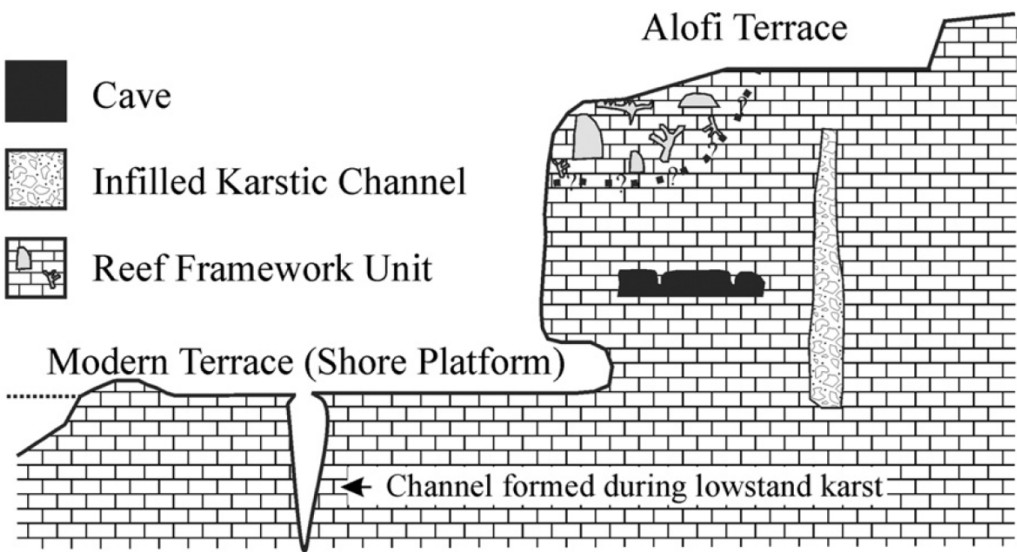

**Figure 20: A schematic cross-section showing the stratigraphic relationships between the Pleistocene and modern terrace on Niue. Reef framework is not ubiquitous along the Alofi Terrace edge (from Kennedy et al., 2012).**

**5.5 Tonga**

LIG reef terraces in Tonga have been reported from outcrops at elevations of about +5 to +7 m. Taylor (1978) and Yonekura (1983) dated emerged reef terraces on the Tongatapu block using the alpha-counting technique. Taylor (1978) described a group of emerged notches and caves at 6.6-7.2 m above present MSL and dated a single sample at 133 ± 12 ka (WALIS U-series ID 1787). Yonekura (1983) described LIG reefs at +5.5 m MHT and provided ages of 135 ± 15 ka and 133 ± 12 ka (WALIS U-series IDs 1785 and 1786; WALIS RSL ID 554). A wide palaeo-water depth range of 0-30 m has been assumed as no taxa identification has been reported. The inaccurate dating of LIG reefs in Tonga does not allow any precise conclusion regarding the LIG RSL. Uplift rate is negligible in the Tonga arc (Yonekura, 1983).

**5.6 Fiji**

The Fiji group of islands has a total land area of 18,270 km². Its geological history is complex owing to its proximity to the Australian–Pacific plate boundary. Several Fiji Islands (e.g. Kadavu, Kaibu) exhibit a series of coral reef terraces arranged around a volcanic core, the lowest one being attributed to the MIS 5e (Nunn and





Omura, 1999; Nunn et al., 2002). LIG deposits from Fiji were first dated by Taylor (1978) who reported ages of

126 ± 7 ka and 136 ± 12 ka (WALIS U-series IDs 1866 and 1867; Yasawas1 and Yasawas2) on two unidentified

corals from a terrace at +2 to +3 m on Yasawa Island (WALIS RSL ID 512). However, the lack of any coral

identification does not allow to precisely consider the sea-level position at that time. Nunn and Omura (1999) have

reported a similar age of 128.7 ± 1.6 ka from a coral sampled from the same reef terrace (WALIS U-series ID

1868; Yasawas3). In addition, Nunn and Omura (1999) reported on Kadavu Island morphological features that

they have attributed to two RSL stands during the LIG without providing robust chronological constraints (see

Fig. 4 in Nunn and Omura, 1999). In addition, an erosional bench reported at +5 m has been interpreted as an early

LIG sea-level maximum at about 136 ka, while a distinctive level at +2 m would correspond to a second LIG

maximum at about 120 ka. Nunn and Omura (1999) have assumed that the LIG reefs are located below present

MSL in the western part of Kadavu Island.

Nunn et al. (2002) and Nunn and Omura (2003) described LIG reef terraces from Kaibu Island (see Fig. 5 in Nunn

et al., 2002; Figs 2 and 3 in Nunn and Omura, 2003), and reported three U-series ages. A *Porites* sampled at 1.45

m above the modern reef (WALIS RSL ID 511) yielded an age of 126.8 ± 2.1 ka (WALIS U-series ID 1865;

sample AO455). Two *Platygyra*, including *Platygyra lamellina* sampled between 3.6 and 3.85 m above the modern

reef (WALIS RSL IDs 509 and 510) gave ages of 132.8 ± 2.3 ka (WALIS U-series ID 1864; AO454), and 131.1

± 2.4 ka (WALIS U-series ID 1863; AO443), respectively. In modern environments, *Platygyra lamellina* typifies

median and average depths of 4 and 4.4 m, respectively (Table 2; WALIS RSL IDs 509 and 510). Nunn et al.

(2002) interpreted the age distribution as reflecting a double sea-level maximum with peaks around 133-130 ka

and 123-120 ka, following the earlier interpretations by Nunn and Omura (1999). The earlier maximum has been

reported about 2 m lower than the later and marked by the growth of a surface reef, the later maximum appearing

to have involved only cutting of erosional shorelines at the +5.0 to +5.5 m level (Nunn and Omura, 2003). Nunn

et al. (2002) and Nunn and Omura (2003) compared levels of emerged shorelines on northeastern Fiji Islands and

show that a LIG shoreline has emerged in northeast Fiji by around +5.0 to +5.2 m (range +4.8 to +5.5 m). In

addition, they compared these levels to those from the western Fiji Islands and concluded that LIG RSL in the

Fijian region ranged from 4.6 to 7.1 m above its present mean level.

**5.7 Tuvalu**

The only available chronological information concerning the potential occurrence of LIG on Tuvalu is based on

the dating of a drill core carried out on Funafuti Atoll by the Coral Reef Expedition of The Royal Society (1896–

1898). It was obtained on an unidentified coral collected at 27.4 m (sample 274) below the reef surface and yielding

a Sr isotope age of 0.14 ± 0.15 Ma (WALIS U-series ID 1734; Ohde et al., 2002). The carbonate sequence
attributed to the LIG is reported from at least 24.4 to 27 m core depth and is limited by unconformities (Ohde et al., 2002).

### 5.8 Kiribati

LIG reefs have been reported from outcrops on Christmas Island and from a drill core collected on Tarawa. Marshall and Jacobson (1985) dated a drill core from an islet on Tarawa using the alpha-counting technique. The
coral at a depth of 17-17.8 m below MSL revealed an age of 125 ± 9 ka (WALIS U-series ID 1811; 81390010; 1-sigma uncertainty). The sample was composed of 100% aragonite and the dating is considered as reliable. Woodroffe and McLean (1998) dated an emerged reef terrace on Christmas Island and obtained a more accurate age using a mass spectrometer. The coral sample at an elevation of +1 m MSL has been dated at 130 ± 5 ka (WALIS U-series ID 1810; 16a). A water depth range of 0-30 m has been assumed as no taxa identification has
been reported. Christmas Island has remained stable throughout the Late Quaternary (Woodroffe and McLean, 1998).

### 5.9 Solomon Islands

The geomorphology of the Solomon Islands has been studied by Stearns (1945), Grover (1958) and Stoddart (1969) describing elevated terraces in different island groups. Grover (1958) described notches on Mone Island at an
elevation of +0.76, +3 and +5.5 m on an 8.5 m-high cliff and at +22 m on a 16.8 to 29.6 m-high cliff. Stoddart (1969) mentioned the occurrence of Pleistocene reef limestones and reported terraces of clastic debris with reef material at elevations of 145 to 135, 80, 60, 44 and 27.5 m above present MSL. He also reported tidal notches on Banika Island with an elevation of 1.5 m to 5 m above present MSL. A terrace on Banika island at 9 m above present MSL was older than 39,700 years. However, it was impossible to relate the relevant benches to specific
Pleistocene RSL changes. The Solomon Islands are affected by uplift. There are 16 arc segments in the Solomon Islands comprising three major tectonic regimes (Chen et al., 2011).



### 5.10 Vanuatu

Uplifted Quaternary reef terraces cover wide areas of Vanuatu islands, specifically Santo, Malakula and Efate at elevations ranging from 600 m on Efate and the North of Malakula, to 860 m in the South of Malakula and up to more than 1000 m in the western part of Santo (see Fig. IV-2 in Jouannic et al., 1982). They form the external part of a succession of terraces overhanging a Holocene coastal reef platform. Since the first geological and morphological investigations in the late 1960s and the 1970s (Mitchell, 1966, 1969, 1971; Robinson, 1969; Mallick and Greenbaum, 1977; Ash et al., 1978), these uplifted terraces have been the topic of studies dealing both with their chronology and the reconstruction of vertical movements in the New Hebrides forearc.

The first chronological study has been carried out by Neef and Veeh (1977) who published the first four U-series ages obtained on corals from MIS 5 raised reefs from Efate and Malo, which occur at +100 m and +96 m, respectively. These raised reefs have been interpreted as coeval to RSL stands VIIa (140,000 yr) and VIIb (120,000 yr) from Huon Peninsula (Chappell, 1974; Bloom et al., 1974), respectively. The two ages obtained by Neef and Veeh (1977) were of 134 ± 10 ka (WALIS U-series ID 1847; Efate 330) and 134 ± 6 ka (WALIS U-series ID; Ma 1) were judged to be more reliable than those which yielded ages of about 120 ka (Efate X and Efate 235) on the basis of mineralogical criteria used in the evaluation of samples (Neef and Veeh, 1977). However, corals that have been dated were not identified, thus hampering the accurate reconstruction of palaeo-water depth during reef growth. Subsequent studies by Bloom et al. (1978), Jouannic et al. (1980, 1982) and Lecolle and Bernat (1985) have been focused on the uplifted terraces from Efate. Two alpha spectrometry ages of 141 ± 8 ka (WALIS U-series IDs 1952 and 1953; E-L-3 and E-T-2) were obtained by Bloom et al. (1978) on unidentified corals from the coral terrace located at +110 to +130 m in Port Havannah area (WALIS RSL IDs 491 and 492). Additional U/Th ages of 124 ± 7 ka (WALIS U-series ID 1950; sample I-7-5), 131 ± 11 ka (WALIS U-series ID 1951; E-L-1), 121.6 ± 7.3 ka (WALIS U-series ID 1942; EKB4-2) and 125.3 ± 7.5 ka (WALIS U-series ID 1941; EKB4) were obtained by Jouannic et al. (1982) and Lecolle and Bernat (1985) on the same terrace (WALIS RSL IDs 491 and 492). This terrace has been therefore attributed to MIS 5e and its duplication in the area of Port Havannah, similar to the one observed by Bloom et al. (1974) on Huon Peninsula (VIIa and VIIb terraces; Bloom et al., 1974), has been interpreted as resulting from low amplitude sea-level fluctuations during MIS 5e (Jouannic et al., 1982). Two ages of 130 ± 7 ka (WALIS U-series ID 1948; sample E-X-2) and 114 ± 6 ka (WALIS U-series ID 1949; sample E-X-4) have been obtained on the coral reef terrace at +85 to +95 m in the same area (Jouannic et al., 1982). The MIS 5e terrace has been dated by Lecolle and Bernat (1985) at different altitudes in several other areas on Efate Island (see Fig. 2 in Lecolle and Bernat, 1985). An age of 121.4 ± 7.3 ka (WALIS U-series ID 1937; sample ESA3)

was reported at +50 m in Siviri area (WALIS RSL IDs 706 through 708), while ages of 128.0 ± 7.7 ka (WALIS
        U-series ID 1938; ERB1) and 133.5 ± 8 ka (WALIS U-series ID 1939; sample EWA03) were obtained on the
        terrace occurring at +80 m in Malafao area (WALIS RSL ID 709), and ages of 117.8 ± 7.0 ka (WALIS U-series
        ID 1945; ETD1) and 123.6 ± 7.4 ka (WALIS U-series ID 1946; ETA4) on the terraces at +80 m and +100 m,
        respectively, in Tukutuk area (WALIS RSL ID 713). However, the palaeo-MSL cannot be further constrained due
to the lack of coral identification. Chronological data obtained on Efate Island, combined with U-series ages
        obtained on Malakula and Santo islands, have been used to reconstruct vertical tectonic movements in the New
        Hebrides Island Arc (Jouannic et al., 1982; Lecolle and Bernat, 1985). The chronological frame of the MIS 5e
        terrace from Efate has been reassessed by Edwards et al. (1987a, b) who identified and dated the two corals
        previously reported by Bloom et al. (1978) at 129.9 ± 1.1 ka and 129.2 ± 1.1 ka for *Oulophyllia crispa* (WALIS
U-series IDs 1435 and 1436; E-T-2-A and E-T-2) and 125.5 ± 1.3 ka for *Porites lutea* (WALIS U-series ID 1854;
        E-L-3). The association of those two coral species indicates probable palaeo-water depths of less than 10 m
        (WALIS RSL ID 492).

        The reef terrace corresponding to MIS 5e (from 119 to 132 ka) was not clearly identified on Malakula Island
        (Jouannic et al., 1982; Cabioch and Ayliffe, 2001; see Fig. 4 in Taylor et al., 1980). Jouannic et al. (1982) also
suggested that some of the discrepancies in terrace altitudes in previous work may have been due to problems
        associated with coral reworking and errors in altitude assignment. An age of 119.8 ± 8.0 ka (WALIS U-series ID
        1947; NH14) has been obtained on the coral reef terrace located at +180 m in Npénanavet/Tenmaru area by
        Jouannic et al. (1982). A similar age of 121.3 ± 1.2 ka (WALIS U-series ID 1846; Espiegle1-188m) has been
        obtained at +188 m by Cabioch and Ayliffe (2001) on an unidentified coral (WALIS RSL ID 490; see Fig. 21).
However, Cabioch and Ayliffe (2001) have considered that the MIS 5e terrace should correspond to the wide
        plateau culminating at +230 m and that, based on field evidence, its composite reef structure may result from minor
        sea-level variations during the MIS 5e RSL stand.

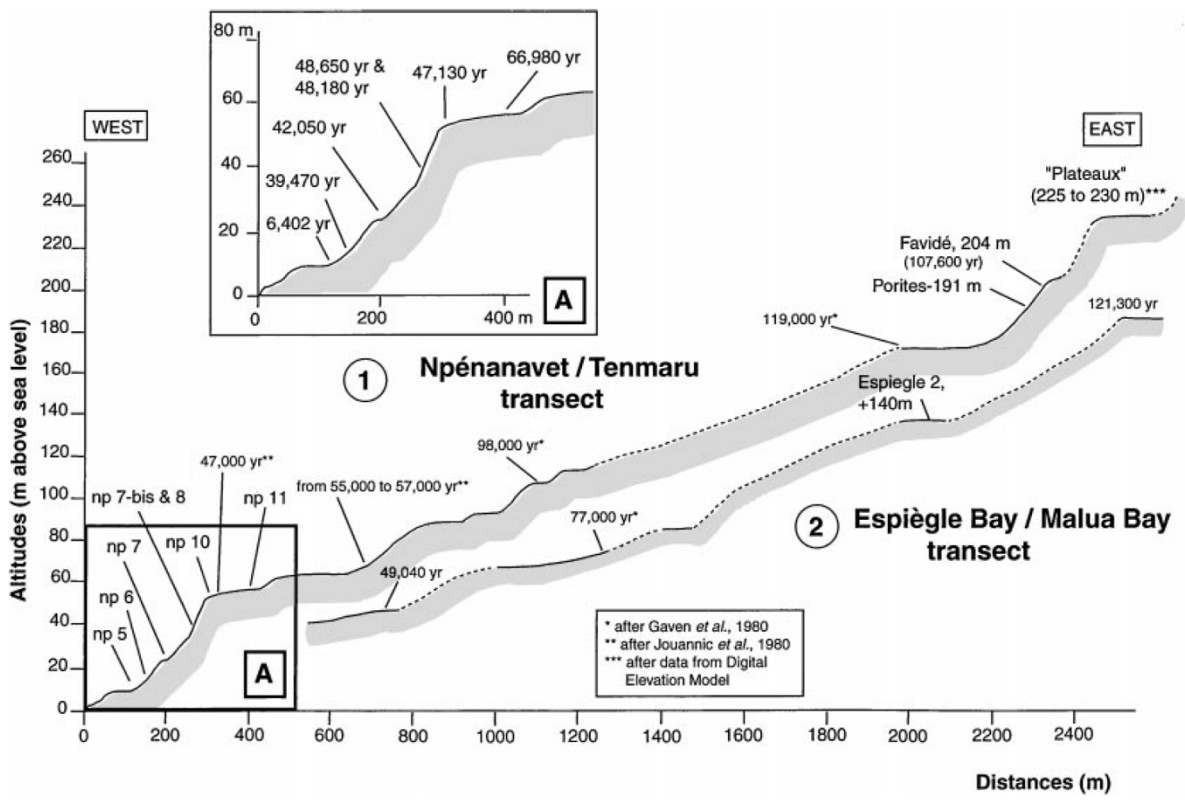

**Figure 21: Profile showing the distribution of reef terraces on Malakula. Transect 1: Npénanavet/Tenmaru transect;**
**transect 2: Espiègle Bay/Malua Bay transect (from Cabioch and Ayliffe, 2001).**

Taylor et al. (1985) have investigated raised coral reef terraces from the Torres Islands to reconstruct rapid vertical
movements in the New Hebrides forearc. They interpreted the very wide terrace at about +100 m on Toga (WALIS
RSL ID 488; see Fig. 22) as formed during the MIS 5e and deduced uplift rates of about 1 mm/yr for at least the

past 125,000 years, i.e. 125 m since 125 ka. Ages assigned to MIS 5e have been reported from the reef terrace
located at +80 m on Loh Island (WALIS RSL ID 487; see Fig. 23) where a *Porites* and a *Leptoria phrygia* yielded
ages of 122 ± 7 ka (WALIS U-series ID 1839; LO-B-2) and 135 ± 11 ka (WALIS U-series ID 1840; LO-B-3),
respectively. In modern reefs, *Leptoria phrygia* typifies shallow-water environments with median and average
depths of 4 and 5 m, respectively; however, this species displays a rather large depth range, from the surface down

to 30 m or even 47 m (Table 2).



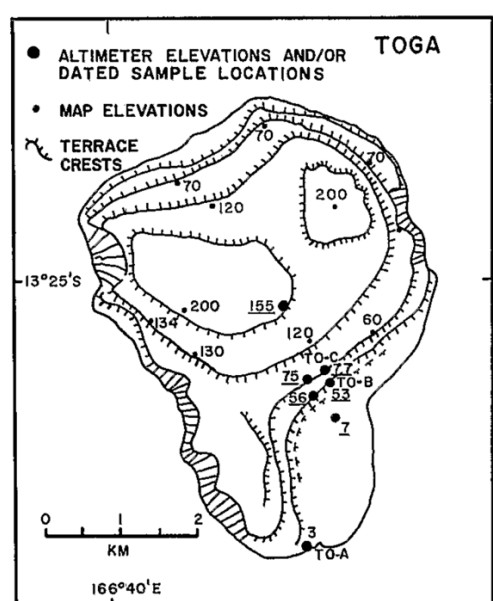

**Figure 22: Raised reef terraces on Toga Island (from Taylor et al., 1985).**

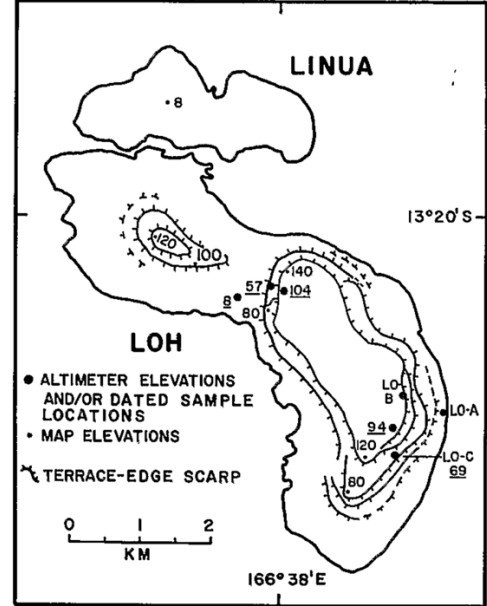

**Figure 23: Raised reef terraces on Loh and Linua (from Taylor et al., 1985).**






On Tanna Island, raised lagoonal deposits currently at +18 m (WALIS RSL ID 506; see Fig. 24) have been attributed to MIS 5e based on an age of 126.7 ± 1.5 ka (WALIS U-series ID 1861; Ta9) obtained on an *Acropora*, which does not give precise indications on the palaeo-water depth during reef growth.

**Figure 24: Raised reef terraces on Tanna Island (from Neef and Hendy, 1988 and Neef et al., 2003).**





### 5.11 New Caledonia

New Caledonia comprises the main island (Grande Terre), the Isle of Pines to the south, the Loyalty Islands to the east (Maré, Lifou, Tiga and Ouvéa), and the Belep Archipelago to the northwest.

LIG reef deposits have been reported both from outcrops in Loyalty Islands (Maré, Lifou and Ouvéa) and from
drill cores that have been carried out throughout the modern barrier and fringing reefs from the western coast of Grande Terre, as well as in the Chesterfield islands that are located about 500 km to the WNW of New Caledonia. Pioneering studies on Pleistocene reef terraces from the Loyalty Islands, including the LIG, have been carried out in the 1970s by Launay and Récy (1972) and Coudray (1976). The first radiometric ages on these raised reefs have been obtained by Bernat et al. (1976), Marshall and Launay (1978) and Gaven and Bourrouilh-Le Jan (1981), and
were used to reconstruct tectonic movements of these islands since Quaternary times.

$^{230}$Th/$^{238}$U ages reported by Bernat et al. (1976) on raised reefs occurring at +11 to +12 m elevation on Lifou and Maré range from 122 to 120 ka and would therefore indicate that this terrace corresponds to the MIS 5e. A coral from an elevation of +20 m on the Isle of Pines, which is located immediately to the southeast of New Caledonia, was dated at 126 ± 5 ka. However, this suite of corals did not satisfy the closed-system criteria because they were
partly recrystallized (Marshall and Launay, 1978). Marshall and Launay (1978) could not obtain reliable ages on a terrace located at +3.5 m on Lifou, but they have reported an age of 117 ± 6 ka on a *Platygyra* colony (WALIS U-series ID 1792; Ouvea3) sampled in the coral reef terrace occurring at +7.5 m on Ouvéa (WALIS RSL ID 737) and correlated this terrace with the 120 ka Waimanalo shoreline from Hawaii (Ku et al., 1974). The distribution of modern *Platygyra* is characterized by median and average depths of 4 and 7 m, respectively; however, their
depth range is very large and extends from the surface down to 80 m (Table 2). Marshall and Launay (1978) have concluded that there is evidence of LIG reef terraces at +11 to +12 m on Lifou, at +20 ± 3 m on the Isle of Pines (Bernat et al., 1976), and at +7.5 m on Ouvea but, because of diagenetic alteration of the Lifou and Isle of Pines coral samples, the exact age of these terraces could not be defined. Gaven and Bourrouilh-Le Jan (1981) have reported ages ranging from 100 to 295 ka on pristine corals sampled in terraces located between 0 and +9 m on
Maré. They have noted that some of these terraces have been karstified and attributed notches at +9 and +12 m to 125 and 205 ka RSL.

The study of drill cores from modern reefs from several areas of the west coast of New Caledonia (Amédée, Kendec and Ténia islets) have provided the opportunity to document reef units attributed to MIS 5e that occur at various depths, thus demonstrating differential tectonic movements along the west coast of New Caledonia (see
Fig. 1 in Cabioch et al., 2008).




Below fringing and barrier reefs, the 125 ka-old reef unit is generally capped by calcretes resulting from a relatively long period of emergence and is locally affected by freshwater alteration, even if the barrier reef is frequently unaltered. Several alpha counting U/Th dates have been obtained on these drill cores: 125 ± 20 ka in the Ténia drill cores (Coudray, 1976), 135 ± 12.8 ka in the Ilot Vert drill cores (Degaugue-Michalski, 1993) and 125 ± 1 ka
and 131 ± 1 ka at Ilot Amédée (in cores Amédée-1 and -2, respectively; Cabioch et al., 1996), where a framework of corals and coralline algae with a few bioclastic beds, typifies reef flat facies. In Amédée-4 drill core, the 125 ka-old reef occurs between 14 to 37 m core depth (WALIS RSL ID 474), and is predominantly composed of biocalcarenites and rare coral colonies (Cabioch et al., 1999). U/Th dating by TIMS carried out on coral samples extracted from Amédée site has demonstrated that the distribution of U and Th isotopes was affected by subaerial
diagenesis during emergence of the reef units, while the Holocene coral sequence seems largely unaffected (Frank et al., 2006; Cabioch et al., 2008). Various correction models were tested to get reliable ages from the late Pleistocene sequences and yielded ages ranging from 102.6 ± 4.4 to 127.5 ± 5.4 ka, with a majority of values around 122 ka (Frank et al. 2006; Cabioch et al., 2008). The reef unit occurring between 37 and 14 m core depth has been therefore attributed to MIS 5, especially to MIS 5e (WALIS U-series IDs 1266 through 1283; WALIS
RSL ID 474; see Fig. 25); based on these results, the local subsidence rate has been estimated to 0.16 mm /yr (Frank et al., 2006), thus a total of 20 m since 125 ka.

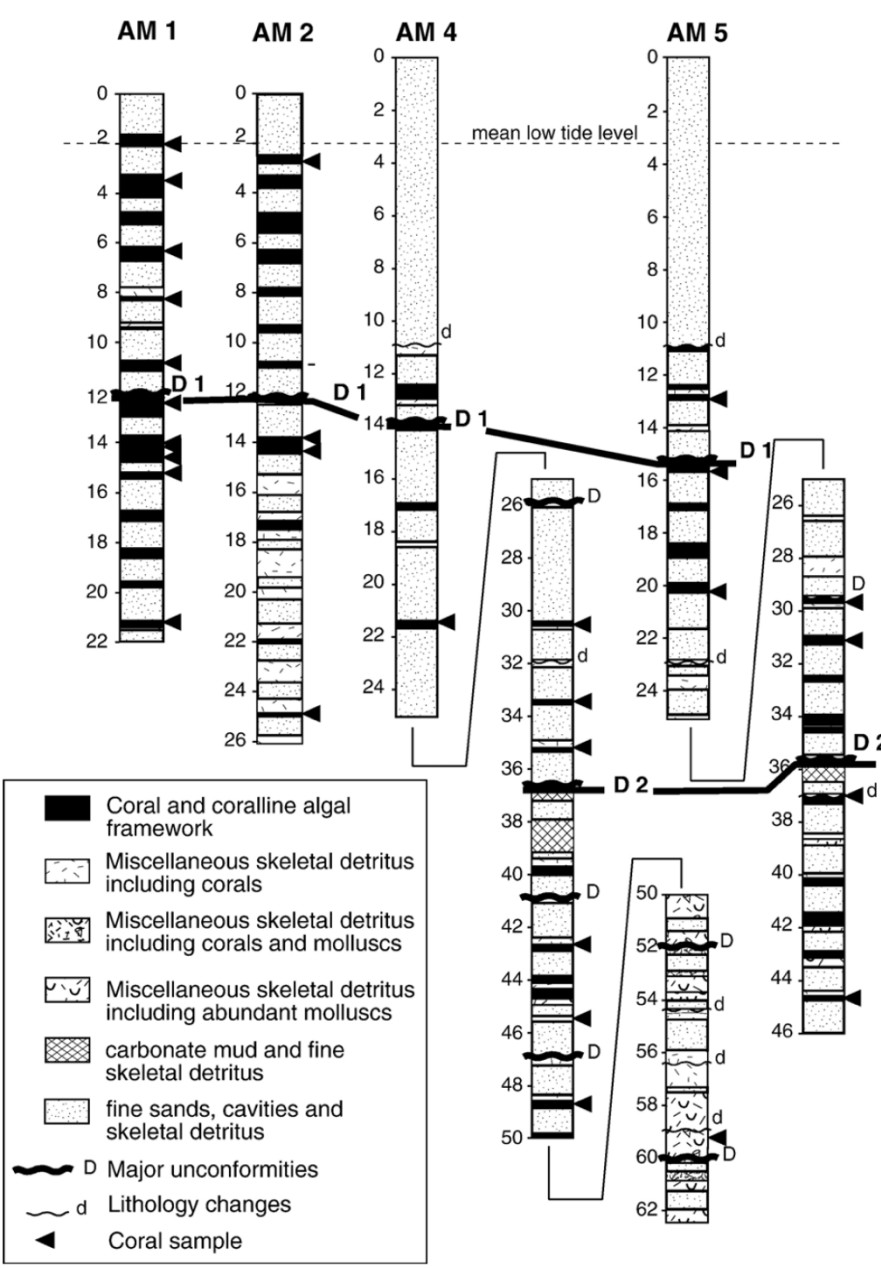

**Figure 25: Synthetic log of the four cores drilled at Amédée Islet. D1 is the Pleistocene/Holocene boundary unconformity; D2 is the bottom of the LIG terrace (from Frank et al., 2006).**






Similar Pleistocene reef units have been identified in drill cores retrieved from the Kendec (Cabioch et al., 2008) and Vert (Cabioch, 1988; Degaugue-Michalski, 1993) islets. Coral identifications (Hongo and Wirrmann, 2015) have shown that the MIS 5 reef units are dominated by important accumulations of shallow-water assemblages including arborescent colonies of *Acropora* (Kendec drill core), corymbose and tabular colonies of *Acropora*

(Kendec and Amédée-4 drill cores) and massive *Porites* colonies (Amédée-5 drill core) that characterize reef edge and upper reef slope environments exposed to strong wave action.

Below the barrier reef that occurs at Ténia islet, the 125 ka-old unit consists of a framework of corals, bryozoans and coralline algae trapped in bioclastic sediments rich in bryozoans, typifying reef flat facies (Coudray, 1976). The reef unit occurring between 14 and 16 m core depth in drill core extracted more recently in this area (Ténia-

3; Montaggioni et al., 2011) has yielded U-series ages of 127.9 ± 3.3 ka (WALIS U-series ID 1788; 14T3), 130.55 ± 3.75 ka (WALIS U-series ID 1789; 16T3) and 132.65 ± 4.7 ka (WALIS U-series ID 1790; 18T3), according to the U/Th dating procedure previously described by Frank et al. (2006).

**5.12 Papua New Guinea**

Quaternary coral reef terraces from Huon Peninsula (Papua New Guinea) are prominent morphological features

that extend more than 80 km along the coast and have been uplifted at a rate that varies from 0.5 mm/yr in the northwest to nearly 4 mm/yr to the southeast (Esat et al., 1999). They range in elevations from 0 to 200 m (see Figs 26 and 27) and consist of an upward succession of fringing and barrier reef types. Seven coral reef terraces, I through VII from the coast landwards, are spaced at about 20-kyr interval (see Figs 26 and 27) and provide a rather complete record of RSL, oxygen isotopes and temperature of the surface tropical ocean over the duration of the

high RSL events of the last glacial cycle (Aharon and Chappell, 1986). These uplifted coral reef terraces have been recognized and extensively studied since the pioneer works of Veeh and Chappell (1970). Huon Peninsula now represents a classical site of RSL change studies in support of the astronomical theory of climate change (Chappell, 1973).

Reef tract VII is a large structure with a barrier reef, a lagoon, and an inner fringing reef occurring at the crest of

the relief (see Figs 28 and 29). The main body of reef tract VII overlies the upper Tertiary limestone of the north Huon Peninsula (the Cromwell Limestone). Terraces VIb, VIa (above VIb), and V (below VIb) are platforms of younger raised fringing reefs that offlap the front of reef tract VII (see Figs 26 to 29). Uplift rates for the Huon Peninsula have been estimated from terrace VIIb, which has been dated at 118 to 120 ka with an assumed palaeo-RSL at 0 to +5 m (Chappell, 1974; Bloom et al., 1974; Aharon and Chappell, 1986; Chappell and Polach, 1991;

Chappell and Shackleton, 1986).



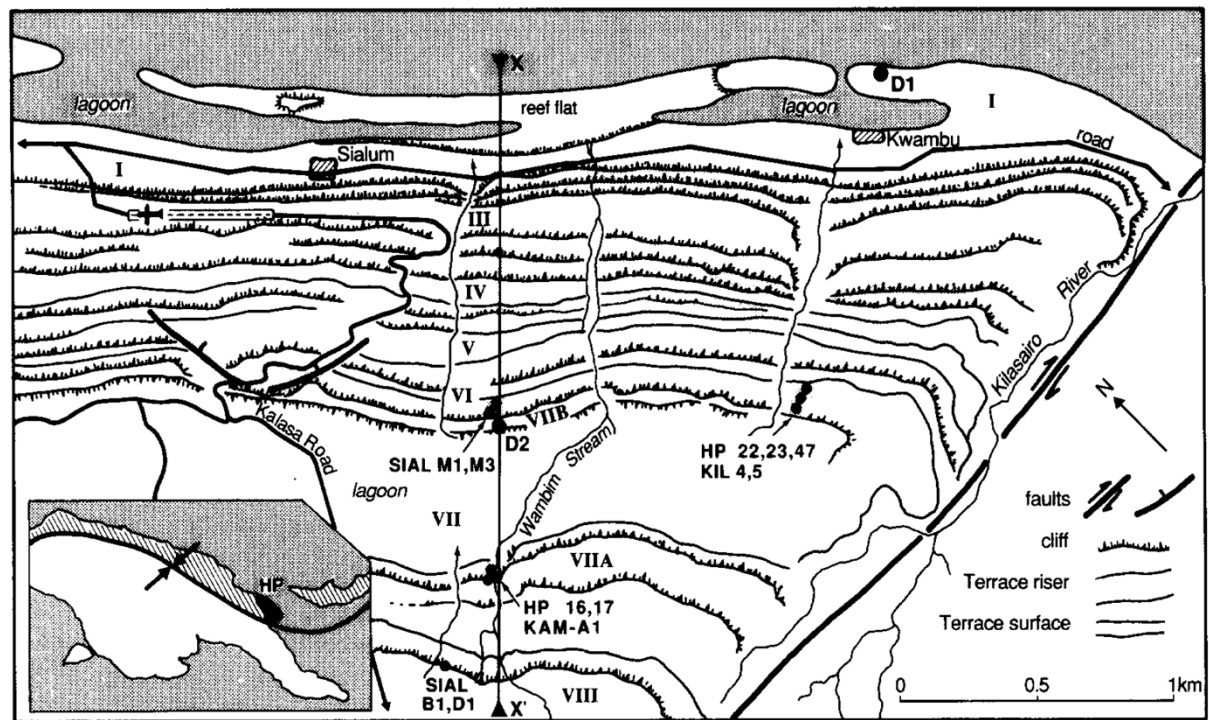

**Figure 26: Location and generalized physiography of the emerged reef terraces on Huon Peninsula (from Stein et al., 1993). Spot elevations in meters. Reef numbering (I to VIII) follows Chappell (1974). Inset map shows position of study area in Papua New Guinea.**


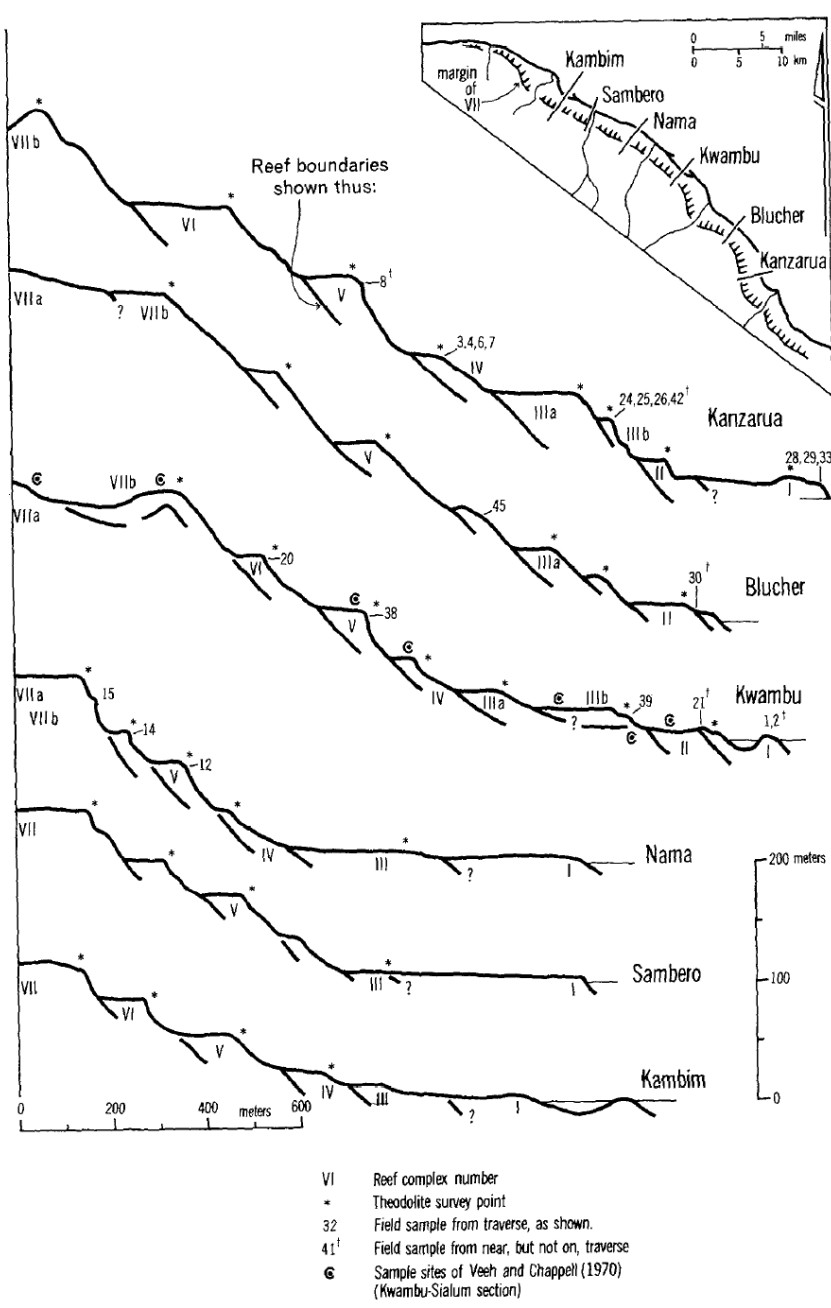

**Figure 27: Emerged reef terraces on Huon Peninsula. Reef complexes I to VII are shown by Roman numerals. Sample numbers are shown as near as possible to their collection localities (from Bloom et al., 1974).**



The first radiometric ages concerning Huon Peninsula reef terraces were obtained by Veeh and Chappell (1970)
        on coral colonies and molluscs (*Tridacna gigas*). Three U-series ages (alpha spectrometry), which could be
        assigned to MIS 5 were obtained on unidentified corals from what they called the 'Reef Complex V', reported as
        'Reef Complex VII' by Bloom et al. (1974) and interpreted as being built during the 125-120 ka time window.
        The oldest ages of 140 ± 10 ka and 133 ± 10 ka were obtained on a sample collected from fringing-reef terrace

VIIa, on the landward side of the lagoon floor of reef complex VII on the Sialum profile. The youngest ages
        reported by Veeh and Chappell (1970) were of 116 ± 7 ka and 119 ± 7 ka and concerned a sample collected from
        near the crest of the barrier reef at Sialum (near the Kwambu profile) referred to as VIIb crest by Bloom et al.
        (1974). The large errors in all these analyses did not allow a definite conclusion about the precise chronology of
        Terrace VII.

Bloom et al. (1974) reported a single datable coral sample from the undercut base of a large cliff at the front of the
        barrier of reef complex VII, about 20 m below the ridge crest, on the Nama traverse. The reported age is of 142 ±
        8 ka and concerns a *Porites lutea* colony, which most commonly occurs at water depths of less than 10 m, with
        median and average depths of 0 and 2.7 m respectively in modern environments (Table 2). However, Bloom et al.
        (1974) had insufficient samples to document the age of terrace VII and selected an arbitrary age of 125 ka for this

terrace. Based on the chronology provided by Bloom et al. (1974), Chappell (1974) has made a parallel between
        the described the raised coral terraces and coastal terraces from other parts of the world, especially Barbados where
        transgression maxima were previously established around 6.4, 29, 35, 60, 74, 118, 137, 185, 218 and >250,000 ka.
        The combined use of previous chronology and oxygen isotope records from *Tridacna gigas* occurring in uplifted
        reef terraces allowed Aharon (1983) and Aharon and Chappell (1986) to quantitatively evaluate ice volume and

temperature factors. They established that sea level rose rapidly by 5 to 6.5 m above present MSL during the LIG,
        which on stratigraphic grounds and $\delta^{18}O$ compositions, can be separated into an early phase whose $\delta^{18}O$ is similar
        to modern values (reef VIIa dated at 138 ka) and a late phase that is 0.6‰ heavier (reef VIIb dated at 118 ka).
        A comprehensive chronological study of the LIG coral reef terraces from Huon Peninsula has been carried out by
        Stein et al. (1993). Valid U-series ages reported from the Kwambu section (Sialum area) at elevations ranging

from +197 to +229 m fall into two tight groups centered at 118 ka and 134 ka (WALIS RSL IDs 464 through 470)
        and replicate with considerably more precision the few previous alpha-counting results. This confirms the
        chronological frame of reef complex VII (see Fig. 28), with ages around 118 ka to the VIIb barrier and 130-140
        ka to the VIIa fringing reef, respectively (Chappell, 1974).

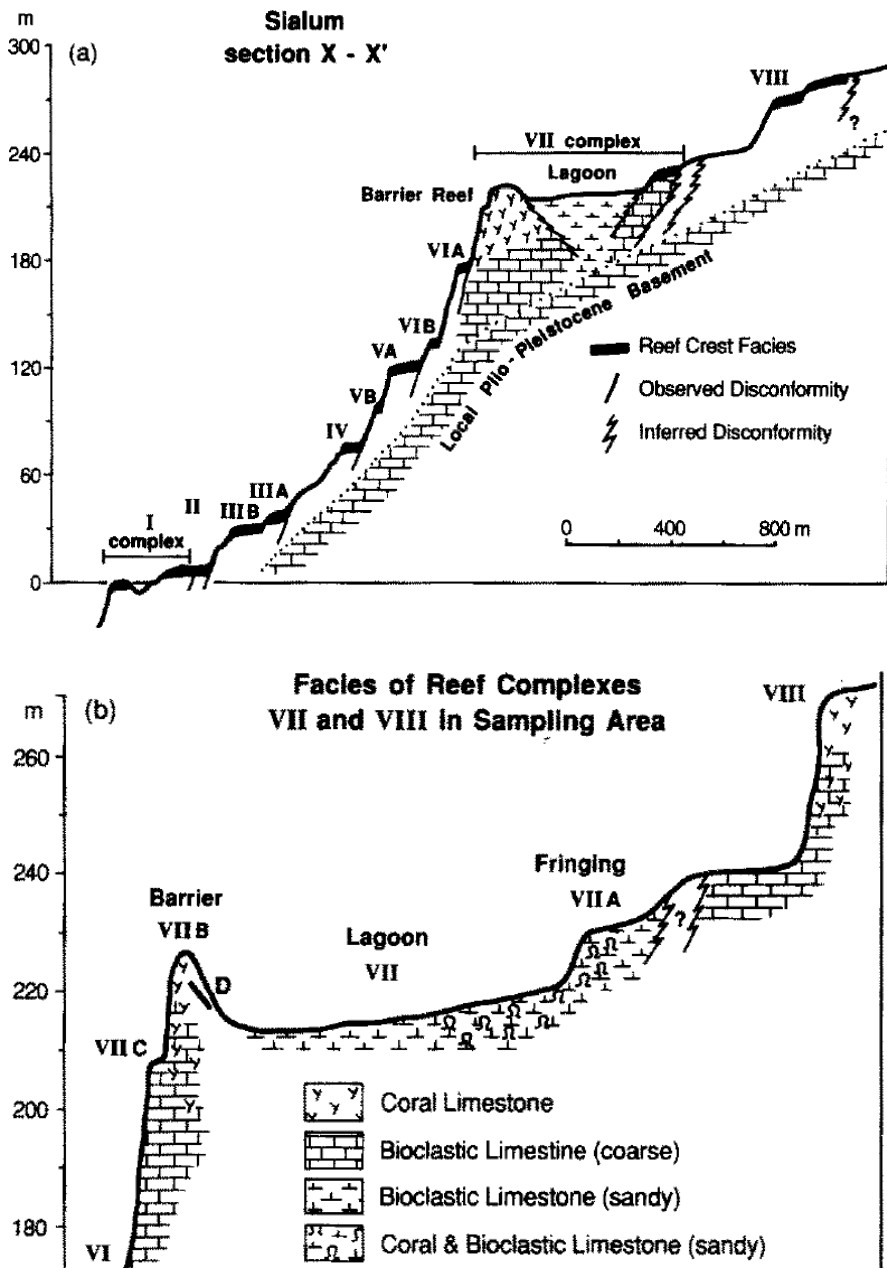


**Figure 28: (a) Vertical profiles of terraces I to VIII on transect x-x'; (b) Enlargement of reef complex VII on transect x-x' (from Stein et al., 1993).**





The group centered at 118 ka includes ages ranging from 116.4 ± 1.8 ka (WALIS U-series ID 1325; KIL-5b) to 119.5 ± 1.2 ka (WALIS U-series ID 1327; KIL-5a-1) and concern colonies of *Porites lutea* that likely indicate a

depositional environment shallower than 10 m, and colonies of *Platygyra lamellina* and *Gardineroseris planulata*, which also likely indicate a reef setting shallower than 10 m (WALIS RSL IDs 466 through 469). The group centered at 134 ka includes ages ranging from 131.9 ± 1.2 ka (WALIS U-series ID 1322; HP-23b) to 136.7 ± 1.2 (WALIS U-series ID 1313) ka and mostly concern colonies of *Gardineroseris planulata* and *Gardineroseris* sp., as well as *Cyphastrea serailia* typifying superficial environments shallower than 6 m (see Table 2; WALIS RSL

IDs 464, 465 and 470). These results show that the barrier VIIb includes corals of the two age groups, and that corals of 118 ka appear in the outer margin of VIIa, 3 m below the crest of VIIb and on the seaward surface of the barrier (30 m below the crest). This configuration suggests that the younger corals (118 ka) were unconformably deposited on the relief of the VIIb barrier. The main structure of the coral reef terrace VII is therefore believed to have developed during a major and continuous sea-level rise that commenced well before 134 ka and continued

until 118 ka (Stein et al., 1993). Results obtained by Stein et al. (1993) have been reassessed by Esat et al. (1999) who noted that these two age clusters are separated vertically by 7 m, thus indicating that the RSL at 136 ka could have been 43 m lower than at 118 ka, given the 1.9 mm/yr uplift rate (Stein et al., 1993).

Esat et al. (1999) have investigated a large ancient sea cave, Aladdin's Cave, at the base of terrace VIb (MIS 5b-5c) in the Kwangam section (see Fig. 29). Most of the acceptable ages of corals from Aladdin's Cave are within

the range of 126 to 134 ka (WALIS U-series IDs 1296 through 1299 and WALIS U-series IDs 1302 through 1312), with the exception of a coral sample dated at 115 ± 0.9 ka (WALIS U-series ID 1301; AC-U12). The corals exposed within Aladdin's Cave include abundant and large heads of *Porites* (up to 1 m in diameter) and various Faviidae (now called Merulinidae; Huang et al., 2014), indicating water depth shallower than 20 m during their growth (WALIS RSL IDs 459, 462 and 463). Ages obtained independently by TIMS and alpha spectrometry on a large

*Porites* colony at the cave entrance are in excellent agreement (WALIS U-series IDs 1307 through 1309; AC-U11a-top, AC-U11a-bottom, AC-U11-bottom) and average 127.6 ± 0.6 ka. The mean age of corals from the crest of reef VII is 134 ± 1 ka, and the mean age of the Aladdin's Cave corals of this time interval is 130 ± 2 ka. The corals at the cave appear to have grown;~4000 years later than those that are found 10 m below VIIb or ~80 m above the cave and appear to have grown in conditions 6°C cooler than present. The conflict between samples

from Aladdin's Cave (about +100 m at about 130 ka) and from the coral reef terrace VII (up to +220 m at about 134 ka) has been explained by a sea-level drop of 60 to 80 m after the development of the reef terrace VIIb (Esat et al., 1999; see Fig. 30). A new rise in sea level after 130 ka is deduced from the dataset and occurred in response

to the major insolation maximum at 126 to 128 ka (Esat et al., 1999). The 118-ka corals of reefs VIIa and VIIb may have also been deposited during an episode of sea-level rise at the end of the LIG (Esat et al., 1999).


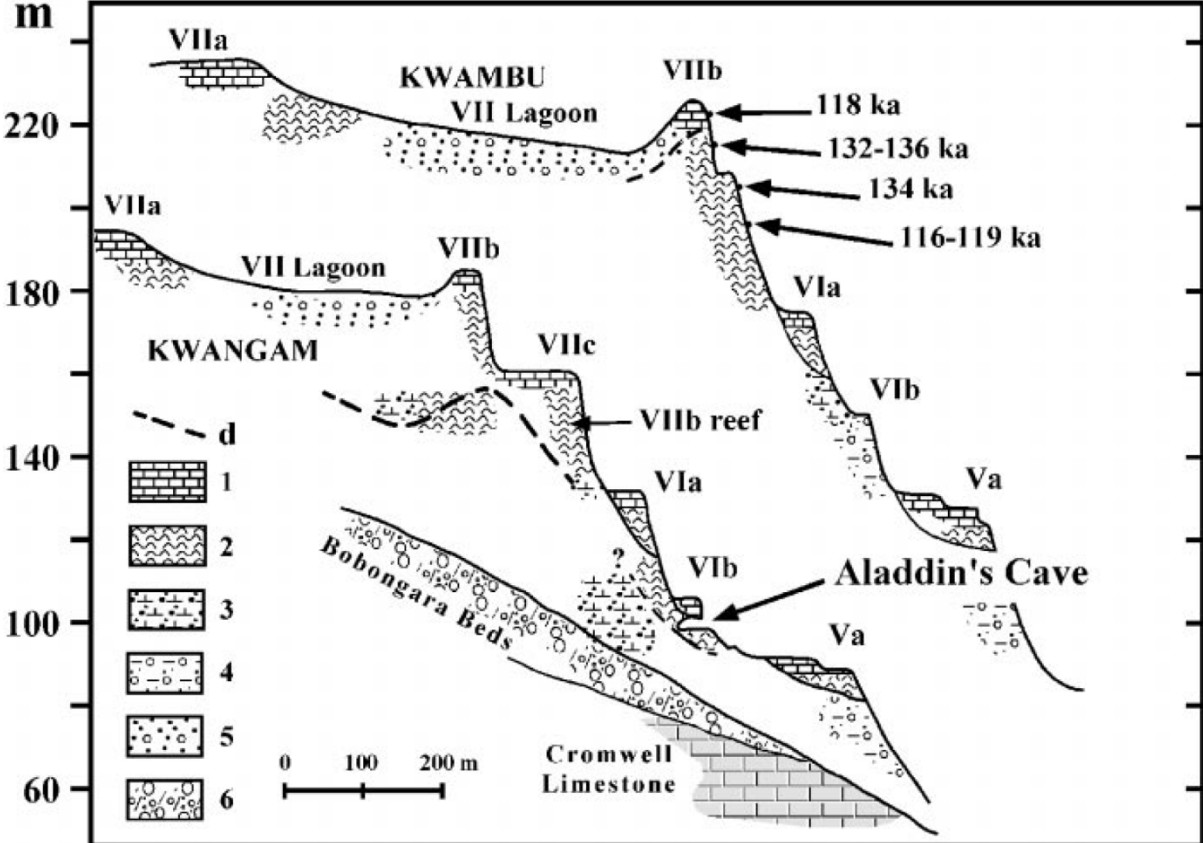

**Figure 29: A profile of coral terraces V to VII, including Aladdin's Cave at the base of reef VIb at west Kwangam and the equivalent reefs at Kwambu. The horizontal scale for both sections is 1:15,000, which was calculated from air photographs; the vertical scale for Kwambu was calculated from a theodolite survey, and the vertical scale for**
**Kwangam was calculated from calibrated digital altimeter (from Esat et al., 1999).**

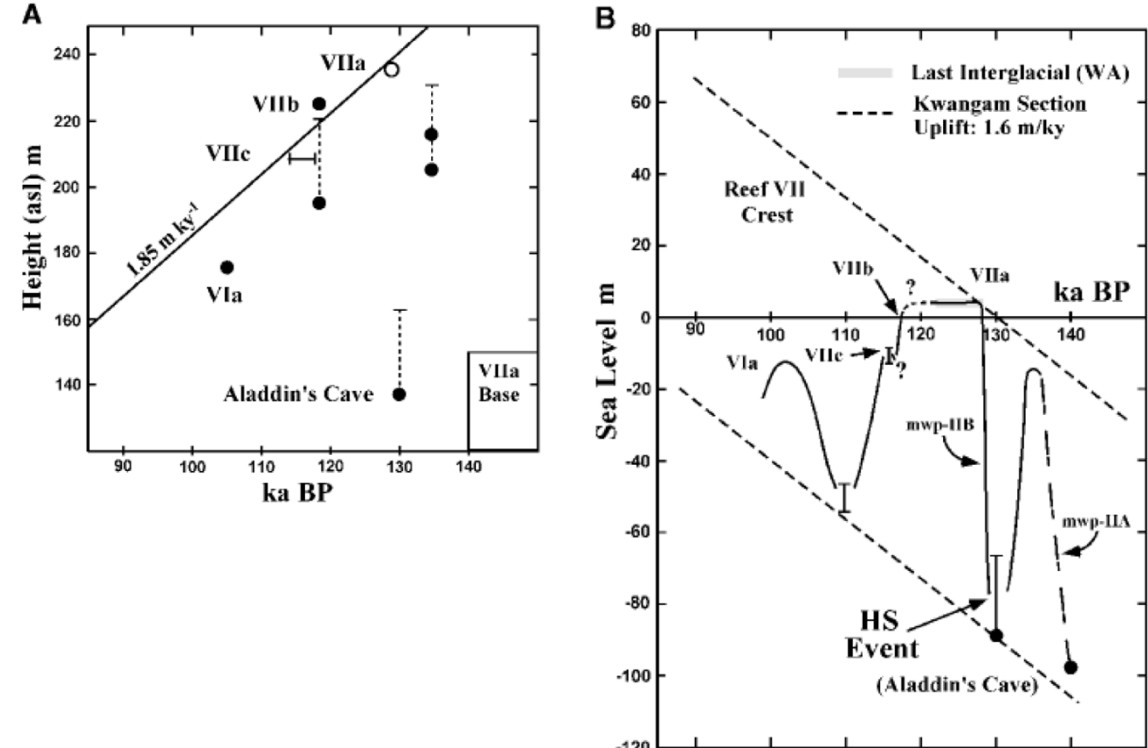


**Figure 30: (A) The age-height relation for dated terraces at Kwambu that was constructed by assuming that the top of terrace VIIa at 118 to 120 ka was 0 to 5 m above the present sea level. The dashed vertical lines indicate the possible**
**range of water depth (0 to 20 m) in which corals grew. (B) Eustatic sea-level curve for the penultimate deglacial period and the Last Interglacial. The smooth sea-level curve glosses over co-seismic uplift events at Huon Peninsula, which are discrete and meter scale and would not be discernible at this scale. The diagonal dashed lines represent the uplift with time for the crest of reef VII and Aladdin's Cave (from Esat et al., 1999).**

Cutler et al. (2003) have dated coral samples from reef terraces VI (Sialum area) and VIIb (Kilasairo area) by applying both $^{230}$Th and $^{231}$Pa dating techniques as a test of age accuracy. The two *Porites* samples from the coral reef terrace VI occurring at elevations of +169 to +177 m gave ages of 119.34 ± 0.76 ka and 121.87 ± 0.78 ka (WALIS U-series ID 1290 and 1291, respectively). The identified corals indicate a rather large palaeo-water depth range, from 0 to 20 m (WALIS RSL IDs 457 and 458). Four samples of *Gardineroseris planulata* from coral reef
terrace VIIb at +195 m yielded ages ranging from 117.77 ± 0.69 to 113.9 ± 0.65 ka (WALIS U-series ID 1289 and 1742, respectively; KIL3a and KIL3b). This result apparently confirms the U-series age of 116.6 ± 1.15 ka



previously obtained by Stein et al. (1993; WALIS U-series ID 1292), but discordant Pa-Th/U-series ages are reported (Cutler et. al., 2003). The reported corals indicate a shallow-water depositional environment, shallower than 10 m (WALIS RSL IDs 455 and 456), with median and average depths of 3 and 5.9 m respectively in modern

environments. Based on their database, Cutler et al. (2003) concluded that MIS 5e ended prior to 113.1 ± 0.7 kyr, when RSL was at -19 m. In addition, uplift rates of 1.85 mm/yr for Sialum, Kwambu, and Kilasairo, 2.6 mm/yr for Kanomi, and 2.8 mm/yr for Kanzarua, were calculated based on the dating results obtained on Terrace VIIb (Cutler et al., 2003).

Recently, Ayling et al. (2017) have investigated the chemical and isotopic distribution of uranium in *Tridacna*

*gigas* from Marine Isotope Stage (MIS) 5e (128–116 ka) and MIS 11 (424–374 ka) reefs and demonstrated that they cannot provide independent, reliable geochronological controls on the timing of past reef growth at Huon Peninsula.

### 5.13 Federated States of Micronesia

Pleistocene limestones, including the LIG period, have been described in this region, however, no dates have been

reported in the reviewed studies. The hydrogeologic study on Pingelap and Pohnpei by Ayers and Vacher (1986) indicated that the Pleistocene aquifer is located at a depth of 20-30 m. Anthony (1996a, b, c) and Anthony and Spengler (1996) provided several hydrogeologic reports on islands in Pohnpei and indicated the presence of Pleistocene deposits with a Holocene-Pleistocene contact at depths of 15-25 m below present MSL. Fletcher and Richmond (2010) summarized in a report that the Pleistocene limestone beneath the islands in this region at depths

from about 8 to 28 m below the modern reef platform was deposited during the LIG period.

### 5.14 Mariana Islands

Early studies in the 1940s to 1960s, including Stearns (1945), Tayama (1952), Cloud (1954), Cloud et al. (1956), Tracey et al. (1959, 1964) and Emery (1962, 1963), described Pleistocene reef terraces in the Mariana Islands. Benches have been described at different elevations on Saipan and Guam: at +2, +5 to +10 and +20 m (Tayama,

1952), +1.22 to +2.44 m (Emery, 1962) and +1.52, +7.62, +13.72, +21.34 and +30.48 m (Stearns, 1945). The Mariana Islands are affected by uplift, but the rates are negligible in the Mariana arc (Yonekura, 1983). Miklavič et al. (2012) concluded an average uplift rate of about 0.1 to 0.2 mm/yr.





### 5.15 Marshall Islands

Data regarding the LIG from the Marshall Islands have been obtained from drill cores. Szabo et al. (1985)

presented alpha-counting U-series datings from two drill cores on the reef flat of Enewetak, which have been

collected during the PACE program. Merulinidae in a coral-algal boundstone in core C-3 at a depth of 9.8 m

revealed ages of $131 \pm 7$ and $132 \pm 7$ ka (WALIS U-series IDs 1780 and 1781; C-3A and C-3B). Three Merulinidae

samples from core C-4, a coral-foraminiferal-skeletal grainstone containing numerous corals and abundant

*Halimeda*, at a depth of 14.9 m have been dated at $128 \pm 7$ ka, $136 \pm 8$ ka and $129 \pm 8$ ka (WALIS U-series IDs

1782-1784; C-4A, C-4B and C-4C). The dated samples contain less than 3% of calcite and the ages can be

considered as reliable. The elevation measurement method has not been reported, but the cores have been probably

hand-leveled and the elevations refer to MSL. Modern Merulinidae mainly live at water depths shallower than 30

m (WALIS RSL IDs 555 and 556). The subsidence rate of Enewetak has been estimated at 0.038-0.052 mm/yr

(Paulay and McEdward, 1990).

### 5.16 Hawaii

Emerged marine carbonate deposits are observed in several Hawaiian Islands and palaeo-shorelines may have been

affected by tsunamis and storms. The LIG Waimanalo Limestone fringes most of Oahu and consists of corals in

growth position, which are often overlain by marine coral-basalt conglomerates. Stearns (1935a, b, c) has been the

first who studied Pleistocene shorelines on the islands of Oahu and Maui describing benches at elevations of +6

to +10.7 m and have been related to the Late Pleistocene (Stearns, 1941). Beach deposits on Ulupau Head, Oahu,

occur up to 10.7 to 12.2 m above present sea level (Stearns, 1935c; Wentworth and Hoffmeister, 1939).

Several studies have provided ages for the deposits of the Waimanalo Limestone on Oahu and Rubin et al. (2000)

provided ages for Hulopoe gravel coral clasts on Lana'i.

LIG reef terraces on Oahu reach an elevation of up to 12.5 m MSL and Veeh (1966) and Ku et al. (1974) have

been the first who dated these emerged terraces. Early studies by Veeh (1966), Ku et al. (1974) and Sherman et al.

(1993) used the alpha-counting technique and later studies by Muhs and Szabo (1994), Szabo et al. (1994) and

McMurtry et al. (2010) used mass spectrometry. In 2002, Muhs et al. performed additional datings with a higher

precision at the Szabo et al. (1994) study sites using high-precision thermal ionization mass spectrometry (TIMS).

Elevations have been mostly hand-leveled and sometimes the elevation measurement method has not been

reported, but the samples have been probably hand-leveled. Uplift of Oahu was first quantified by Ku et al. (1974).

Waimanalo LIG deposits have been dated from several sites around Oahu (see Fig. 12; Fig. 1 in McMurtry et al.,

2010 and Fig. 1 in Szabo et al., 1994). Based on the depth distribution of their modern counterparts, the coral taxa





described below indicate a median and average depth of 0 to 15.6 m; however, *Pocillopora* displays a wide depth range (see Table 2). Cross-sections for some of these sites are displayed on Figure 31. Information on each site is

provided below.

1) Kahuku Point (WALIS RSL ID 542): Eight *Porites* and *Pocillopora*, of those five *in situ*, at an elevation of +1 ± 1 m MSL have been dated at 113 ± 0.7 to 137 ± 11 (alpha counting, 1-sigma uncertainty) and more accurately at 122.1 ± 1.3 ka (2-sigma uncertainty; WALIS U-series IDs 1221-1224, 1170, 1171, 1771, 1778; Ku et al., 1974; Szabo et al., 1994; Muhs et al., 2002). All ages have been accepted.

2) Mokapu Point (WALIS RSL ID 543): *Porites* and *Pocillopora*, of those six *in situ*, at an elevation of +9.5 ± 2.5 m MSL have been dated at 123.2 ± 2.6 to 134 ± 4 ka (n = 15 accepted out of 20 ages; WALIS U-series IDs 1193, 1194, 1226-1239, 1761-1762, 1800-1801; Ku et al., 1974; Szabo et al., 1994; Muhs and Szabo, 1994; Muhs et al., 2002). Muhs and Szabo (1994) dated reworked *Pocillopora* from coral-bearing beach or sublittoral sands (WALIS U-series ID 1800) and marine conglomerates with coral cobbles and heads (WALIS U-series ID 1801).

3) Alala Point (WALIS RSL ID 539): *Porites* and one *Pocillopora*, not *in situ* or not reported if *in situ*, at an elevation of +2.4 ± 2 m MSL have been dated at 119.4 ± 3 to 134.5 ± 1.4 ka (n = 6 accepted out of 8 ages; WALIS U-series IDs 1172-1177, 1207, 1208; Szabo et al., 1994; Muhs et al., 2002).

4) Maunalua Bay (WALIS RSL ID 546): *Porites* and *Pocillopora*, not reported if *in situ*, at an elevation of +7.1 ± 1 m MSL have been dated at 118 ± 3 to 121 ± 7 ka (n = 4; 1-sigma uncertainty; WALIS U-series IDs 1756-1759;

Ku et al., 1974). All ages have been accepted.

5) Black Point (WALIS RSL ID 545): *Porites*, not reported if *in situ*, at an elevation of +7 ± 1 m MSL have been dated at 112 ± 6 to 128 ± 8 ka (n = 8; 1-sigma uncertainty; WALIS U-series IDs 1748-1755; Ku et al., 1974). All ages have been accepted. *Porites* is living at median and average depths of 0 and 18.5 m, respectively in modern environments.

6) Diamond Head (WALIS RSL ID 544): *Porites* and *Pocillopora*, not *in situ* or not reported if *in situ*, at an elevation of +1.25 ± 1.25 m MSL have been dated at 114.6 ± 2.6 to 137 ± 11 ka (n = 7; 1-sigma uncertainty; WALIS U-series IDs 1209, 1210, 1773-1777; Ku et al., 1974; Szabo et al., 1994). All ages have been accepted.

7) Barbers Point (WALIS U-series ID 1779): Sherman et al. (1993) dated a single, *in situ Porites lobata* from a coral bafflestone at an elevation of +3 ± 3 m MSL revealing an (accepted) age of 115 ± 10 ka.

8) Kahe Point (WALIS RSL ID 530): Reworked *Porites* and *Pocillopora,* and one *in situ Leptoseris* (WALIS U-series ID 1240) at an elevation of +9.75 ± 2.75 m MSL have been dated at 110.9 ± 3.5 to 134 ± 4 ka (n = 14 accepted out of 17 ages; WALIS U-series IDs 1188-1192, 1216-1220, 1240-1244, 1802, 1803; Szabo et al., 1994;



Muhs and Szabo, 1994; Muhs et al., 2002; McMurtry et al., 2010). Two samples have been collected from coral-bearing conglomerates (WALIS U-series IDs 1802 and 1803; Muhs and Szabo, 1994).

9) Kaena Point State Park (WALIS RSL ID 540): *Porites*, *Pocillopora* and one *Pavona*, not *in situ* or not reported if *in situ*, at an elevation of +2 ± 2 m MSL have been dated at 115 ± 6 to 127.5 ± 0.8 ka (n = 11 accepted out of 13 ages; WALIS U-series IDs 1178-1187, 1214, 1215, 1760; Ku et al., 1974; Szabo et al., 1994; Muhs et al., 2002).

10) East of Kaena Point (WALIS RSL ID 541): *Porites* and *Pocillopora*, of those eight *in situ*, at an elevation of +1.5 ± 1.5 m MSL have been dated at 110.5 ± 3.8 to 138 ± 4 ka (n = 15 accepted out of 22 ages; WALIS U-series

IDs 1195-1206, 1211-1213, 1793-1799; Szabo et al., 1994; Muhs and Szabo, 1994; Muhs et al., 2002). Three samples have been collected from coral cobble conglomerates (WALIS U-series IDs 1797-1799; Muhs and Szabo, 1994).

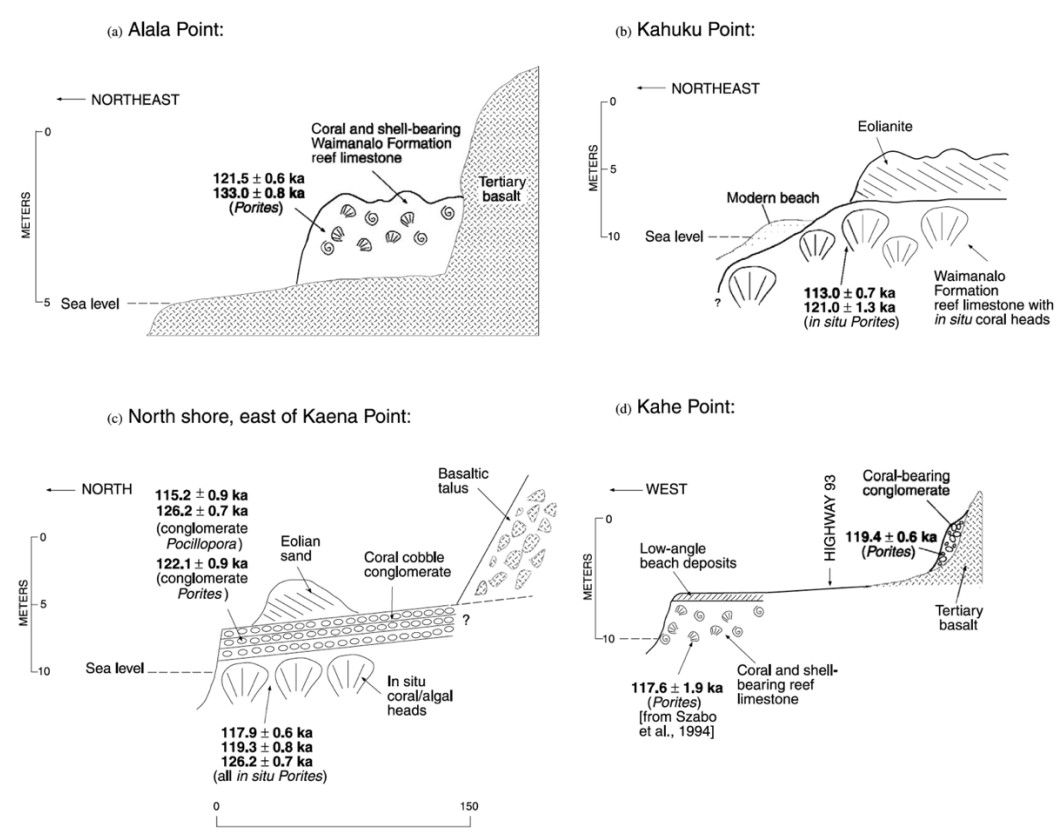

**Figure 31: Cross-sections of outcrops of the Waimanalo Formation exposed on (a) Alala Point, (b) Kahuku Point, (c)**
**the north shore of Oahu east of Kaena Point and (d) Kahe Point along with reliable U-series ages of corals (from Muhs**
**et al., 2002).**



Veeh (1966) studied several sites on Oahu, which are detailed in Veeh (1965). *In situ Leptastrea* at an elevation of +2.25 ± 1.25 m MLWS have been dated at 110 ± 20 to 140 ± 30 ka (n = 4; WALIS RSL ID 547; WALIS U-series IDs 1739-1741, 1743; OAHU2, OAHU5, OAHU9 and OAHU22). However, the use of the alpha-counting technique resulted in large age uncertainties. *Leptastrea* lives at median and average depths of 7 and 17 m, respectively.

Emerged reef terraces on Oahu described in all reviewed studies reach an elevation of up to +12.5 m MSL. The majority of ages was obtained from the first 4 m of the reef terraces and they range between 110.5 ± 3.8 and 138 ± 4 ka (n = 48 accepted ages). The terrace at about +7 m reveals ages of 112 ± 6 to 128 ± 8 ka (n = 12 accepted ages). The +9 to +12 m terraces have been dated at 110.9 ± 3.5 to 134 ± 4 ka (n = 29 accepted ages). Sherman et al. (1993) deduced the occurrence of two highstands during the LIG. Szabo et al. (1994) reported on a LIG sea-level high stand on Oahu, which lasted about 17,000 years from 114 ka to 131 ka. Based on corals in growth position, Szabo et al. (1994) estimated that RSL was at least 7 m higher than present at 131 ka and remained higher until at least 115 ka. These authors mention a considerable 1-27 m depth range of living *Porites* and *Pocillopora*. Muhs et al. (2002) performed additional datings with a higher precision at the Szabo et al. (1994) study sites and confirmed the concept of a long LIG period on Oahu as proposed by Szabo et al. (1994). In 2002, Muhs et al. performed additional datings with a higher precision at the Szabo et al. (1994) study sites and confirmed the concept of a long LIG on Oahu as proposed by Szabo et al. (1994). However, they did not report any evidence for two separate sea-level stands as concluded by Sherman et al. (1993). McMurtry et al. (2010) did not consider the age data by Szabo et al. (1994) (white triangles in Fig. 32) as they seem to be biased toward lower elevation estimates. Furthermore, these authors did not consider the ages of elevated deposits from stages 5e and 9 (colored triangles) as they derived from energetic deposits related to storms or tsunamis rather than being elevated reefs with corals in growth position. Besides MIS 5, other interglacials from MIS 7 to 13 can be found in the coral reef record from Oahu. MIS 7 reefs have been documented around Oahu where they form a gently sloping nearshore submerged terrace (Sherman et al., 2014). The most reliable maximum elevation and the mean age for each highstand or terrace from Oahu is summarized in Figure 32. The MIS 5 terrace, the Waimanalo formation, is usually found at an elevation of +7.5 m. Based on this dataset, a linear uplift for Oahu has been estimated at 0.060 ± 0.001 mm/yr over the past 500 ka (McMurtry et al., 2010), resulting in an uplift of about 7.5 m since 125 ka.



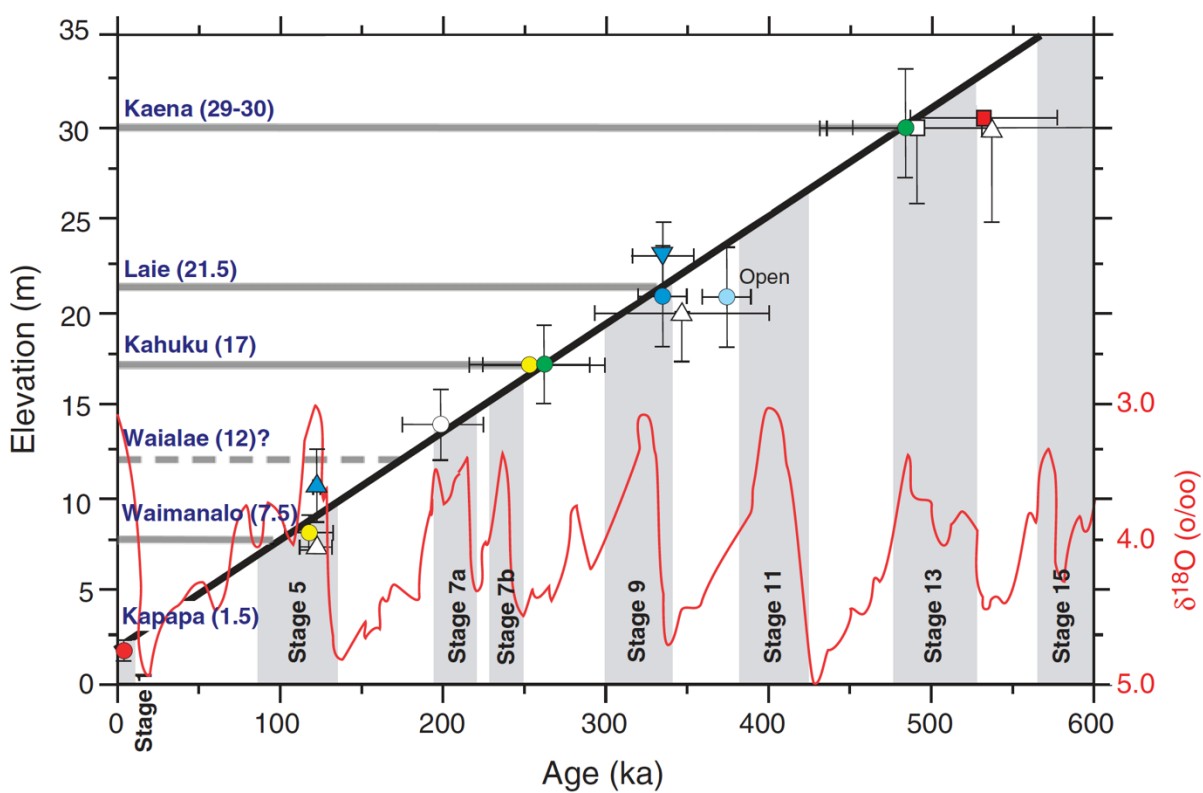


**Figure 32: Mean or best ages versus maximum measured terrace elevation for known elevated reefs on Oahu (from McMurtry et al., 2010).**

Uncertainties of elevation measurements and datings are often large and do not allow an accurate conclusion about

the timing and magnitude of the LIG RSL on Oahu. Furthermore, taxa have been often not determined, and therefore, an accurate palaeo-RSL could not be reconstructed. Determinations at the genus level often allow only rough estimates of palaeo-RSL.

Rubin et al. (2000) provided high-precision TIMS U-series ages for emerged coral reef terraces on Lana'i. These authors dated Hulopoe gravel coral clasts (no taxa determination, not *in situ*) at elevations of +19.5 ± 0.5 m MSL

and +21 ± 2 m MSL and obtained ages of 130.9 ± 1.5 ka and 134.8 ± 0.9 ka (WALIS RSL ID 552; WALIS U-series IDs 1806 and 1807; Lan1-2h-1 and Lan3-6-2) as well as 136.4 ± 0.9 ka and 136.7 ± 1.0 ka (WALIS RSL ID 553; WALIS U-series IDs 1808 and 1809; Lan3-5-3 and Lan2-4-3), respectively. The dated samples contained less than 2.3% of calcite and the ages can be considered as reliable. The elevation measurement method has not





been reported, but the samples have been probably hand-leveled and the elevations refer to MSL. A water depth

range of 0-30 m has been assumed as no taxa identification has been reported. The mean uplift rate of Lana'i has

been estimated at $0.15 \pm 0.22$ mm/yr (Rubin et al., 2000).

## 6 Further details

### 6.1 Last Interglacial sea-level fluctuations

Studies regarding tropical Pacific islands have contributed to the long-lasting and ongoing debate regarding the

course of the LIG RSL curve, including the timing, duration and amplitude of the MIS 5e highstand. This is of

prime importance for many tropical Pacific islands, as the elevation of the LIG peak has been considered as a

reference to assess rates of vertical displacement, uplift or subsidence.

The occurrence of two sea-level peaks during the LIG highstand was inferred from the study of LIG reef terraces

from various tropical Pacific islands: Papua New Guinea (Chappell and Veeh, 1978), Vanuatu (Jouannic et al.,

1982), Fiji (Nunn and Omura, 1999; Nunn et al., 2002) and Hawaii (Sherman et al., 1993) and, since then, in a

number of regions, including the Caribbean (Blanchon et al., 2009; Thompson et al., 2011), even if the timing of

this event is not always substantiated by radiometric ages (see Camoin and Webster, 2015).

- In Papua New Guinea, ages obtained on the VIIa and VIIb terraces, that correspond to fringing and barrier

1015        reefs, respectively (Bloom et al., 1974; Chappell, 1974), fall into two tight groups centered at 118 ka and

134 ka. These two age clusters are separated vertically by 7 m, thus indicating that the RSL at 136 ka

could have been 43 m lower than at 118 ka, given the 1.9 mm/yr uplift rate (Stein et al., 1993). The

conflict between samples from Aladdin's Cave (about 100 m above present MSL and about 130 ka) and

from the coral reef terrace VII (up to 220 m above present MSL and about 134 ka) has been explained by

1020        a sea-level drop of 60 to 80 m after the development of the reef terrace VIIb (Esat et al., 1999). A new

rise in sea level after 130 ka is deduced from the dataset and occurred in response to the major insolation

maximum at 126 to 128 ka (Esat et al., 1999). The 118-ka corals of reefs VIIa and VIIb may have also

been deposited during an episode of sea-level rise at the end of the LIG (Esat et al., 1999).

- On Vanuatu, the duplication of the MIS 5e terrace in the Port Havannah area displays striking similarities

1025        with the distribution of VIIa and VIIb reef terraces from Papua New Guinea (Bloom et al., 1974;

Chappell, 1974) and has been interpreted as resulting from low amplitude sea-level fluctuations during

MIS 5e (Jouannic et al., 1982).

- On Fiji Islands, the age distribution is thought to reflect a double sea-level maximum with peaks around

133-130 ka and 123-120 ka (Nunn et al., 2002). The earlier maximum has been reported about 2 m lower





than the later and marked by the growth of a surface reef, the later maximum appearing to have involved only cutting of erosional shorelines at the +5.0 to +5.5 m level (Nunn and Omura, 2003).

- The occurrence of two separate sea-level highstands on Oahu during the LIG has been considered by Sherman et al. (1993). In contrast, the concept of a long LIG period of about 17,000 years, from 114 ka to 131 ka, on Oahu has been proposed by Szabo et al. (1994) and later confirmed by Muhs et al. (2002).

This concept involves an apparent local sea-level position 7 m higher than its current one by 131 ka, which remained higher than present until at least 115 ka (Szabo et al., 1994).

- On the Cook Islands, Woodroffe et al. (1991) did not observe two periods of high RSL separated by a period of lower RSL during the LIG. However, they describe several stratigraphic features, such as erosional benches and notches, that might indicate RSL fluctuations during MIS 5e. However, these

fluctuations have not been substantiated by chronological results.

### 6.2 MIS 5a and MIS 5c

Since our database is specifically focused on MIS 5e reef records, the table below lists data concerning MIS 5a and 5c, which were obtained on tropical Pacific islands but not inserted in our database. These data might be useful for future research on MIS 5 shorelines in tropical Pacific regions and are reported in this description paper for

completeness.

| Region | Sub-region / Site | Description |
|---|---|---|
| Vanuatu | Malakula | 5c terrace at +204 m with an age of 107.6 ± 1.1 ka (Cabioch and Ayliffe, 2001)<br>Terrace at +30 m with an age of 98 ± 11 ka. The terrace is assumed to correspond to the 103 ka palaeo- RSL.<br>Terraces correspond to 82 and 103 ka terraces from Bloom et al., 1974 (Jouannic et al., 1980, 1982) |
| | Efate | Terrace at +80 m dated at 106.4 ± 6.4 ka BP and at 103.5 ka (Tututuk area; Lecolle and Bernat, 1985) |



|  |  |  |
|---|---|---|
|  |  | Terraces correspond to 82 and 103 ka terraces from Bloom et al., 1974 (Lecolle and Bernat, 1985; Jouannic et al., 1982)<br>Terrace at + 70 m dated at 108 ka (Pangona area; Lecolle and Bernat, 1985)<br>Terrace at +20 m dated at 107.9 ka (Siviri area; Lecolle and Bernat, 1985) |
|  | Santo | Terraces correspond to 82 and 103 ka terraces from Bloom et al., 1974 (Jouannic et al., 1980, 1982) |
|  | Malo | Terraces correspond to 82 and 103 ka terraces from Bloom et al., 1974 (Jouannic et al., 1982) |
|  | Hiw, Tegua, Loh, Toga | Terraces correspond to 82 and 103 ka terraces from Bloom et al., 1974 (Taylor et al., 1985). These reefs probably correspond, respectively, to the 80 and 103 ka reefs on New Guinea and Barbados. |
|  | Tanna | Terraces attributed to stages 5a, 5b and 5c (Neef et al., 2003) |
| New Caledonia | Lifou | Terrace at +2.2 m dated at $90 \pm 4$ ka (Bernat et al., 1976) |
|  | Maré | Terrace at +2 to +5 m gave ages of the order of 90-98 ka (Bernat et al., 1976). |
|  | Grande Terre (Amédée drill cores) | Unit 10 (37 to 14 mbsl) assigned to MIS 5 (100 to 130 ka; Cabioch et al., 2008) |




| Papua New Guinea | | Prominent reef terrace of reef complex V dated at 85 ka (Bloom et al., 1974)<br>Reef terraces V, VI, VIIb taken as 82, 103, and 124 ka (Bloom et al., 1974)<br>High RSL at 105 ka (Chappell, 1974)<br>Reef terraces V and VI dated at 85 ± 1.4 ka and 107 ± 7.5 ka, respectively (Aharon and Chappell, 1986)<br>Reef crest VIa assumed to have formed at 105 ka at 10 m below the present MSL (Esat et al., 1999)<br>MIS 5b RSL was -57 m at 92.6 ± 0.5 ka; drop of about 40 m in approximately 10 kyr during MIS 5c-5b transition. Sea-level rise more than 40 m during the MIS 5b-5a transition, also in about 10 kyr. MIS 5a lasted until at least 76.2 ± 0.4 kyr, at a level of -24 m at that time (Cutler et al., 2003). |
| Hawaii | Oahu | Presence of MIS 5a and/or 5c on Oahu (Sherman, 1999)<br>Presence of MIS 5a-d on Oahu (Sherman et al., 2014) |

**Table 5: MIS 5a and MIS 5c reviewed in this study.**





### 6.3 Other interglacials

The table below includes data concerning the description and/or the dating of other Pleistocene interglacial periods on tropical Pacific islands from a number of studies. These data are not included in our database, which concerns
only MIS 5e reef records, but are of pivotal importance for further projects dealing with the reconstruction of sea-level changes and shoreline evolution during the Pleistocene in tropical Pacific islands.

| Region | Sub-region / Site | Description |
|---|---|---|
| French Polynesia | Moruroa | MIS 7 and 9 reported (Camoin et al., 2001; Braithwaite and Camoin, 2011) |
| | Takapoto | MIS 7 and 9 reported (Montaggioni et al., 2019a) |
| Cook Islands | Atiu, Mauke, Mitiaro, Mangaia | MIS 7 reported (Woodroffe et al., 1991) |
| | Pukapuka, Rakahanga | MIS 7, 9, 11, 13 and 15 indicated (Gray et al., 1992) |
| Niue | | MIS 7 reported (Kennedy et al., 2012) |
| Tonga | Tongatapu | MIS 7 probably present (Taylor and Bloom, 1977) |
| Fiji | Kadavu, Kaibu | Reef terrace at +3.2 to +4.8 m dated from at 223 ± 7.6 to 207.2 ± 5.9 ka on Kadavu Island (Nagigia area; Nunn and Omura, 1999). These dates may indicate that the two reefs cannot be considered |





| | | |
|---|---|---|
| | | as separate entities and that the Nagigia reef is thus interpreted as having grown upwards to a RSL at least 7.1 m higher than present MSL, perhaps c. 208 ka. Nunn and Omura (2003) have considered that, in the absence of U-series ages, the two higher terraces on Kaibu Island, at 8.0-9.2 m and 12-14 m, could have formed during Stage 7 RSL maxima. |
| Tuvalu | Funafuti | Reef sequence between 27 and 58 m core depth in Funafuti drill cores attributed to stage 7. A Sr isotope age of $0.21 \pm 0.21$ Ma obtained on an unidentified coral collected at 36.6 m below the reef surface (Ohde et al., 2002). |
| Vanuatu | Tanna | MIS 7 reported (Neef et al., 2003) |
| | Futuna | MIS 7, 9, 11 and 13 (Neef and McCulloch, 2001) |
| New Caledonia | Loyalty Islands | Terrace at +2.5 m dated at ca 225 ka (Bourrouilh-Le Jan, 1985) |
| | Grande Terre (Amédée drill cores) | MIS 7 and probably 9 or 11 (Frank et al., 2006) MIS 7 (221 to 248 ka) relate to Unit 9 (40–37 m core depth; Cabioch et al., 2008) |
| | Grande Terre (Ténia drill cores) | MIS 7 not identified; MIS 9 and probably MIS 11 and 13 reported (Montaggioni et al., 2011) |





| Papua New Guinea | | MIS 7 reported (Omura et al., 1994) |
|---|---|---|
| Marshall Islands | Enewetak | MIS 7 or 9, MIS 9 or 11 or 13 probably found (Szabo et al., 1985) |
| Hawaii | Oahu | MIS 11, 13 or 15 probably present (Szabo et al., 1994)<br>MIS 7 reported (Sherman et al., 1999; 2014)<br>MIS 11 reported (Hearty, 2002)<br>MIS 7, 9 and 13 reported; no evidence for MIS 11 (McMurtry et al., 2010) |
| | Lana'i, Moloka'i | MIS 7 reported (Rubin et al., 2000) |

**Table 6: Other interglacials reviewed in this study.**







### 6.4 Holocene sea-level indicators


The table below summarizes data concerning Holocene sea-level records from tropical Pacific islands. It especially includes results obtained on a mid-Holocene RSL highstand that appears as a significant feature in this region (Grossman et al., 1998; Dickinson, 2001, 2003, 2004; Woodroffe and Horton, 2005; Nunn, 2007; Hallmann et al., 2018). Studies of Holocene sea-level indicators for the reviewed islands are listed chronologically. These data are not inserted in our database, which specifically concerns MIS 5e reef records of RSL changes.


| Region | References |
|---|---|
| French Polynesia | Lalou et al. (1966); Guilcher et al. (1969); Labeyrie et al. (1969); Montaggioni and Pirazzoli (1984); Trichet et al. (1984); Montaggioni (1985); Pirazzoli et al. (1985a, b); Pirazzoli and Montaggioni (1986); Pirazzoli (1987); Pirazzoli and Montaggioni (1987); Pirazzoli et al. (1987); Pirazzoli and Montaggioni (1988); Pirazzoli et al. (1988a, b); Camoin et al. (2001); Braithwaite and Camoin (2011); Rashid et al. (2014); Gischler et al. (2016); Hallmann et al. (2018); Gischler et al. (2019); Montaggioni et al. (2019b) |
| Cook Islands | Schofield (1970); Stoddart (1972, 1975); Scoffin et al. (1985); Spencer et al. (1988); Yonekura et al. (1988); Woodroffe et al. (1990, 1991); Allen (1992); Gray et al. (1992); Ellison (1994); Yonekura (1994); Chikamori (1996); Allen (1998); Chikamori (1998, 2001); Moriwaki et al. (2006); Goodwin and Harvey (2008); Allen et al. (2016) |
| Samoa | Mayor (1921, 1924); Stearns (1944); Kear and Wood (1959); Fairbridge (1961); Grant-Taylor and Rafter (1962); Shepard (1963); Stice and McCoy (1968); Matsushima et al. (1984); Rodda (1988); Sugimura et al. (1988a); Nunn (1991); Goodwin and Grossman (2003) |





| Niue | Schofield (1959); Paulay and Spencer (1992); Nunn and Britton (2004); Kennedy et al. (2012) |
|---|---|
| Tonga | Taylor and Bloom (1977); Kirch (1978); Taylor (1978); Ellison (1988); Nunn (1991); Dickinson et al. (1994), Nunn (1994, 1995a); Nunn and Finau (1995); Spenneman (1997) |
| Fiji | Taylor (1978); Berryman (1979); Green (1979); Rodda (1986); Ash (1987); Miyata et al. (1988); Nunn (1988); Roy (1988); Sugimura et al. (1988b); Miyata et al. (1990); Nunn (1990a, b); Shepard (1990); Nunn (1995b); Spriggs (1997); Nunn (2000); Nunn et al. (2000); Nunn and Peltier (2001); Thomas et al. (2004); Mörner and Matlack-Klein (2017) |
| Tuvalu | David and Sweet (1904); Schofield (1977a); Kaplin (1981); McLean and Hosking (1991); Dickinson (1999); Ohde et al. (2002) |
| Kiribati | Tracey (1972); Schofield (1977a, b); Marshall and Jacobson (1985); Woodroffe and McLean (1998); Woodroffe and Morrison (2001); Woodroffe et al. (2012); Yamano et al. (2017) |
| Solomon Islands | Kirch and Yen (1982); Thirumalai et al. (2015) |
| Vanuatu | Jouannic et al. (1980, 1982); Taylor et al. (1985); Cabioch and Ayliffe (2001); Neef et al. (2003) |
| New Caledonia | Baltzer (1970); Coudray and Delibrias (1972); Bourrouilh-Le Jan (1985); Cabioch et al. (1989, 1995); Frank et al. (2006); Cabioch et al. (2008); Wirrmann et al. (2011) |



| Papua New Guinea | Chappell and Polach (1972, 1991); Ota et al. (1993); Chappell (1994); Chappell et al. (1996); Riker-Coleman et al. (2006) |
|---|---|
| Federated States of Micronesia | Wiens (1962); Shepard et al. (1967); Tracey (1968); Curray et al. (1970); Bloom (1970); Newell and Bloom (1970); Leach and Ward (1981); Matsumoto et al. (1986); Athens (1995); Kawana et al. (1995); Fujimoto et al. (1996) |
| Mariana Islands | Kuenen (1933); Stearns (1941); Cloud et al. (1956); Tracey et al. (1964); Curray et al. (1970); Kayanne et al. (1993) |
| Marshall Islands | Wiens (1962); Thurber et al. (1965); Shepard et al. (1967); Curray et al. (1970); Newell and Bloom (1970); Tracey and Ladd (1974); Buddemeier et al. (1975); Dickinson (1999); Kayanne et al. (2011) |
| Hawaii | Stearns (1941); Gross et al. (1969); Ladd et al. (1970); Stearns (1974); Easton and Olson (1976); Athens and Ward (1991); Fletcher and Jones (1996); Allen (1997); Athens (1997); Grigg (1998); Grossman and Fletcher (1998); Matsumoto et al. (1988); Jones (1992, 1998); Calhoun and Fletcher (1996); Rubin et al. (2000); Carson (2003, 2004); Nunn et al. (2007) |

**Table 7: Holocene sea-level indicators reviewed in this study.**

**6.5 Controversies**

Besides past debates regarding the course of LIG RSL curve (see 6.1 'Last Interglacial sea-level fluctuations'), the study of MIS 5e reefs on tropical Pacific islands has generated several controversies mostly related to the interpretation of reef terraces. For example, McMurtry et al. (2010) did not consider the ages obtained by Szabo et al. (1994) on MIS 5e reef terrace on Oahu as they interpreted these deposits as being derived from energetic deposits related to storms or tsunamis rather than as corals in growth position. A similar controversial issue has concerned the interpretation of LIG deposits on Lana'i seen either as highstand deposits, then mega-tsunami



deposits, and eventually re-interpreted as highstand shoreline features deposited during the last two interglacials (see review in Webster et al., 2007).

**6.6 Uncertainties and data quality**

Overall, 284 data points summarized in 84 RSL indicators and based on 35 studies on MIS 5 coral reef records in the tropical Pacific indicators have been inserted in the database. One key aspect concerning the data quality is that 25 studies are older than 20 years and 14 studies are even older than 30 years. This impacts the quality of elevation measurements and the age information as methods improved considerably over the last decades.

The data quality is mostly affected by the quality of the RSL data and, to a lesser extent, by the quality of age information. About 80% of the data points reveal a very poor RSL quality as the coral taxa have been determined only at the genus (about 70% of the data points) or family (about 10% of the data points) level. For about 15% of the data points no coral taxa has been reported at all. This introduces very large RSL uncertainties of ± 7-15 m and excludes any constraint regarding the RSL position during MIS 5e. Some coral genera, e.g. *Porites* and *Pocillopora*

that are the most commonly studied genera, display wide depth ranges in modern environments with median-average depths of 0-18.5 m and 0-15.6 m, respectively, thus precluding an accurate RSL reconstruction. Additional 10% of the data points reveal a poor RSL quality with final RSL uncertainties of more than three metres. Only less than 10% of the RSL data are of an average or good quality with final RSL uncertainties between one and three metres. These data are based on coral genera, e.g. *Leptoria*, which live at narrower depth ranges and can therefore

be used to reconstruct RSL with uncertainties of about ± 2 m. Consequently, 90% of the RSL information is not reliable and does not allow accurate RSL reconstructions during MIS 5e. Coral assemblages would need to be identified carefully and their palaeo-water depth significance determined accurately. A special attention should be given to RSL indicators typified by a narrow depth range to better constrain RSL reconstructions.

Age information quality of the data points is significantly higher than their RSL quality. Age information of one

third of the data points is of excellent quality, i.e. ages have been accurately measured with very narrow age ranges and uncertainties of up to ± 2 ka. Another third of the data points are still of good to average quality concerning the age information. 'Good quality' means that the ages of the coral samples could be determined with narrow age ranges and uncertainties of ± 2-4 ka allowing the attribution to a specific substage of MIS 5. 'Average quality' refers to ages with uncertainties of ± 4-15 ka so that the RSL data point can be attributed only to a generic

interglacial (e.g. MIS 5). The use of the alpha-counting technique resulted in large age uncertainties (± 4-20 ka). Based on this method, another third of the ages reveal large uncertainties with more than ± 15 ka. Ages have been accepted in about 50% of the original studies, about 20% of the ages have not been accepted and for the data points

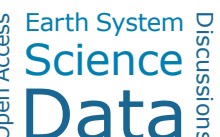

it is not known. A further problem is that the preservation of the samples is often unknown, implying that the reliability of the ages presented in the reviewed studies cannot be addressed. Only 40% of the studies report a

calcite content of less than 3%, i.e. the samples are well preserved and the obtained ages can be considered as reliable. The other 60% of the studies do not mention the preservation of the dated coral samples, so that the published age cannot be confirmed.

Another source of uncertainties are inaccurate elevation measurements. Many studies used hand levelling and metered tape to determine the elevations of coral samples resulting in uncertainties of about ± 0.5-1 m. Some

studies did even not report elevations and thus they had to be extracted from published sketches, which introduced large uncertainties. Overall, the elevation uncertainties are significant and might be up to 2-4 m, but they are still overprinted by the very large palaeo-water depth uncertainties introduced by the common lack of accurate studies of coral assemblages. Accurate GPS measurements would be needed to better constrain the elevation of RSL indicators.

A further important aspect for sea-level studies is that the corals need to be in growth position to provide reliable information on past sea levels. However, for about 85% of the data points the original studies do not mention if the dated corals have been collected in growth position. This introduces large uncertainties, even when these corals have been accurately dated, as they cannot be therefore considered as reliable sea-level indicators.

The overall quality of this database is very poor mainly due to the poorly constrained palaeo-water depth range of

coral assemblages, which introduces the largest uncertainty. The inaccuracy of elevation measurements, the lack of information on coral preservation, as well as the poor/missing information regarding the coral occurrence in LIG reefs introduce additional biases. Future work using more accurate dating methods and elevation measurements as well as a more detailed identification of the collected coral samples are essential to better constrain the MIS 5e RSL dataset concerning the tropical Pacific islands.


**7 Future research directions**

RSL change reconstructions require the combination of reliable radiometric ages and elevation measurements, as well as an accurate estimate of palaeo-water depths deduced from the modern distribution of relevant reef communities. Our newly compiled database (Hallmann and Camoin, 2020) demonstrates that most of the studies

that have been carried out on tropical Pacific islands do not satisfy these requirements, thus hampering the accurate reconstruction of LIG RSL history.

Future research directions may therefore require to revisit LIG reef records from tropical Pacific islands, especially the key 'reference sites' (e.g. Papua New Guinea, Hawaii, Vanuatu), in order to collect the missing information

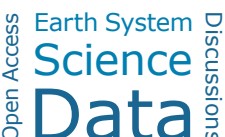

that is crucially needed to reconstruct properly LIG RSL changes. More accurate constraints on the timing, duration

and amplitude of the highstand is pivotal to better constrain glacial isostatic adjustments during the LIG. In addition, despite the extensive literature on LIG RSL changes, only a few papers have tried to estimate suborbital-scale interglacial RSL variability during that time window (see reviews in Dutton and Lambeck, 2012; Woodroffe and Webster, 2014; Camoin and Webster, 2015). Higher resolution dating combined with accurate sedimentological studies of LIG reefs will be needed to decipher and reconstruct high-frequency RSL oscillations

related to shorter climate excursions during the LIG period.

Finally, the full understanding of the impact of RSL changes at various time scales on LIG reef growth will require the study of reef development patterns that have been so far barely investigated.

## 8 Data availability

The database is available open access, and kept updated as necessary, at the following link:

http://doi.org/10.5281/zenodo.3991672 (Hallmann and Camoin, 2020). The files at this link were exported from the WALIS database interface on 19/08/2020 and collated into a single .xls file from the different users who inserted them (G. Camoin and N. Hallmann). Description of each field in the database is contained at this link: https://doi.org/10.5281/zenodo.3961543 (Rovere et al., 2020), that is readily accessible and searchable here: https://walis-help.readthedocs.io/en/latest/. More information on the World Atlas of Last Interglacial Shorelines

can be found here: https://warmcoasts.eu/world-atlas.html. Users of our database are encouraged to cite the original sources alongside with our database and this article.

## Author contribution

NH and GC wrote the manuscript and compiled the database; MH provided the palaeo-water depth significance of coral assemblages; MH and JW contributed to the manuscript during the final stage; NH produced the figures

## Competing interests

The authors declare that they have no conflict of interest.



**Acknowledgments**

The data used in this study were compiled in WALIS, a sea-level database interface developed by the ERC Starting
Grant "WARMCOASTS" (ERC-StG-802414), in collaboration with PALSEA (PAGES / INQUA) working group.
The database structure was designed by A. Rovere, D. Ryan, T. Lorscheid, A. Dutton, P. Chutcharavan, D. Brill,
N. Jankowski, D. Mueller, M. Bartz. The data points used in this study were contributed to WALIS by G. Camoin
and N. Hallmann. Basemap: "National Geographic Map", with data from National Geographic, Esri, Garmin,
HERE, UNEP-WCMC, USGS, NASA, ESA, METI, NRCAN, GEBCO, NOAA, INCREMENT P Corp (Figs 5A,
6, 7A, 8A-B, 9-12).

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
