# Peer review of "A standardized database of Marine Isotopic Stage 5e sea-level proxies on tropical Pacific Islands"

_Earth System Science Data, 2020_

## Short Comment (SC1) · 9 Dec 2020

First, looking at Table 1, Relative SL indicators, there are problems with the narrow indicative range of coral terraces inferred by Rovere et al 2016.

It is stated that coral reef 'terraces' develop between the mean low water and breaking depth. However this ignores the fact that in the open Pacific coralgal reef flats lie as much as 7 m above mean ocean level due to a combination of year-round ocean swell, wave-set up and lagoonal ponding (yes I was shocked too by the magnitude). This is also the case on reef-flat terraces in areas with macro-tidal ranges, where corals can grow above mean sea level and survive exposure during daily low tides. This makes coral depth ranges in Table 2 somewhat debatable given that none show the possibility

of growth above low tide level.

This important point that has been largely ignored by those who claim a mid-Holocene highstand based on reef terrace data.

Second, on line 290 it is claimed that in contrast to individual coral species, which have large depth ranges, "...Accurate palaeo-water depth intervals can be usually defined based on coral assemblages..." I think you mean 'more precise' depth intervals can be defined from coral assemblages. This claim is also problematic given that it requires that corals be identified to species level, and that the species concept in Scleractinia is debatable due to hybridization in space and time (see Veron's '95 book on the subject). So individual coral depth ranges and coral assemblages defined from modern reefs are unlikely to be applicable to fossil reefs from the LIG.

The truth is that only by combining coralgal assemblages with facies analysis can you precisely and accurately determine SL from fossil reefs. And that requires determining the facies sequence in modern reefs which has been largely ignored due to the difficulty of drilling in zones with high wave energy. You might argue that facies sequences are subject to the same caveat, but facies are controlled by both physical and ecological interactions and are therefore better conserved in time than coral biology. An additional difficulty in comparing modern vs fossil coral assemblages is the problem of taphonomic bias identified from the comparison of life and death assemblages. It is common that fossil assemblages have significant differences in the proportions of branching and fragile corals which are commonly removed by tropical cyclones. Again such problems could be avoided by comparing sub-surface facies on modern reefs, rather than their surface assemblages.

Finally a comment on the interpretation of the data set from the Huon Peninsula. As I state in my 2011 review: "...It is clear ... that the interpretation of LIG reef architecture in the Huon Peninsula has been obfuscated by various attempts to obtain a terrace chronology. The fundamental problem has been that unreliable U-series age determinations, which clearly fail any test of stratigraphic consistency, have driven the analysis of the stratigraphy itself, rather than the other way round. This has led to disconformities being inferred between stratigraphically inverted ages, or inverted ages simply being ignored, rather than acknowledging that many dates have large true-age variabilities and that more than 50% of those with "reliable" isotopic signatures have discordant 231Pa ages (Scholz and Mangini, 2007). As a result, SL histories determined from these inadequate data are premature, especially interpretations of extreme excursions based on ages alone (Esat et al., 1999)."

In summary, these additional uncertainties should be added to the discussion at the end of the paper.
* * *

---

## Author Comment (AC1) · 17 Dec 2020

This review paper relies on data that have been produced by various authors since several decades. The section 6.6 details the uncertainties and the data quality concerning the 284 data points that have been used. In this section, we have stressed that the data quality is mostly affected by uncertainties related to RSL data and, to a lesser extent, to age information. Section 7 concerns recommendations for future research in this region, especially regarding better constraints on RSL data, the establishment of an accurate chronology of LIG reef deposits and the study of their development patterns.

We notice that most of P. Blanchon's comments raise general issues that are beyond the scope of this paper, including the species concept in Scleractinia and eventually

the use of coral assemblages to deduce palaeo-water depths in the fossil record.

Hybridisation was traditionally considered rare on coral reefs. However, a rapid increase in hybrid studies over the last 20 years has revealed that hybridisation on coral reefs is common and widespread. However, to our knowledge, the impact of such processes on taxonomy of fossil corals has not been evaluated. No reference other than Veron's seminal book is given in P. Blanchon's comment.

In his comment, P. Blanchon questions the use of coral assemblages to reconstruct sea-level change in the fossil record and states that coralgal assemblages must be only used in combination with facies analysis. We disagree with that statement. The composition and distribution of reef assemblages typically reflect environmental parameters such as light conditions, turbidity, water energy and nutrient levels because reef dwelling organisms are sensitive to subtle ecological changes affecting their environment (Camoin and Webster, 2015). Over the two last decades, in situ reef assemblages, including corals and associated biota (i.e., coralline algae, vermetid gastropods and encrusting foraminifera) rather than monospecific coral communities, have increasingly been used to more accurately constrain palaeo-water depths as well as other important palaeo-environmental parameters (see review in Camoin and Webster, 2015). Interpretation of fossil assemblages of corals and associated biota relies on the occurrence, ecology and zonation of modern counterparts, and the presumption that similar factors controlled their distribution. To do that, one does not have to identify at the species level every single member of that fossil assemblage, but only the dominant species or morpho-groups (closely related species with similar morphological characteristics) indicative of a specific modern coral community and reef setting.

In our review of MIS 5e reef deposits from tropical Pacific Islands, data reported in the literature do not include details regarding the composition of reef assemblages and concern coral identification mostly at the genus or family levels. This introduces generally very large RSL uncertainties and excludes any constraint regarding the RSL position during MIS 5e. Our Table 2 summarizes the best estimates of palaeo-water depth

intervals for corals that were quoted in the literature, based on the OBIS and other databases that list modern counterparts from the whole Indo-Pacific region. With two exceptions, all concerned islands/groups of islands are not characterized by macro-tidal ranges.

Facies sequences are also an important component to reconstruct sea-level change and have been considered in parallel to the analysis of reef assemblages in many studies focused on fossil reefs from the Indo-Pacific (see Camoin and Webster, 2015 and references therein). However, it is difficult to assess which additional information related to facies sequences could be obtained from the subsurface of modern reefs compared to their surface, as the vertical growth of modern reefs has been limited since the end of the last deglacial sea-level rise.

"The truth is rarely pure and never simple" (Oscar Wilde)

---

## Referee Comment (RC1) · Clark Sherman (Referee) · 21 Dec 2020

This paper provides a thorough compilation of last interglacial (LIG)/marine isotope stage 5e (MIS 5e) relative sea-level (RSL) data derived from reef records on Pacific Islands covering a broad range of tectonic and oceanographic settings. The authors have done an excellent job in collecting, organizing, describing and to some extent evaluating this voluminous dataset. I have only a few comments for consideration related to the evaluation of the data and in the organization of the manuscript.

The description of RSL indicators given in Section 3 and Table 1 seems somewhat narrow keying on "the highest in situ corals". While coral assemblages are mentioned as providing more accurate indicators of paleosea levels, this does not seem to be part of

the final evaluation of the data, which appears restricted to assessing the depth ranges of individual species (or genera) and how precise they are in establishing past sea level positions. Additionally, there is no mention of coralline algae as important indicators of paleosea level, though realistically there is likely rare mention of coralline algae in the LIG studies examined. I would echo earlier comments by Blanchon that facies analysis is another critical component of paleosea level determination in conjunction with identification of coral-algal assemblages. Recognizing that the studies reviewed in this effort provide a broad range of sedimentological and paleoecological data, from very little to very thorough, evaluating the validity of each paleosea level determination is a daunting task. However, I think it would be worthwhile to establish what an "ideal" dataset would consist of so that studies may be evaluated based upon how close they come to achieving this ideal.

Assessing the quality of the radiometric ages based on their margin of error is a practical approach. Those ages with very narrow margins of error are predominantly more recently determined by mass spectrometric techniques and are likely the most accurate. Still, there is no mention of geochemical criteria (e.g., U and Th concentrations, initial $\delta$234-U, etc.) for evaluating closed versus open system behavior of radioisotopes. These criteria are probably only practical in evaluating more recent studies (the last 20 years or so) where this information is provided, but are extremely important in establishing quality of radiometric ages. Potential problems of recrystallization of original coral-skeletal aragonite to calcite are discussed in Section 6.6 and the authors importantly note that this information is lacking in many early studies. In previous sections of the manuscript ages are often described as 'reliable' or 'accepted' based upon a reported calcite content (indicative of recrystallization) of <3%. Calcite content is a critical first filter for U-Th ages. However, even samples that pass this mineralogic test can later be found to exhibit open-system behavior with respect to U and/or Th when precise geochemical measurements are conducted as part of the dating process (i.e., the geochemical criteria that I mention above).

I am not sure if there are formatting guidelines that dictate the order of the different sections. However, I would recommend placing "Section 6.6 - Uncertainties and data quality" near the beginning of the manuscript, at least prior to Section 5. In reading through the paper and seeing ages or RSL indicators deemed as reliable or accepted, I kept wondering what criteria were being used. I did not encounter this info until near the very end of the manuscript. I think it would be more useful up front. Section 5 might be better titled as "Relative sea-level records", to better differentiate it from Section 3. Finally in Section 3, I was confused by the statement: "Records which had the following characteristics were excluded from the analysis: (1) those for which the difference between Max Depth and Min Depth were > 0,2 m". Forgive me, but I don't understand what is being said here. Is this ecologic range?

---

## Author Comment (AC2) · 12 Jan 2021

We thank Clark Sherman for his comments on our review of MIS 5e reef deposits from tropical Pacific Islands.

**RSL indicators**

*In situ* reef assemblages, including corals and associated biota (i.e., coralline algae, vermetid gastropods and encrusting foraminifera), provide more accurate constraints on palaeo-water depths compared to reconstructions based only on individual coral species or genera. This approach has been increasingly applied over the two last decades, especially regarding the study of last deglacial and Holocene reef sequences

(see review in Camoin and Webster, 2015). However, the dataset that supports our review of MIS 5e reef deposits from tropical Pacific Islands, does not include details regarding the composition of reef assemblages and concerns coral identification mostly at the genus or family levels. The analysis of RSL indicators given in Section 3 and Table 1 is focused on "the highest *in situ* corals" and is seemingly narrow as it is restricted to assessing depth ranges of individual coral taxa. Table 1 follows the template of the database, but the need to identify reef assemblages and RSL indicators with a narrow depth range to better constrain RSL reconstructions is already mentioned in Section 6.6 'Uncertainties and data quality'; in addition, requirements for future studies are listed in Section 7 'Future research directions'. However, we will specifically mention in the revised manuscript the importance of studying the associated biota (i.e., coralline algae, vermetid gastropods and encrusting foraminifera) as well as facies analysis to reconstruct depositional environments, thus following recommendations made by Camoin and Webster (2015) regarding the establishment of an "ideal" dataset. The overall quality of the observations in the database appears poor, mainly due to the lack of constraints on the palaeo-water depth range of coral taxa, which introduces large uncertainties and limits the accuracy of RSL reconstructions during MIS 5e. This remark may apply to a number of past studies of Quaternary reef systems.

**Quality of radiometric ages**
Evaluation of the quality of radiometric ages includes the mineralogical assessment, but also the closed versus open system behaviour of radioisotopes. Datapoints obtained outside closed-system conditions are highlighted in the database. The information on the acceptance of radiometric ages has been often directly taken from the original publication. In addition, the quality of the ages that are reported in the WALIS database has been reviewed by Peter Chutcharavan and Andrea Dutton who compiled a U-series database for MIS 5e corals (see this Special Issue, manuscript in review).

**Depth range of taxa**

We clarify hereafter the statement "Records which had the following characteristics were excluded from the analysis: (1) those for which the difference between Max Depth and Min Depth were > 0,2 m". In the OBIS database the "Max Depth" and "Min Depth" values define the depth range of each record. In this database, if a coral taxon is observed between 0 and 10 m, then the Min Depth would be 0 and the Max Depth would be 10. To be able to graphically represent the depth distribution of each coral taxon and see where it peaked, we have selected records associated with a specific depth (or nearly so – with a limit set arbitrarily to 0.2 m), following the method used by Hibbert et al. (2016, 2018). For the maximum depths at which the taxa have been recorded, the whole data set has been considered.

**Organization of the manuscript**

Concerning the organization of the manuscript, the order of the different sections follows formatting guidelines.

---

## Short Comment (SC2) · 17 Jan 2021

I think the tropical Pacific islands are a key area for sea-level reconstruction. I was curious to see the data delivered by these islands for the late Pleistocene. I was puzzled by the approach used here: the sea-level indicator is a coral-reef terrace; its IR is determined from the average water depth of a single coral species; the two selected databases (OBIS, IUCN) used to determine the average water depth provide similar, but in some cases also very different minimum and maximum living ranges for individual species. As a result, the IR of the indicator range from 2.7 m to 30.0 m. For the Yucatan peninsula Simms (2020) uses the same indicator and determines an IR ranging 1.0 – 9.1 m because his corals grow in Atlantic waters and he follows Hibbert et al 2016. For Rovere et al. (2016) the IR of the terrace range from the lower low water

down to the "end of forereef" and the wave-breaking depth, respectively. Knowing that the islands are different in terms of their geological setting, the generalised indicator ("coral reef terrace, general description") prompts questions. Do the Pacific tropical islands all show a coral-reef terrace and is the flat surface of the individual terrace indeed the surrogate for the contemporary sea level? Low-lying reef islands (e.g. Marshall Islands) are often characterised by net gain of sediment mostly through ocean over-wash especially during highstand (e.g. Tuck et al. 2019; Geology). For the Tuamotu atolls Camoin et al. (2001) describe three different facies zones that would characterise the reef flat, each with a slightly different relationship to energy and nutrient supply, hence local parameters. Do these facies zones respond coherently to the change of accommodation space? The spur and groove studies (e.g., da Silva et al., 2020; Coral Reefs) do not support this idea. The reef terrace may be a useful indicator when the uplift is faster than the eustatically-induced change of accommodation space. Therefore, using a coral terrace for calculating the uplift rate of an emerging shelf situated at the plate boundary (e.g., Huon Peninsula) may be a reasonable approach. Gino de Gelder and others show that the hypothesis can be tested. Looking at the studies of Grigg (1982), Perry et al. (2011, 2012), Tuck et al. (2019), Montaggioni et al. (2019), Masselink et al. (2020) or Duce et al (2020; QSR) – all draw a highly divers picture of coral-reef islands. This suggests that the palaeo-sea level should not be calculated from a single datapoint and that the reef terrace should not be the sole indicator. One size doesn't fit all.

---

## Short Comment (SC3) · 23 Jan 2021

As guest editor of the WALIS SI, I would like to thank Paul Blanchon and Barbara Mauz for taking the time to comment on this paper.

Paul Blanchon brings up a potential weakness with the approach we propose for coral reef terraces in Rovere et al., 2016 (later modified and coded by Lorscheid and Rovere, 2019). I remain convinced that, for most reefs globally, the indicative range we propose for coral reef terraces (MLLW to Breaking depth) is reasonable. But I am aware that, as Blanchon points out, there are exceptions (dealing with natural environments, that's hardly a surprise).

It is worth pointing out that we tried to be clear in both papers cited above that the

"remote" indicative meaning (i.e., based only on hydrodynamic considerations), should only be adopted only when no quantitative data on modern analogs is available (for example, see Rovere et al., 2016, page 423, subpoint "Modern analog" and Lorscheid and Rovere, page 6). For this particular MS, I would suggest the authors highlight this potential caveat in the text, maybe advising the readers of the limits of this "remote" approach, inviting future researchers in these areas to properly address the modern analog problem.

Unfortunately, many studies on LIG indicators do not report modern analog quantitative information (a practice that is instead more consolidated in Holocene sea-level studies), therefore I suspect that many authors in WALIS will use the "remote" indicative meaning. For this reason, we will try to re-iterate the limitations of this approach in the editorial that will collate all the contributions, to properly inform readers. For which concerns the use of single corals as RSL indicators (brought up by both Blanchon and Mauz), I would point out to another paper in discussion in ESSD, where this issue is also partially addressed: https://essd.copernicus.org/preprints/essd-2020-381/. Also, the pros and cons of this approach are also highlighted in Rovere et al., 2016. We will also try to elaborate on this point within the upcoming editorial, keeping in mind that there is no one-fits-all approach to establish the indicative meaning of LIG datapoints, with local conditions and preservation often dictating the choice.

One further consideration (that might also apply to parts of the comment by Barbara Mauz) is that WALIS aims to collate data into a unique, standardized database. As already stressed in Rovere et al. (2016), the quantification of the indicative meaning for LIG proxies should be regarded as a "geological interpretation", which must be reported separately from other primary data (e.g., elevation). This is implemented in the WALIS structure. When all the data will be standardized within a single database, it will be relatively easy for any end-user to back-calculate paleo RSL from the primary data using different indicative ranges for selected proxies, in case better modern analog data or better interpretations will become available. It will be also possible to

group several indicators within the same area/outcrop and apply statistical analyses to condense several indicators into a single paleo RSL estimate. This will be up to the end-users: in this sense, WALIS aims to be a starting point for further studies, following the example set by other existing databases (I am thinking, for example, of PALEO DB, https://paleobiodb.org/ or SISAL, https://researchdata.reading.ac.uk/256/).

Overall, I take note of this interesting discussion and I will try to highlight these aspects in the editorial that will close the SI (that will be co-authored by the SI guest editors). Thank you again for sharing your thoughts.

Alessio Rovere

References cited: Lorscheid, T. and Rovere, A., 2019. The indicative meaning calculator–quantification of paleo sea-level relationships by using global wave and tide datasets. Open Geospatial Data, Software and Standards, 4(1), pp.1-8.

Rovere, A., Raymo, M.E., Vacchi, M., Lorscheid, T., Stocchi, P., Gomez-Pujol, L., Harris, D.L., Casella, E., O'Leary, M.J. and Hearty, P.J., 2016. The analysis of Last Interglacial (MIS 5e) relative sea-level indicators: Reconstructing sea-level in a warmer world. Earth-Science Reviews, 159, pp.404-427.

―――――――――――――――――

---

## Author Comment (AC3) · 29 Jan 2021

We would like to thank Barbara Mauz for her comments on our review of MIS 5e reef deposits from tropical Pacific Islands.

We agree that tropical Pacific islands are a key area for sea-level reconstruction. As stated in section 7 'Future Research Directions', "*RSL change reconstructions require the combination of reliable radiometric ages and elevation measurements, as well as an accurate estimate of palaeo-water depths deduced from the modern distribution of relevant reef communities*". However, our newly compiled database (Hallmann and Camoin, 2020), which is based on data from about 300 published papers and

284 data points from 35 studies concerning MIS 5e, has demonstrated that most of the studies that have been carried out on tropical Pacific islands do not satisfy these requirements, thus hampering the accurate reconstruction of LIG RSL history. In the reviewed studies, potential sea-level indicators are restricted to corals that are generally identified at the genus or family levels and no modern analog quantitative information is reported. In addition, sedimentological and morphological data are barely described, thus hampering the accurate reconstruction of the relevant reef systems.

Mauz' comment: "*the sea-level indicator is a coral-reef terrace; its IR is determined from the average water depth of a single coral species; the two selected databases (OBIS, IUCN) used to determine the average water depth provide similar, but in some cases also very different minimum and maximum living ranges for individual species. As a result, the IR of the indicator range from 2.7 m to 30.0 m. For the Yucatan peninsula Simms (2020) uses the same indicator and determines an IR ranging 1.0 – 9.1 m because his corals grow in Atlantic waters and he follows Hibbert et al. (2016)*", does not refer to a specific coral taxon, so that the issue cannot be precisely identified. The depth range of 2.7 m to 30.0 m that is indicated suggests that this could concern *Porites lutea* (see Table 2 in our manuscript). However, Simms (2020) uses depth ranges published in Hibbert et al. (2016) for *Montastrea annularis*, *Acropora palmata* and *Acropora cervicornis* (derived from the OBIS database), but does not refer to *Porites lutea*.

We have summarized the best estimates of palaeo-water depth intervals for corals that were quoted in the literature, based on the OBIS and other databases (e.g., IUCN) that list modern counterparts from the whole Indo-Pacific region. IUCN reports only the maximum and minimum depths for each species, based on published data. OBIS is a more detailed database, which allows to define distribution curves and to predict

the depth ranges at which the relevant species can be found with the highest probability.

No study related to LIG reef systems from the tropical Pacific islands includes a detailed description of *in situ* reef assemblages that is required to more accurately constrain palaeo-water depths, as it has been demonstrated over the last two decades by the pioneer works of Montaggioni et al., 1997; Cabioch et al., 1999a, b and Camoin et al., 1999 (see review in Camoin and Webster, 2015). This therefore implies that the "RSL from single coral" description scheme needs to be improved and that there is no "one fits all" approach when it comes to sea-level data standardization, especially when data is older and not all metadata is reported. We agree with Alessio Rovere that "*'remote' indicative meaning (i.e., based only on hydrodynamic considerations), should only be adopted only when no quantitative data on modern analogs is available*".

In section 7 of our review, we state that "*Future research directions may therefore require to revisit LIG reef records from tropical Pacific islands, especially the key 'reference sites' (e.g., Papua New Guinea, Hawaii, Vanuatu), in order to collect the missing information that is crucially needed to reconstruct properly LIG RSL changes*". We note that Alessio Rovere stated in his comment that "*(. . .) when all the data will be standardized within a single database, it will be relatively easy for any end-user to back-calculate paleo RSL from the primary data using different indicative ranges for selected proxies, in case better modern analog data or better interpretations will become available*". This demonstrates that the standardized MIS 5e database provides an excellent tool for future LIG sea-level research as any missing information can be added to the WALIS database at a later stage for a more accurate reconstruction of MIS 5e RSL changes.

---

## Author Comment (AC4) · 29 Jan 2021

We thank Alessio Rovere for his feedback on the comments by Paul Blanchon and Barbara Mauz.

We have highlighted in our manuscript that: "*RSL change reconstructions require the combination of reliable radiometric ages and elevation measurements, as well as an accurate estimate of palaeo-water depths deduced from the modern distribution of relevant reef communities*". The latter has been demonstrated over the last two decades by the pioneer works of Montaggioni et al. (1997), Cabioch et al. (1999a,b) and Camoin et al. (1999), and then summarized in Camoin and Webster (2015). Unfortunately, most

of the studies that have been carried out so far on tropical Pacific islands do not fulfill these requirements, thus implying that these LIG reef systems will have to be studied in more details with such a perspective.

———————————

---

## Referee Comment (RC2) · Blake Dyer (Referee) · 9 Apr 2021

This work by Hallman et al. represents a significant effort to organize and contextualize decades of MIS 5e Pacific Island data. This effort will be invaluable at facilitating future research on MIS 5e sea level and Pacific Island record of MIS 5e. I would confidently recommend this database and manuscript, as is, to any student or researcher looking for a starting point on MIS 5e in the Pacific. However, upon reflecting on the manuscript and previous comments in the discussion, it is clear that some of the detail is lost in the conversion of decades of field work to site-by-site RSL interpretations.

I have some reservations that the 84 summary RSL indicators will be used without any consideration of the assumptions made in the conversion. My understanding is that

the authors have followed well documented approaches to estimate paleo-RSL, and that these approaches are standardized throughout the WALIS database. I suspect experts will have no trouble navigating where each paleo-RSL number comes from, but the text could offer more guidance and caution to non-experts. I note that the authors have clearly considered this issue and adressed it in the text, so I only have a few minor suggestions that they may wish to incorporate:

First, if we consider a single site as an example: 5.1 French Polynesia. The text presents a logical narrative from data (ages and species or species assemblages) to RSL. Paraphrasing from Lines 356-359: ..past workers reported an elevation of Leptoria at 3.85 m, Leptoria lives today down to 7.5 m, and so paleo-RSL is 8.75 +/- 2.3. As a non-expert or someone not intimately familiar with WALIS, trying to unravel the conversion to paleo-RSL is impossible. Moreover, the details will differ some for each of the 84 summary sites. I suggest that the introduction to section 5 include a general workflow from data to paleo-RSL interpretations. This small addition could include a reference to the online WALIS documentation or any other manuscripts within this special issue that may document more fully the process. An additional sentence after line 44 in the introduction could be helpful too, although my next comment is related to this possible approach.

Second, the authors address the uncertainties and data quality extensively in section 6.6. I do believe this discussion is appropriately detailed. However, I strongly agree with the comment from Clark that the information seems to come too late in the manuscript. The authors reply to that comment mentions that the format is fixed, so I would encourage the authors to include in the introduction a very brief summary of the uncertainties and challenges associated with estimating paleo-RSL from the data in the database. This section could refer readers to section 6.6 for a more complete discussion.

---

## Author Comment (AC5) · 1 May 2021

We thank Blake Dyer for his comments on our review of MIS 5e reef deposits from tropical Pacific islands.

**Estimation of palaeo-RSL**
Blake Dyer mentioned that the manuscript needs more guidance for users who are non-experts or not familiar with WALIS as the conversion to palaeo-RSL is not clear. We followed his suggestion and added a figure to the introduction to Section 5 - 'Relative sea-level records' explaining the general workflow from reported palaeo-elevations to the estimated palaeo-RSL. Furthermore, we revised Section 1 - 'Introduction' by explaining the structure of the manuscript, i.e., the content of the different sections, to

better guide the reader.

**Uncertainties and data quality**

As both referees stated that Section 6.6 - 'Uncertainties and data quality' comes too late in the manuscript, we followed Blake Dyer's suggestion and added a brief summary of the uncertainties and data quality to Section 1 - 'Introduction' while referring the reader to Section 6.6 for more details. Concerning the quality of the ages, we also added to Section 1 that the "information on the acceptance of radiometric ages has been often directly taken from the original publication" and we refer to Chutcharavan and Dutton (this Special Issue) who reviewed the quality of the ages.

---

## Author Response (AR1)

**RESPONSES TO REFEREES**

*Colour code:*

*Blue = Referees' comments*

*Black = Authors' responses*

**RESPONSES TO REFEREE #1: Clark Sherman**

**1) RSL indicators**

*"description of RSL indicators given in Section 3 and Table 1 seems somewhat narrow keying on "the highest in situ corals""*

*"final evaluation of the data, which appears restricted to assessing the depth ranges of individual species (or genera)"*

*"there is no mention of coralline algae as important indicators of paleosea level"*

*"facies analysis is another critical component of paleosea level determination in conjunction with identification of coral-algal assemblages"*

*In situ* reef assemblages, including corals and associated biota (i.e., coralline algae, vermetid gastropods and encrusting foraminifera), provide more accurate constraints on palaeo-water depths compared to reconstructions based only on individual coral species or genera. However, the dataset that supports our review of MIS 5e reef deposits from tropical Pacific Islands, does not include details regarding the composition of reef assemblages and concerns coral identification mostly at the genus or family levels.

The revised manuscript (Sections 6.6 - 'Uncertainties and data quality' and 7 - 'Future research directions') mentions the importance of studying the associated biota (i.e., coralline algae, vermetid gastropods and encrusting foraminifera) as well as facies analysis to reconstruct depositional environments, thus following recommendations made by Camoin and Webster (2015) regarding the establishment of an "ideal" dataset.

**2) Quality of radiometric ages**

*"Calcite content is a critical first filter for U-Th ages. However, even samples that pass this mineralogic test can later be found to exhibit open-system behavior with respect to U and/or Th when precise geochemical measurements are conducted as part of the dating process"*

Evaluation of the quality of radiometric ages includes the mineralogical assessment, but also the closed versus open system behaviour of radioisotopes. Datapoints obtained outside closed-system conditions are highlighted in the database. The information on the acceptance of radiometric ages has been often directly taken from the original publication. In addition, the quality of the ages that are reported in the WALIS database has been reviewed by Peter Chutcharavan and Andrea Dutton who compiled a U-series database for MIS 5e corals (see this Special Issue, manuscript in review).

This clarification has been added to Section 1 - 'Introduction'.

**3) Depth range of taxa**

*"I was confused by the statement: "Records which had the following characteristics were excluded from the analysis: (1) those for which the difference between Max Depth and Min Depth were > 0,2 m"."*

In the OBIS database the "Max Depth" and "Min Depth" values define the depth range of each record. In this database, if a coral taxon is observed between 0 and 10 m, then the Min Depth would be 0 and the Max Depth would be 10. To be able to graphically represent the depth distribution of each coral taxon and see where it peaked, we have selected records associated with a specific depth (or nearly so – with a limit set arbitrarily to 0.2 m), following the method used by Hibbert et al. (2016, 2018). For the maximum depths at which the taxa have been recorded, the whole data set has been considered.

**4) Organization of the manuscript**

*"I would recommend placing "Section 6.6 - Uncertainties and data quality" near the beginning of the manuscript, at least prior to Section 5. In reading through the paper and seeing ages or RSL indicators deemed as reliable or accepted, I kept wondering what criteria were being used."*

We added a brief summary of the uncertainties and data quality to Section 1 - 'Introduction' while referring the reader to Section 6.6 for more details. Concerning the quality of the ages, we also added to Section 1 that the "information on the acceptance of radiometric ages has been often directly taken from the original publication" and we refer to Chutcharavan and Dutton (this Special Issue) who reviewed the quality of the ages.

*"Section 5 might be better titled as "Relative sea-level records", to better differentiate it from Section 3."*

Changed.

**RESPONSES TO REFEREE #2: Blake Dyer**

**1) Estimation of palaeo-RSL**

*"As a non-expert or someone not intimately familiar with WALIS, trying to unravel the conversion to paleo-RSL is impossible."*

*"I suggest that the introduction to section 5 include a general workflow from data to paleo-RSL interpretations."*

*"An additional sentence after line 44 in the introduction could be helpful too"*

We added a new figure (Figure 14) to the introduction to Section 5 - 'Relative sea-level records' explaining the general workflow from reported palaeo-elevations to the estimated palaeo-RSL. Furthermore, we revised Section 1 - 'Introduction' by explaining the structure of the manuscript, i.e., the content of the different sections, to better guide the reader.

**2) Uncertainties and data quality**

*Section 6.6: "I strongly agree with the comment from Clark that the information seems to come too late in the manuscript."*

*"I would encourage the authors to include in the introduction a very brief summary of the uncertainties and challenges associated with estimating paleo-RSL from the data in the database. This section could refer readers to section 6.6 for a more complete discussion."*

Both referees stated that Section 6.6 - 'Uncertainties and data quality' comes too late in the manuscript. See reply to Clark Sherman's comment 4).

**ADDITIONAL CHANGES**

- Data points for Saipan, Northern Mariana Islands, from Muhs et al. (2020) have been added to the WALIS database and accordingly text has been added to Sections 2, 5 and 6.1. A map showing the site locations (Figure 12) has been added.

- RSL data points for Oahu, Hawaii, from Hearty et al. (2007) have been added to the WALIS database and accordingly text has been added to Sections 2 and 5.

- New coral taxa have been added to Table 2.

- References for the Mariana Islands have been added to Section 6.4

- Two tables (Tables 8 and 9) have been added to Section 6.6 to provide a better overview of the evaluation of the RSL data and the age information.

- Six references have been added to the list of references.